# Greedy Pruning with Group Lasso Provably Generalizes for Matrix Sensing

**Nived Rajaraman**
EECS Department
University of California, Berkeley
nived.rajaraman@eecs.berkeley.edu

**Devvrit**
CS Department
The University of Texas at Austin
devvrit.03@gmail.com

**Aryan Mokhtari**
ECE Department
The University of Texas at Austin
mokhtari@austin.utexas.edu

**Kannan Ramchandran**
EECS Department
University of California, Berkeley
kannanr@eecs.berkeley.edu

## Abstract

Pruning schemes have been widely used in practice to reduce the complexity of trained models with a massive number of parameters. In fact, several practical studies have shown that if a pruned model is fine-tuned with some gradient-based updates it generalizes well to new samples. Although the above pipeline, which we refer to as pruning + fine-tuning, has been extremely successful in lowering the complexity of trained models, there is very little known about the theory behind this success. In this paper, we address this issue by investigating the pruning + fine-tuning framework on the overparameterized matrix sensing problem with the ground truth $U_\star \in \mathbb{R}^{d \times r}$ and the overparameterized model $U \in \mathbb{R}^{d \times k}$ with $k \gg r$. We study the approximate local minima of the mean square error, augmented with a smooth version of a group Lasso regularizer, $\sum_{i=1}^{k} \|U e_i\|_2$. In particular, we provably show that pruning all the columns below a certain explicit $\ell_2$-norm threshold results in a solution $U_{\text{prune}}$ which has the minimum number of columns $r$, yet close to the ground truth in training loss. Moreover, in the subsequent fine-tuning phase, gradient descent initialized at $U_{\text{prune}}$ converges at a linear rate to its limit. While our analysis provides insights into the role of regularization in pruning, we also show that running gradient descent in the absence of regularization results in models which are not suitable for greedy pruning, i.e., many columns could have their $\ell_2$ norm comparable to that of the maximum. To the best of our knowledge, our results provide the first rigorous insights on why greedy pruning + fine-tuning leads to smaller models which also generalize well.

## 1 Introduction

Training overparameterized models with a massive number of parameters has become the norm in almost all machine learning applications. While these massive models are successful in achieving low training error and in some cases good generalization performance, they are hard to store or communicate. Moreover, inference with such large models is computationally prohibitive. To address these issues, a large effort has gone into compressing these overparameterized models via different approaches, such as quantization schemes [1, 2], unstructured [3] and structured [4–6] pruning mechanisms, and distillation techniques using student-teacher models [7, 8]. Among these approaches, *greedy pruning*, in which we greedily eliminate the parameters of the trained model

37th Conference on Neural Information Processing Systems (NeurIPS 2023).

based on some measure (e.g., the norms of the weight vectors associated with individual neurons) has received widespread attention [3, 9–13]. This is mostly due to the fact that several practical studies have illustrated that training an overparameterized model followed by greedy pruning and fine-tuning leads to better generalization performance, compared to an overparameterized model trained without pruning [14]. Furthermore, a phenomenon that has been observed by several practical studies, is that different forms of regularization during training, such as $\ell_0$ or $\ell_1$ regularization [15–17] or $\ell_2$ regularization including group Lasso [18] lead to models that are better suited for pruning, and leading to better generalization post fine-tuning. While the greedy pruning framework has shown impressive results, there is little to no theory backing why this pipeline works well in practice, nor understanding of the role of regularization in helping generate models which are suitable for greedy pruning. In this work, we address the following questions:

> *Does the greedy pruning + fine-tuning pipeline provably lead to a simple model*
> *with good generalization guarantees? What is the role of regularization in pruning?*

In this paper, we use the symmetric matrix sensing problem [19] as a test-ground for the analysis of greedy pruning framework, a model very closely related to shallow neural networks with quadratic activation functions [20, 19]. In this setting, the underlying problem for the population loss (infinite samples) is defined as $\|UU^T - U_\star U_\star^T\|_F^2$, where $U_\star \in \mathbb{R}^{d \times r}$ is an unknown ground-truth rank-$r$ matrix, with $r$ also being unknown, and $U \in \mathbb{R}^{d \times k}$ is the overparameterized learning model with $k \gg r$. As we discuss in the Appendix F, the columns of $U$ can be thought of as the weight vectors associated with individual neurons in a 2-layer shallow neural network with quadratic activation functions, a connection first observed in [21]. Thus, the data generating model has $r$ neurons, while the learner trains an overparameterized model with $k$ neurons.

While the statistical and computational complexity of the overparameterized matrix sensing problem has been studied extensively, we use it as a model for understanding the efficacy of greedy pruning. In particular, we aim to answer the following questions: Does there exist a simple pruning criteria for which we can *provably* show that the pruned model generalizes well after fine-tuning *while having the minimal necessary number of parameters*? What is the role of regularization during training in promoting models which are compatible with greedy pruning? Finally, what generalization guarantees can we establish for the pruned model post fine-tuning?

**Contributions.** Our main contribution is to show that the discussed pruning pipeline not only recovers the correct ground-truth $U_\star U_\star^T$ approximately, but also automatically adapts to the correct number of columns $r$. In particular, we show that training an overparameterized model on the empirical mean squared error with an added group Lasso based regularizer to promote column sparsity, followed by a simple norm-based pruning strategy results in a model $U_{\text{prune}}$ having exactly the minimum number of columns, $r$. At the same time, we show that $\|U_{\text{prune}} U_{\text{prune}}^T - U_\star U_\star^T\|_F^2$ is small, but non-zero. Hence, the pruned model can subsequently be fine-tuned using a small number of gradient steps, and in this regime, $\|U_t U_t^T - U_\star U_\star^T\|_F^2$ shows linear convergence to its limit. Moreover, the pruned model can be shown to admit finite sample generalization bounds which are also statistically optimal for a range of parameters. In particular, we show that to obtain a model $U_{\text{out}}$ that has exactly $r$ columns and its population error is at most $\|U_{\text{out}} U_{\text{out}}^T - U_\star U_\star^T\|_F \leq \varepsilon$, our framework requires $O(dk^2 r^5 + \frac{rd}{\varepsilon^2})$ samples, which is statistically optimal for sufficiently small $\varepsilon$. We should also add that our framework does not require any computationally prohibitive pre- or post-processing (such as SVD decomposition) for achieving this result.

While there are several works [22, 19, 23] establishing that gradient descent in the "exactly-parameterized" setting requires $O(rd/\varepsilon^2)$ samples to achieve a generalization error of $\varepsilon$, and converges linearly to this limit, the picture is different in the overparameterized setting. In [23], the authors showed that in the overparametrized setting, vanilla gradient descent requires $O(kd/\varepsilon^2)$ samples to achieve a generalization error of $\|U_{\text{gd}} U_{\text{gd}}^T - U_\star U_\star^T\|_F \leq \varepsilon$, degrading with the overparameterization of the model. Moreover the resulting solution does not have the correct column sparsity. In order to obtain a model which can be stored concisely, $U_{\text{gd}}$ has to be post-processed by computing its SVD, which is computationally expensive in the high dimensional regime.

As our second continuation, we show that use of explicit regularization to promote column sparsity while training is important to learn models suitable for greedy pruning. Specifically, we show that while implicit regularization [19, 24] suffices to learn models with the correct rank, these approaches learn solutions with a large number of columns having $\ell_2$-norms comparable to that of the maximum, even if $r = 1$. Hence, it is unclear how to sparsify such models based on the norms of their columns.

## 2 Setup and Algorithmic Framework

Given $n$ observation matrices $\{A_i\}_{i=1}^n$, in the matrix sensing framework, the learner is provided measurements $y_i = \langle A_i, U_\star U_\star^T \rangle + \varepsilon_i$ where $\langle \cdot, \cdot \rangle$ indicates the trace inner product and $\varepsilon_i$ is measurement noise assumed to be distributed i.i.d. $\sim \mathcal{N}(0, \sigma^2)$[1]. Here $U_\star \in \mathbb{R}^{d \times r}$ is the unknown parameter, and the rank $r \leq d$ is unknown. The goal of the matrix sensing problem is to learn a candidate matrix $X$ such that $X \approx U_\star U_\star^T$. For computational reasons, it is common to factorize $X$ as $UU^T$ for $U \in \mathbb{R}^{d \times k}$. In this paper, we study the factored model in the overparameterized setting, where $k \gg r$. The empirical mean squared error is,

$$\mathcal{L}_{\text{emp}}(U) = \frac{1}{n} \sum_{i=1}^n \left( \langle A_i, UU^T \rangle - y_i \right)^2. \tag{1}$$

For the case that $A_i$'s are sampled entry-wise i.i.d. $\mathcal{N}(0, 1/d)$ and as $n \to \infty$, up to additive constants which we ignore, the population mean square error can be written down as,

$$\mathcal{L}_{\text{pop}}(U) = \|UU^T - U_\star U_\star^T\|_F^2. \tag{2}$$

There is an extensive literature on how to efficiently learn the right product $U_\star U_\star^T$ in both finite sample and population settings [25, 26]. In particular, there are several works on the efficiency of gradient-based methods with or without regularization for solving this specific problem [19, 23]. While these approaches guarantee learning the correct product $UU^T \approx U_\star U_\star^T$, in the overparameterized setting the obtained solutions are not column sparse and storing these models requires $\widetilde{\Theta}(kd)$ bits of memory (ignoring precision). As a result, in order to obtain a compressed solution $U_{\text{out}} \in \mathbb{R}^{d \times r}$ with the correct number of columns, one has to post-process $U$ and do a singular value decomposition (SVD), an operation which is costly and impractical in high-dimensional settings.

The goal of this paper is to overcome this issue and come up with an efficient approach to recover a solution $U_{\text{out}}$ which generalizes well, in that $\mathcal{L}_{\text{pop}}(U)$ is small, while at the same time having only a few non-zero columns, i.e. is sparse. Specifically, we show that via some proper regularization, it is possible to obtain a model that approximately learns the right product $U_\star U_\star^T$, while having only a few significant columns. As a result, many of its columns can be eliminated by a simple $\ell_2$ norm-based greedy pruning scheme, without significantly impacting the training loss. In fact, post pruning we end up with a model $U_{\text{prune}} \in \mathbb{R}^{d \times r}$ that has the correct dimensionality and its outer product $U_{\text{prune}} U_{\text{prune}}^T$ is close to the true product $U_\star U_\star^T$. When the resulting "exactly parameterized" model is fine-tuned, the generalization loss $\mathcal{L}_{\text{pop}}(U)$ can be shown to converge to $0$ at a linear rate.

To formally describe our procedure, we first introduce the regularization scheme that we study in this paper, and then we present the greedy pruning scheme and the fine-tuning procedure.

**Regularization.** The use of regularization for matrix sensing (and matrix factorization) to encourage a low rank solution [27–33], or to control the norm of the model for stability reasons [34, 35], or to improve the landscape of the loss by eliminating spurious local minima [36] has been well-studied in the literature. While these approaches implicitly or explicitly regularize for the rank of the learned matrix, the solution learned as a result is often not column sparse. Indeed, note that a matrix can be low rank and dense at the same time, if many columns are linear combinations of the others. We propose studying the following regularized matrix sensing problem with a group Lasso based regularizer [37, 18]. In the population case, the loss is defined as

$$\mathcal{L}_{\text{pop}}(U) + \lambda \mathcal{R}(U), \qquad \text{where} \quad \mathcal{R}(U) = \sum_{i=1}^k \|Ue_i\|_2. \tag{3}$$

Note that $\lambda > 0$ is a properly selected regularization parameter. Imposing $\mathcal{R}$ as a penalty on the layer weights of a neural network is a special case of a widely used approach commonly known as *Structured Sparsity Learning* (SSL) [12]. The regularizer promotes sparsity across "groups" as discussed by [37]. Here, the groups correspond to the columns of $U$. The use of matrix mixed-norms as a regularizer for sparsity was also considered in [38]. Furthermore, a connection of the group

---

[1]All proofs in the paper go through as long as the noise is i.i.d. sub-Gaussian with variance proxy $\sigma^2$. We study the Gaussian case for simplicity of exposition.

Lasso penalty with the Schatten-$1/2$ norm was considered in [39], where the authors show that stationary points of the factored objective also serve as stationary points of the unfactored objective and heuristically argue that the regularizer promotes sparse solutions.

As we prove in Section 4, the solution obtained by minimizing a smooth version of the regularized loss in (3), denoted by $U_{\text{out}}$, is approximately column-sparse, i.e., $k - r$ of its columns have small $\ell_2$ norm. As a result, it is suitable for the simple greedy pruning scheme which we also introduce in Algorithm 1. Interestingly, in Section 3, we will first show a negative result - the model obtained by minimizing the *unregularized* loss, $\mathcal{L}_{\text{pop}}(U)$ (eq. (2)) using gradient descent updates could fail to learn a solution which is suitable for greedy pruning. This shows the importance of adding some form of regularization during the training phase of the overparameterized model.

**Greedy Pruning.** The greedy pruning approach posits training a model (possibly a large neural network) on the empirical loss, followed by pruning the resulting trained network greedily based on some criteria. The resulting model is often fine-tuned via a few gradient descent iterations before outputting. In the literature, various criteria have been proposed for greedy pruning. Magnitude-based approaches prune away the individual weights/neurons based on some measure of their size such as $\ell_1/\ell_2$ norm of the associated vectors [16, 10].

In this work, we also focus on the idea of greedy pruning and study a mechanism to prune the solution obtained by minimizing the regularized empirical loss. Specifically, once an approximate second-order stationary point of the loss in Section 4, we only keep its columns whose $\ell_2$ norm are above a threshold (specified in Algorithm 1) and the remaining columns with smaller norm are removed. We further show that post pruning, the obtained model $U_{\text{prune}}$ continues to have a small empirical loss, i.e., small $\mathcal{L}_{\text{emp}}(U_{\text{prune}})$, while having exactly $r$ columns.

**Post-pruning fine-tuning.** As mentioned above, it is common to fine-tune the smaller pruned model with a few gradient updates before the evaluation process [40]. We show that the pruned model $U_{\text{prune}}$ has the correct rank and is reasonably close to the ground-truth model $U_\star$ in terms of its population loss. By running a few gradient updates on the mean square error, it can be ensured that $UU^T$ converges to $U_\star U_\star^T$ at a linear rate. Algorithm 1 summarizes the framework that we study.

## 3 Implicit regularization does not lead to greedy pruning-friendly models

In various overparameterized learning problems, it has been shown that first-order methods, starting from a small initialization, implicitly biases the model toward "simple" solutions, resulting in models that generalize well [41, 42], a phenomenon known as implicit regularization. In particular, for matrix sensing, [43, 19, 24] show that in the absence of any regularization, running gradient descent on the population loss *starting from a small initialization* biases $UU^T$ to low rank solutions, and learns the correct outer product $UU^T \approx U_\star U_\star^T$. However, as discussed earlier, low-rank solutions are not necessarily column sparse, nor is it clear how to sparsify them without computing an SVD. It is unclear whether implicit regularization suffices to learn models that are amenable for greedy pruning.

In this section, we address this question and show that minimizing the unregularized population loss $\mathcal{L}_{\text{pop}}$ leads to models which are not suitable for greedy pruning, i.e., have many columns with large $\ell_2$ norm. Specifically, we show that by running gradient flow from a small random initialization, even if the ground truth $U_\star$ is just a single column and $r = 1$, the learnt solution $U$ has a large number of columns that are "active", i.e. having $\ell_2$-norm comparable to that of the column with maximum norm. Thus, in the absence of the knowledge of $r$, it is unclear how to determine it from just observing the columns $\ell_2$ norm. We thus claim that such trained models are not compatible with greedy pruning. In the following theorem, we formally state our result.

**Theorem 1.** *Consider the population loss in eq. (2), for $r = 1$ and $k \gg 1$ and suppose $\|U_\star\|_{op} = 1$. Further, suppose the entries of the initial model $U_0$ are i.i.d. samples from $\mathcal{N}(0, \alpha^2)$, where $\alpha \leq c_1/k^3 d \log(kd)$ for some constant $c_1 > 0$. For another absolute constant $c_1' > 0$, as $t \to \infty$, the iterates of gradient flow with probability $\geq 1 - O(1/k^{c_1'})$ converge to a model $U_{gd}$ where $\widetilde{\Omega}(k^{c_1'})$ active columns satisfy*

$$\frac{\|U_{gd}e_i\|_2}{\max_{j \in [k]} \|U_{gd}e_j\|_2} \geq 0.99. \tag{4}$$

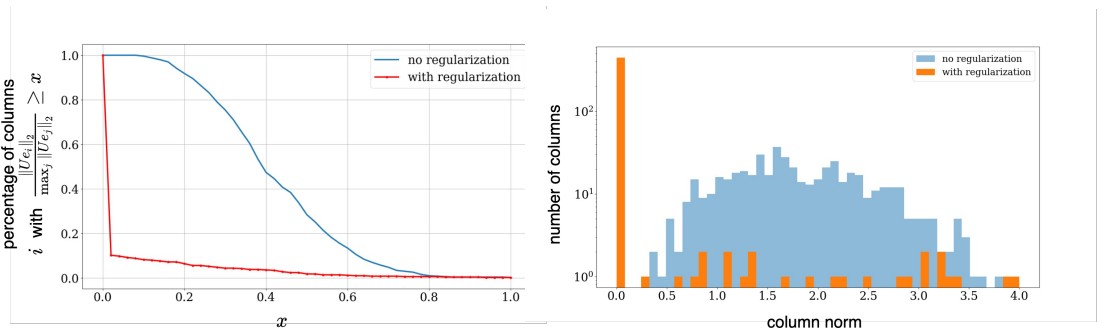

Figure 1: We run gradient updates to minimize the loss in eq. (2) until convergence. The model is highly overparameterized with $d = k = 500$ and $r = 4$. In the left figure, the vertical axis represents the fraction of columns having their norm at least $x$ times the largest norm across columns, with $x \in [0, 1]$ on the horizontal axis. In the right figure, for the same experiment we plot a histogram of column norms. Using regularization leads to solutions with most of the columns having small $\ell_2$ norm, and only a few significant columns. Without regularization, a large number of columns have their $\ell_2$ norm comparable to the largest column norm. Thus, training with explicit regularization leads to models that are more suitable for pruning.

Theorem 1 claims that the iterates of gradient descent converge to a solution for which a large number ($\Omega(k^{c'_1})$) of columns of $U_{\mathrm{gd}}$ have their $\ell_2$ norm at least $0.99$ times the maximum. An inspection of the proof shows that the constant $0.99$ is arbitrary and can be extended to any constant bounded away from $1$. Therefore, the models learnt by vanilla gradient descent cannot be reduced to the right sparsity level by simply pruning away columns with $\ell_2$ norm below any fixed constant threshold.

We provide a general guarantee in Theorem 22 (Appendix A). It is worth noting that from [19] that the learned model $U_{gd} \in \mathbb{R}^{d \times k}$ is guaranteed to achieve a small generalization error, i.e., $U_{gd} U_{gd}^T \approx U_\star U_\star^T$. Hence, the algorithm does learn the correct ground-truth product - what we show is that the learnt model is not "pruning-friendly" - it cannot be sparsified to the correct level by on any simple norm based pruning strategies. This is also observed in Figure 1: the model trained with implicit regularization has many significant columns and it is unclear how many to prune. We provide a proof sketch below, and refer the reader to Appendix A for more details.

**Proof sketch of Theorem 1.** Without loss of generality we assume in the proof that $\|U_\star\|_2 = 1$. Defining $U(t)$ as the iterate at time $t$, gradient flow on $\mathcal{L}_{\mathrm{pop}}$ follows the below dynamic,

$$\frac{dU}{dt} = -\nabla \mathcal{L}_{\mathrm{pop}}(U) = -(UU^T - U_\star U_\star^T)U, \tag{5}$$

where for ease of notation we drop the explicit dependence on $t$ in $U(t)$. In our proof, we exactly characterize the limiting point of gradient flow as a function of the initialization in a certain sense, which may be of independent interest. In particular, up to a small multiplicative error, we show,

$$\forall i \in [k], \ \|U_{\mathrm{gd}} e_i\|_2 \approx \frac{|\langle U_\star, U(0)e_i\rangle|}{\|U_\star^T U(0)\|_2}. \tag{6}$$

Also, with Gaussian initialization, $\langle U_\star, U(0)e_i\rangle \overset{\text{i.i.d.}}{\sim} \mathcal{N}(0, \alpha^2)$ across different values of $i$. In particular, we show that for some constant $c'_1 > 0$, $\widetilde{\Omega}(k^{c'_1})$ columns will have correlations comparable to the maximum, with $|\langle U_\star, U(0)e_i\rangle| \geq 0.99 \max_{j \in [k]} |\langle U_\star, U(0)e_j\rangle|$. For any of these columns, $i$,

$$\|U_{\mathrm{gd}} e_i\|_2 \approx \frac{|\langle U_\star, U(0)e_i\rangle|}{\|U_\star^T U(0)\|_2} \geq 0.99 \max_{j \in [k]} \frac{|\langle U_\star, U(0)e_j\rangle|}{\|U_\star^T U(0)\|_2} \approx 0.99 \max_{j \in [k]} \|U_{\mathrm{gd}} e_j\|_2, \tag{7}$$

which completes the proof sketch of Theorem 1.

## 4 Explicit regularization gives pruning-friendly models: population analysis

In this section, we study the properties of the squared error augmented with group Lasso regularization in the population (infinite sample) setting. We show that second-order stationary points of the

---

**Algorithm 1** Greedy pruning based on group-Lasso regularization

---

**Inputs**: Measurements $\{(A_i, y_i)$ where $y_i = \langle A_i, U_\star U_\star^T \rangle + \varepsilon_i\}_{i=1}^n$ (in the population setting $n = \infty$);
**Initialization**: Set parameters: $\lambda, \beta, \epsilon, \gamma$ and $m_{\text{fine-tune}}$.
**Greedy pruning phase:**

1: Find an $(\epsilon, \gamma)$-approximate SOSP of $f_{\text{emp}}$ (resp. $f_{\text{pop}}$), $U$, satisfying $\|U\|_{\text{op}} \leq 3$.
2: Let $S = \{i \in [k] : \|Ue_i\|_2 \leq 2\sqrt{\beta}\}$ denote the set of columns with small $\ell_2$ norm.
  Create a new matrix $U_{\text{prune}}$ which only preserves the columns of $U$ in $[k] \setminus S$, deleting the columns in $S$.
**Fine-tuning phase:**

1: Run $m_{\text{fine-tune}}$ iterations of gradient descent on $\mathcal{L}_{\text{pop}}(U)$ (resp. $\mathcal{L}_{\text{emp}}(U)$) initialized at $U_{\text{prune}}$ to get $U_{\text{out}}$.
2: **return** $U_{\text{out}}$.

---

regularized loss are suitable for greedy pruning, while at the same time achieving a small but non-zero generalization error. Note that the $\ell_2$ norm is a non-smooth function at the origin, and therefore, the overall regularized loss is non-smooth and non-convex. While there are several notions of approximate stationary points for non-differentiable and non-convex functions, for technical convenience, we replace $\mathcal{R}$ by a smooth proxy. In particular, for a smoothing parameter $\beta > 0$, define a smooth version of the $\ell_2$ norm, and the corresponding smooth regularizer $\mathcal{R}_\beta$ as,

$$\mathcal{R}_\beta(U) = \sum_{i=1}^{k} \ell_2^\beta(Ue_i), \quad \text{where} \quad \ell_2^\beta(v) = \frac{\|v\|_2^2}{\sqrt{\|v\|_2^2 + \beta}}. \tag{8}$$

Note that the smaller the value of $\beta$ is, the closer is $\ell_2^\beta(v)$ to $\|v\|_2$. Considering this definition, the overall regularized loss we study in the population setting is

$$f_{\text{pop}}(U) = \mathcal{L}_{\text{pop}}(U) + \lambda \mathcal{R}_\beta(U), \tag{9}$$

where $\mathcal{L}_{\text{pop}} = \|UU^T - U_\star U_\star^T\|_F^2$ as defined in eq. (2). The above optimization problem is nonconvex due to the structure of $\mathcal{L}_{\text{pop}}(U)$, and finding its global minimizer can be computationally prohibitive. Fortunately, for our theoretical results, we do not require achieving global optimality and we only require an approximate second-order stationary point of the loss in eq. (9), which is defined below.

**Definition 2.** *We say that $U$ is an $(\epsilon, \gamma)$-approximate second-order stationary point of $f$ if,*

1. *The gradient norm is bounded above by $\epsilon$, i.e., $\|\nabla f(U)\|_2 \leq \epsilon$.*

2. *The eigenvalues of the Hessian are larger than $-\gamma$, i.e., $\lambda_{\min}(\nabla^2 f(U)) \geq -\gamma$.*

The full algorithmic procedure that we analyze in this section is summarized in Algorithm 1. Once we find an $(\epsilon, \gamma)$-approximate second-order stationary point (SOSP) of eq. (9) for some proper choices of $\epsilon$ and $\gamma$, we apply greedy pruning on the obtained model $U$ and by eliminating all of its columns with $\ell_2$ norm below a specific threshold. As a result, the pruned model $U_{\text{prune}}$ has fewer columns than the trained model $U$. In fact, in our following theoretical results, we show that if the parameters are properly selected, $U_{\text{prune}}$ has exactly $r$ columns, which is the same as $U_\star$. Finally, we fine-tune the pruned solution by running gradient descent on the unregularized loss $\mathcal{L}_{\text{pop}}$ (resp. $\mathcal{L}_{\text{emp}}$). Next, we state the properties of the pruned model generated by Algorithm 1 in the infinite sample setting.

**Theorem 3.** *Consider the population loss with regularization in eq. (9), where $U_\star$ has rank $r$ and its smallest singular value is denoted by $\sigma_r^\star$. Moreover, consider $U_{prune}$ as the output of the pruning phase in Algorithm 1 with parameters $\beta, \lambda, \epsilon, \gamma$ satisfying the conditions*[2],

$$\beta = c_\beta \frac{(\sigma_r^\star)^2}{r}, \qquad \lambda = c_\lambda \frac{(\sigma_r^\star)^3}{\sqrt{kr}}, \qquad \gamma \leq c_\gamma \frac{(\sigma_r^\star)^3}{\sqrt{k}r^{5/2}}, \qquad \epsilon \leq c_\epsilon \frac{(\sigma_r^\star)^{7/2}}{\sqrt{k}r^{5/2}}, \tag{10}$$

*for some absolute constants $c_\beta, c_\lambda, c_\epsilon, c_\gamma > 0$. Then, we have*

1. *$U_{prune}$ has exactly $r$ columns.*

2. *$\|U_{prune}U_{prune}^T - U_\star U_\star^T\|_F \leq \frac{1}{2}(\sigma_r^\star)^2$.*

---

[2]This style of result exists for LASSO as well (see [44, Theorem 1]), where the optimal choice of the regularization parameter, $\lambda^\star$, depends on the true sparsity $r$, but a general guarantee can be established as well, which degrades as $\lambda$ deviates from $\lambda^\star$. In practice $\lambda$ is chosen using cross-validation. For simplicity of presentation, we state the result when $r$ and $\sigma_r^\star$ are known up to constants.

This result relies on showing that all bounded SOSPs of the regularized loss in eq. (9) are suitable for greedy pruning: removing the columns of $U$ below a certain $\ell_2$-norm threshold results in a solution $U_{\text{prune}}$ having exactly $r$ columns, while at the same time having a small generalization error. Hence, it can serve as a proper warm-start for the fine-tuning phase.

**Proof sketch of Theorem 3.** The key idea is to identify that if we have a matrix $U$ such that $UU^T = U_\star U_\star^T$, and the columns of $U$ are orthogonal to one another, then $U$ has exactly $r$ non-zero columns, where $r$ is the rank of $U_\star$. This statement can be shown to hold even when $UU^T \approx U_\star U_\star^T$ and the columns of $U$ are only approximately orthogonal. The main observation we prove is that a bounded $(\epsilon, \gamma)$-approximate SOSPs of eq. (9) denoted by $U$ satisfies the following condition:

$$\forall i, j : \|Ue_i\|_2, \|Ue_j\|_2 \geq 2\sqrt{\beta}, \quad \frac{\langle Ue_i, Ue_j \rangle}{\|Ue_i\|_2 \|Ue_j\|_2} \approx 0. \tag{11}$$

In other words, all the large columns of $U$ have their pairwise angle approximately $90°$. Thus, by pruning away the columns of $U$ that have an $\ell_2$ norm less than $2\sqrt{\beta}$, the remaining columns of $U$, i.e., the columns of $U_{\text{prune}}$, are now approximately at $90°$ angles to one another. If $\beta$ is chosen to be sufficiently small, after deleting the low-norm columns, the approximation $U_{\text{prune}}U_{\text{prune}}^T \approx UU^T$ holds. By the second order stationarity of $U$, we also have that $UU^T \approx U_\star U_\star^T$. Together, this implies that $U_{\text{prune}}U_{\text{prune}}^T \approx U_\star U_\star^T$ and $U_{\text{prune}}U_{\text{prune}}^T$ is close to a rank $r$ matrix. Since $U_{\text{prune}}$ has orthogonal columns, this also means that it has exactly $r$ columns. Finally, to establish a bound on the approximation error, we simply use the triangle inequality that $\|U_{\text{prune}}U_{\text{prune}}^T - U_\star U_\star^T\|_F \leq \|UU^T - U_\star U_\star^T\|_F + \|UU^T - U_{\text{prune}}U_{\text{prune}}^T\|_F$. The former is small by virtue of the fact that $U$ is an approximate second order stationary point of $f_{\text{pop}} \approx \mathcal{L}_{\text{pop}}$ when $\lambda$ is small; the latter term is small by the fact that only the small norm columns of $U$ were pruned away.

Now the only missing part that remains to justify is why the $\mathcal{R}_\beta$ regularizer promotes orthogonality in the columns of approximate second order stationary points and the expression in eq. (11) holds. This is best understood by looking at the regularized loss for the case $\beta = 0$, which is equivalent to $\|UU^T - U_\star U_\star^T\|_F^2 + \lambda \sum_{i=1}^k \|Ue_i\|_2$ and consider any candidate first-order stationary point $U$ of this objective. Let $Z \in \mathbb{R}^{d \times k}$ be a variable constrained to satisfy $ZZ^T = UU^T$. Stationarity of $U$ implies that the choice $Z = U$ must also be a first-order stationary point of the constrained optimization problem,

$$\text{Minimize: } \|ZZ^T - U_\star U_\star^T\|_F^2 + \lambda \sum_{i=1}^k \|Ze_i\|_2, \qquad \text{Subject to: } ZZ^T = UU^T. \tag{12}$$

The first term in the objective is a constant under the constraint and we may remove it altogether. When $U$ is a full-rank stationary point, constraint qualification holds, and it is possible to write down the necessary KKT first-order optimality conditions, which reduce to,

$$\forall i \in [k], \quad -\lambda \frac{Ze_i}{\|Ze_i\|_2} + (\Lambda + \Lambda^T)Ze_i = 0 \tag{13}$$

where $\Lambda \in \mathbb{R}^{d \times d}$ is the set of optimal dual variables. Since $Z = U$ is a first-order stationary point of the problem in eq. (12) and it satisfies eq. (13), the above condition means that the columns $Ue_i$ are the eigenvectors of the symmetric matrix $\Lambda + \Lambda^T$. If all the eigenvalues of $\Lambda + \Lambda^T$ were distinct, then this implies that the eigenvectors are orthogonal and $Ue_i \perp Ue_j$ for all $i \neq j$.

While this analysis conveys an intuitive picture, there are several challenges in extending this further. It is unclear how to establish that the eigenvalues of $\Lambda + \Lambda^T$ are distinct. Moreover, this analysis only applies for full-rank stationary points and does not say anything about rank deficient stationary points $U$, where constraint qualification does not hold. Furthermore, it is even more unclear how to extend this analysis to approximate stationary points. Our proof will circumvents each of these challenges by $(a)$ showing that at approximate SOSPs, even if the eigenvalues of $\Lambda + \Lambda^T$ are not distinct, the columns of $U$ are orthogonal, and $(b)$ directly bounding the gradient and Hessian of the regularized loss, rather than studying the KKT conditions to establish guarantees even for approximate stationary points which may be rank deficient. Having established guarantees for the pruning phase of the algorithm in the population setting, we next prove a result in the finite sample setting. ∎

# 5 Finite sample analysis

Next, we extend the results of the previous section to the finite sample setting. Here, we also focus on the smooth version of the regularizer and study the following problem

$$f_{\text{emp}}(U) = \mathcal{L}_{\text{emp}}(U) + \lambda \mathcal{R}_\beta(U), \tag{14}$$

where the empirical loss $\mathcal{L}_{\text{emp}}$ is defined in eq. (1) and the smooth version of the group Lasso regularizer $\mathcal{R}_\beta(U)$ is defined in eq. (8). In the finite sample setting, we assume that the measurement matrices satisfy the restricted isometry property (RIP) [45], defined below.

**Assumption 1.** *Assume that the measurement matrices $\{A_1, \cdots, A_n\}$ are $(2k, \delta)$-RIP. In other words, for any $d \times d$ matrix with rank $\leq 2k$,*

$$(1 - \delta)\|X\|_F^2 \leq \frac{1}{n}\sum_{i=1}^n \langle A_i, X \rangle^2 \leq (1 + \delta)\|X\|_F^2. \tag{15}$$

*This condition is satisfied, for instance, if the entries of the $A_i$'s were sampled $\mathcal{N}(0, 1/d)$ (i.e. Gaussian measurements), as long as $n \gtrsim dk/\delta^2$ [45].*

**Theorem 4.** *Consider the empirical loss with smooth regularization in eq. (14), where $U_\star$ has rank $r$ and unit spectral norm, with its smallest singular value denoted $\sigma_r^\star$, and noise variance $\sigma^2$. Consider $U_{prune}$ as the output of the pruning phase in Algorithm 1 with parameters $\beta, \lambda, \epsilon, \gamma$ chosen as per eq. (10). If Assumption 1 is satisfied with $\delta \leq c_\delta \frac{(\sigma_r^\star)^{3/2}}{\sqrt{k}r^{5/2}}$ and the number of samples is at least $n \geq C_4 \frac{\sigma^2}{(\sigma_r^\star)^4} dk^2 r^5 \log(d/\eta)$, where $C_4 > 0$ is a sufficiently large constant, then with probability at least $1 - \eta$,*

*1. $U_{prune}$ has exactly $r$ columns.*

*2. $U_{prune}$ satisfies the spectral initialization condition: $\|U_{prune}U_{prune}^T - U_\star U_\star^T\|_F \leq \frac{1}{2}(\sigma_r^\star)^2.$*

With Gaussian measurements, the overall sample requirement (including that from the RIP condition) in Theorem 4 is satisfied when $n \geq \widetilde{\Omega}\left(\frac{1+\sigma^2}{(\sigma_r^\star)^4} dk^2 r^5\right)$. The high level analysis of this result largely follows that of Theorem 3 in the population setting – we approximate the finite-sample gradient and Hessian by their population counterparts and show that the approximation error decays with the number of samples as $O(1/\sqrt{n})$.

## 5.1 Fine-tuning phase: Realizing the benefits of pruning

In the fine-tuning phase, the learner runs a few iterations of gradient descent on $\mathcal{L}_{\text{emp}}$ initialized at the pruned solution $U_{\text{prune}}$. Since the model is no longer overparameterized after pruning (by Theorem 4), there are several works analyzing the generalization performance and iteration complexity of gradient descent. Here we borrow the local convergence result of [22] which requires the initial condition that $\|U_0 U_0^T - U_\star U_\star^T\|_F \leq c(\sigma_r^\star)^2$, where $c$ is any constant less than 1. As shown in part (b) of Theorem 4, this initial condition is satisfied by $U_{\text{prune}}$.

**Theorem 5.** *[22, Corollary 2] Suppose $\|U_\star\|_{op} = 1$. If we use the output of the greedy pruning in Algorithm 1 denoted by $U_{prune} \in \mathbb{R}^{d \times r}$ as the initial iterate for the fine-tuning phase, then after $t \geq m_{fine\text{-}tune} = C_5 (\sigma_1^\star/\sigma_r^\star)^{10} \cdot \log(\sigma_r^\star/\sigma_1^\star \cdot n/d)$ iterations, for some sufficiently large absolute $C_5 > 0$, the iterates $\{U_t\}_{t \geq 1}$ of factored gradient descent on $\mathcal{L}_{emp}(\cdot)$ satisfy,*

$$\|U_t U_t^T - U_\star U_\star^T\|_F \lesssim \frac{\sigma}{(\sigma_r^\star)^2}\sqrt{\frac{rd}{n}}. \tag{16}$$

Theorem 5 shows that in the fine-tuning phase, the iterates of gradient descent converges at a linear rate to the generalization error floor of $\sigma/(\sigma_r^\star)^2 \cdot \sqrt{rd/n}$, which is also known to be statistically (minimax) optimal [46, 47]. This is possible because the learner is able to correctly identify the rank of the model in the pruning phase and operate in the exactly specified setting in the fine-tuning phase. In contrast, one may ask how this guarantee compares with running vanilla factored gradient descent in the overparameterized setting. This corresponds to the case where no pruning is carried out to reduce

the size of the model. [23] presented a guarantee for the convergence of factored gradient descent from a warm start in the overparameterized setting. In comparison with the exactly specified case, the generalization error floor $\lim_{t \to \infty} \|U_t U_t^T - U_\star U_\star^T\|_F$ is shown to scale as $O(\sigma/(\sigma_r^\star)^2 \cdot \sqrt{kd/n})$[3] which now depends on $k$. Furthermore, linear convergence can no longer be established for vanilla gradient descent because of the ill-conditioning of the objective. This is not just an artifact of the proof - experimentally too, the convergence slowdown was noticed in [23, Figure 1].

This discussion shows that greedy pruning the model *first*, prior to running gradient descent, in fact generates solutions which generalize better and also converge much faster.

**Remark 6.** *Based on Theorem 4 and Theorem 5 (eq. (16)), under Gaussian measurements, given*

$$n \geq n_{\varepsilon,\eta} = O\left(\frac{\sigma^2}{(\sigma_r^\star)^4}\frac{rd}{\varepsilon^2} + \frac{1+\sigma^2}{(\sigma_r^\star)^4}dk^2r^5\log(d/\eta)\right) \tag{17}$$

*samples, with probability $\geq 1 - \eta$, Algorithm 1 produces $U_{out}$ that has exactly $r$ columns and satisfies $\|U_{out}U_{out}^T - U_\star U_\star^T\|_F \leq \varepsilon$. Note that the sample complexity depends on the amount of overparameterization, $k$, only in the lower order (independent of $\varepsilon$) term.*

Note that this result uses the fact that under Gaussian measurements, the RIP condition required in Theorem 4 is satisfied if $n \geq n_\varepsilon$.

## 6 Implementing Algorithm 1: Smoothness and optimization oracles

In this section we instantiate the optimization oracle in Algorithm 1, which outputs an approximate SOSP with bounded operator norm. First, we establish that the loss $f_{\text{emp}}$ is well behaved on the domain $\{U : \|U\|_{\text{op}} \leq 3\}$, in that its gradient and Hessian are Lipschitz continuous. These conditions are required by many generic optimization algorithms which return approximate second order stationary points [48, 49]. We establish these properties for the population loss for simplicity and leave extensions to the empirical loss for future work.

**Theorem 7.** *Consider the population loss $f_{pop}$ in eq. (9). Assume $\lambda \leq \min\{\beta, \sqrt{\beta}\}$ and $\|U_\star\|_{op} = 1$. The objective $f_{pop}(\cdot)$ defined in eq. (9) satisfies for any $U, V \in \mathbb{R}^{d \times k}$ such that $\|U\|_{op}, \|V\|_{op} \leq 3$,*

*(a) Lipschitz gradients: $\|\nabla f_{pop}(U) - \nabla f_{pop}(V)\|_F \lesssim \|U - V\|_F$,*

*(b) Lipschitz Hessians: $\|\nabla^2 f_{pop}(U) - \nabla^2 f_{pop}(V)\|_{op} \lesssim \|U - V\|_F$.*

Under the Lipschitz gradients and Lipschitz Hessian condition, a large number of algorithms in the literature show convergence to an approximate SOSP. A common approach for finding such a point is using the noisy gradient descent algorithm [50]. However, note that we establish these Lipschitzness properties on the bounded domain $\{U : \|U\|_{\text{op}} \leq 3\}$, and it remains to verify whether these algorithms indeed approach such points. A similar concern is present with Algorithm 1, which requires access to an optimization oracle which finds an $(\epsilon, \delta)$-approximate SOSP of $f_{\text{emp}}$ which are also bounded, in that $\|U\|_{\text{op}} \leq 3$. The final step is to identify conditions under which these algorithms indeed output stationary points which are bounded, satisfying $\|U\|_{\text{op}} \leq 3$. We establish this behavior for a wide family of perturbed gradient based methods.

**Perturbed gradient descent:** We consider gradient descent with the following update rule: starting from the initialization $U_0$, for all $t \geq 0$,

$$U_{t+1} \leftarrow U_t - \alpha(\nabla f_{\text{pop}}(U_t) + P_t) \tag{18}$$

where $P_t$ is a perturbation term, which for example, could be the explicit noise ($P_t \sim \text{Unif}(\mathbb{B}(r))$ for appropriate $r$) added to escape strict saddle points in [50]. Over the course of running the update rule eq. (18), we show that $\|U_t\|_{\text{op}}$ remains bounded under mild conditions if the algorithm is initialized within this ball. In combination with Theorem 7, this shows that the noisy gradient descent approach of [50] can be used to find the SOSPs required for Algorithm 1.

**Theorem 8.** *Consider optimization of the regularized population loss using the update rule in eq. (18). Assume that $\|U_\star\|_{op} = 1$ and suppose the parameters are selected as $\alpha \leq 1/8$ and $\lambda \leq \sqrt{\beta}$, and we have $\|P_t\|_{op} \leq 1$ almost surely for each $t \geq 0$. Then, assuming that the condition $\|U_0\|_{op} \leq 3$ is satisfied at initialization, for every $t \geq 1$, we have $\|U_t\|_{op} \leq 3$.*

---

[3]The result is often stated in terms of the smallest non-zero eigenvalue of $X_\star = U_\star U_\star^T$, which equals $(\sigma_r^\star)^2$.

Thus, when perturbed gradient descent is carried out to optimize the regularized loss $f_{\text{pop}}$ eq. (9), the algorithm always returns bounded iterates. In conjunction with Theorem 7, this implies, for example, that noisy gradient descent [50] converges to an approximate second-order stationary point of the regularized population objective. Moreover, the authors of the same paper show the number of gradient calls made by noisy gradient descent to find an $(\epsilon, \gamma)$-SOSP is upper bounded by $\widetilde{O}\left(1/\min\{\epsilon^2, \gamma^2\}\right)$ for a non-convex objective with $O(1)$-smooth gradients and Hessians. Combining with the choices of $\epsilon$ and $\gamma$ in eq. (10) results in the following theorem.

**Theorem 9.** *Assume Gaussian measurements and rescale so that* $\|U_\star\|_{op} = 1$. *Given* $n \geq n_{\varepsilon, \eta}$ *samples (eq.* (17)*), consider the algorithm which uses the output of greedy pruning (Algorithm 1),* $U_{prune} \in \mathbb{R}^{d \times r}$, *as the initial iterate for the fine-tuning phase, and runs noisy gradient descent [50] on* $\mathcal{L}_{emp}$ *in the fine-tuning phase. Overall, the algorithm makes,*

$$T_\epsilon = \widetilde{O}\left( \frac{1}{(\sigma_r^\star)^{10}} \log\left(\frac{n}{d\sigma_r^\star}\right) + \frac{kr^5}{(\sigma_r^\star)^7} \right) \tag{19}$$

*gradient calls and with probability* $\geq 1 - \eta$, *returns a* $U_{out}$ *satisfying* $\|U_{out}U_{out}^T - U_\star U_\star^T\|_F \leq \varepsilon$.

## 7 Conclusion

In this paper, we studied the efficacy of the greedy pruning + fine-tuning pipeline in learning low-complexity solutions for the matrix sensing problem, as well as for learning shallow neural networks with quadratic activation functions. We showed that training on the mean squared error augmented by a natural group Lasso regularizer results in models which are suitable for greedy pruning. Given sufficiently many samples, after pruning away the columns below a certain $\ell_2$-norm threshold, we arrived at a solution with the correct column sparsity of $r$. Running a few iterations of gradient descent to fine-tune the resulting model, the population loss was shown to converge at a linear rate to an error floor of $O(\sqrt{rd/n})$, which is also statistically optimal. We also presented a negative result showing the importance of regularization while training the model. To the best of our knowledge, our results provide the first theoretical guarantee on the generalization error of the model obtained via the greedy pruning + fine-tuning framework.

## Acknowledgements

The research of A. Mokhtari is supported in part by NSF Grants 2007668 and 2127697, ARO Grant W911NF2110226, the National AI Institute for Foundations of Machine Learning (IFML), the Machine Learning Lab (MLL), and the Wireless Networking and Communications Group (WNCG) industrial affiliates program at UT Austin. Kannan Ramchandran would like to acknowledge support from NSF CIF-2002821 and ARO fund 051242-00.

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
