# Supplementary material

## A   Failure of implicit regularization in pruning: Proof of Theorem 1

In this section, we prove the negative result showing that running gradient descent trained with implicit regularization results in dense networks that are not compatible with greedy pruning.

For scale reasons, and to simplify the exposition, we will assume that $\|U_\star\|_2 = 1$ through the remainder of this proof. The gradient flow update rule can be written as,

$$\frac{dU}{dt} = -\nabla\mathcal{L}(U) = -(UU^T - U_\star U_\star^T)U \qquad (20)$$

where $U = U(t)$ is the weight matrix at time $t$. Define the column vector $r(t) = U^T U_\star \in \mathbb{R}^k$ capturing the correlation/alignment between the weights on the neurons of $U_t$ with the ground truth $U_\star$, the signal component. Wherever convenient, we will drop the time argument in $r(t)$ and simply refer to it as $r$.

We will show that given that gradient flow is run starting from a small initialization, at convergence, the column norms $\|Ue_i\|_2$'s approximately distribute themselves proportional to the alignment of the

corresponding column of $U$ with $U_\star$ initially at $t = 0$. Namely,

$$\lim_{t \to \infty} \|U(t)e_i\|_2 \propto |\langle r(0), e_i \rangle| \tag{21}$$

Given a random initialization where the entries of $U(0)$ are i.i.d., $|\langle r(0), e_i \rangle|$ for each $i$ are independent. Moreover, when $U(0)$ follows a Gaussian distribution, we expect no single coordinate $|\langle r(0), e_i \rangle| = |\langle (U(0))^T U_\star, e_i \rangle|$ to be much larger than the others, a statement which we formally prove in Lemma 21. Combining this with eq. (21) results in a proof of Theorem 1.

The rest of this section is devoted to proving a formal version of eq. (21), which we state below.

**Lemma 10.** *Suppose the initialization scale parameter $\alpha \leq c\eta^2/k^3 d \log(kd)$ for a sufficiently small absolute constant $c > 0$. With probability $\geq 1 - O(\frac{1}{k})$ over the initialization, for each column $i \in [k]$,*

$$(1 - 5\eta)\frac{|\langle r(0), e_i \rangle|}{\|r(0)\|_2} \leq \lim_{t \to \infty} \|U(t)e_i\|_2 \leq \frac{|\langle r(0), e_i \rangle|}{\|r(0)\|_2}(1 + 4\sqrt{\eta}) \tag{22}$$

As a counterpart to the signal component $r(t)$, the noise component is $E(t) = (I - U_\star U_\star^T)U$. At a high level, the proof of Lemma 10 will be to establish that with a small initialization, the noise component satisfies $E \approx 0$ approximately, while the signal component $\|r(t)\|_2$ grows exponentially fast to 1 with $t$, and $\frac{d\langle r(t), e_i \rangle}{dt} \approx \langle r(t), e_i \rangle$ until $\|r(t)\|_2$ gets sufficiently close to 1. This will imply that $\langle r(t), e_i \rangle \approx \langle r(0), e_i \rangle e^t$ until then, and by extension,

$$\frac{\langle r(t), e_i \rangle}{\langle r(t), e_j \rangle} \approx \frac{\langle r(0), e_i \rangle}{\langle r(0), e_j \rangle}. \tag{23}$$

Since the noise component $E \approx 0$, $\|Ue_i\|_2 \approx |\langle r(t), e_i \rangle|$, plugging which into eq. (23) results in the ratio form of Lemma 10. By the gradient flow equation eq. (20), it is a short calculation to see that $r$ and $E$ evolve per the following differential equations,

$$\frac{dr}{dt} = r(1 - \|r\|_2^2) - E^T E r. \tag{24} \qquad\qquad \frac{dE}{dt} = -EU^T U. \tag{25}$$

In the sequel, we will show that as a function of $t$, $E^T E r$ decays linearly. The resulting differential equation $\frac{dr}{dt} \approx r(1 - \|r\|_2^2)$ shows linear convergence of $r$ until the point when $\|r\|_2$ approaches 1. We place these intuitions formally by discussing a formal proof of Lemma 10.

### A.1  Proof outline of Lemma 10

Firstly, as a function of the scale parameter $\alpha$, we bound the energy of the signal and noise components at initialization in Lemma 11. In Lemma 12 we first establish that $\|E(t)\|_2$ does not increase with time. In Lemma 14, we establish an upper bound on the signal norm $\|r(t)\|_2$ as a function of time. This shows that the signal energy cannot grow to be 1 too rapidly, which is necessary to show that the error term $E^T E r$ has sufficient time to decay to 0. In Lemma 15 we show that the signal norm $\|r(t)\|_2^2$ does not fall below a threshold of $3/4$ after $t$ grows to be sufficiently large. This is essential in proving Lemma 16 which shows that the norm of the error term $\|E^T E r\|_2$ in eq. (24) begins decaying after $\|r(t)\|_2^2$ becomes larger than $1/2$. Finally, in Lemmas 17 and 18 we use these results to prove a refined bound on the rate at which $\|r(t)\|_2 \to 1$. These results are collectively used in Lemmas 19 and 20 to prove the upper and lower bounds on $|\langle r(t), e_i \rangle| \approx \|Ue_i\|_2$.

### A.2  Understanding gradient flow: Proof of Lemma 10

In this section we prove Lemma 10 formally. First we establish bounds on the scale of parameters at initialization.

**Lemma 11.** *With probability* $\geq 1 - O(\frac{1}{k})$, *at initialization,*

$$\|E(0)\|_F^2, \|r(0)\|_2^2 \leq 2\alpha^2 kd \tag{26}$$

$$\|r(0)\|_2^2 \geq \cdot \left[\frac{k\alpha^2}{10}, 10k\alpha^2\right] \tag{27}$$

$$\forall i \in [k], |\langle r(0), e_i\rangle| \geq \frac{\alpha}{k^2} \tag{28}$$

*Proof.* Recall that $\langle r, e_i\rangle$ for each $i \in [k]$ is distributed $\sim \mathcal{N}(0, \alpha^2)$. By Gaussian anti-concentration [51] and union bounding, $\mathbb{P}\left(\min_{i\in[k]} |\langle r(0), e_i\rangle| \geq \frac{\alpha}{k^2}\right) \gtrsim \frac{1}{k}$ simultaneously for all $i \in [k]$. On the other hand, since every entry of $U(0)$ is iid distributed $\sim \mathcal{N}(0, \alpha^2)$, by tail bounds for $\chi^2$-distributed random variables [52, Lemma 1], $\|E(0)\|_F^2 \leq \|U(0)\|_F^2 \leq 2\alpha^2 kd$ with probability $\geq 1 - \exp(-kd) \geq 1 - O(\frac{1}{k})$. The same bound applies for $\|r(0)\|_2^2$ since it is also upper bounded by $\|U(0)\|_F^2$. Finally, the upper and lower bounds on $\|r(0)\|_2$ follows by noting that $\|r(0)\|_2^2 = \sum_{i=1}^k \langle U_\star, Ue_i\rangle^2$, which concentrates around $k\alpha^2$. The result directly follows by concentration of $\chi^2$-random variables. $\square$

As a corollary of Lemma 11 when the intialization parameter satisfies the upper bound in Lemma 10, $\|E(0)\|_F$ and $\|r(0)\|_2$ are upper bounded by small constants. We will use this fact several times in proving Lemma 10. Next we establish that the error $\|E\|_F$ does not grow with time.

**Lemma 12.** *At any time* $t \geq 0$, $\|E(t)\|_F \leq \|E(0)\|_F$.

*Proof.* The proof follows by showing that the time derivative of $\|E(t)\|_F^2$ is non-positive. Indeed,

$$\frac{d\|E\|_F^2}{dt} = 2\left\langle E, \frac{dE}{dt}\right\rangle = -2\mathsf{Tr}(E^T E U^T U) = -2\|UE^T\|_F^2 \leq 0. \tag{29}$$

$\square$

Starting from a small initialization, this means that the error matrix $E$ remains small over the course of gradient flow. In for any coordinate $i \in [k]$, by eq. (24), the differential equation governing $\langle r, e_i\rangle = \langle r(t), e_i\rangle$ is,

$$\frac{\langle r, e_i\rangle}{dt} = \langle r, e_i\rangle(1 - \|r\|_2^2) - e_i^T E^T E r. \tag{30}$$

By Lemma 12 we expect the error term $E^T E r$ to be small (not necessarily decaying) in comparison with the first term as long as $\|r(t)\|_2$ is smaller than an absolute constant. In this regime, the differential form is easy to control since we expect $\langle r, e_i\rangle$ to grow linearly and across the different coordinates $i \in [k]$, we expect its value to be proportional to that at initialization, $\langle r(0), e_i\rangle$.

However, when $\|r\|_2$ eventually approaches 1, the relative contribution of the signal term $\langle r, e_i\rangle(1 - \|r\|_2^2)$ and the error term $E^T E r$ become important. This is the main technical challenge in proving Lemma 10. We will show that even as $t \to \infty$, the error term cannot change $\langle r, e_i\rangle$ by more than a constant factor. For a sufficiently large constant $C > 0$, define,

$$T_0 \triangleq -\frac{1}{2}\frac{\log(\|r(0)\|_2^2) - 1}{1 - \|E(0)\|_F^2}. \tag{31}$$

As we will later show, $T_0$ controls the amount of time it takes the signal norm $\|r(t)\|_2$ to grow to an absolute constant. Firstly, we show that when the initialization scale $\alpha$ is sufficiently small, $T_0$ is approximately $-\log \|r(0)\|_2$.

**Lemma 13.** *When the initialization scale* $\alpha \leq c/k^3 d \log(kd)$ *for a sufficiently small absolute constant* $c > 0$,

$$-\log(\|r(0)\|_2) + \frac{1}{2} \leq T_0 \leq -\log(\|r(0)\|_2) + \frac{3}{2}. \tag{32}$$

*Proof.* This follows from the fact that,

$$T_0 = -\log(\|r(0)\|_2) - \frac{\|E(0)\|_F^2}{1 - \|E(0)\|_F^2} \log(\|r(0)\|_2) + \frac{1}{2(1 - \|E(0)\|_F^2)} \tag{33}$$

The lower bound on $T_0$ follows readily, noting that $\|r(0)\|_2, \|E(0)\|_F < 1$ from the initialization bounds in Lemma 11 when $\alpha \leq c/k^3 d \log(kd)$. For the upper bound on the other hand, note that the last term of eq. (33) is upper bounded by,

$$\frac{1}{2(1 - \|E(0)\|_F^2)} \leq 1 \tag{34}$$

And the middle term of eq. (33) is upper bounded by,

$$-\frac{\|E(0)\|_F^2}{1 - \|E(0)\|_F^2} \log(\|r(0)\|_2) \leq \frac{\|E(0)\|_F^2}{1 - \|E(0)\|_F^2} \frac{1}{\|r(0)\|_2} \overset{(i)}{\leq} \frac{2\alpha^2 kd}{(1 - 2\alpha^2 kd)} \frac{10}{\sqrt{k}\alpha} \leq \frac{1}{2}, \tag{35}$$

where the last inequality assumes that $\alpha \leq c_{35}/\sqrt{kd}$ for a small constant $c_{35} > 0$, and inequality $(i)$ uses the bounds on $\|E(0)\|_F$ and $\|r(0)\|_2$ proved in Lemma 11. Combining eqs. (34) and (35) with eq. (33) results in the proof of Lemma 13. $\qquad\square$

The next result we establish is an upper bound on the signal norm $\|r(t)\|_2$. The proof follows by upper bounding the rate of change of $\|r(t)\|_2$ and integrating the resulting bound.

**Lemma 14.** *At any time $t \geq 0$, the signal norm is upper bounded by,*

$$\|r(t)\|_2^2 \leq \frac{e^{2(t-T_0)+1+\eta}}{1 + e^{2(t-T_0)+1+\eta}} \leq 1. \tag{36}$$

*Proof.* From the differential equation governing $r$ we can infer that,

$$\frac{d\|r\|_2^2}{dt} = 2\|r\|_2^2(1 - \|r\|_2^2) - 2\|Er\|_2^2 \leq 2\|r\|_2^2(1 - \|r\|_2^2) \tag{37}$$

By a theorem of [53] on differential inequalities, the trajectory of $\|r(t)\|_2^2$ as a function of $t$ starting from some reference point $\|r(0)\|_2^2$ is pointwise lower bounded by the trajectory of $\|r(t)\|_2^2$ obtained when the inequality is set to be an equality. In particular, by integrating the differential equation this results in the lower bound,

$$\log \frac{\|r(t)\|_2^2}{1 - \|r(t)\|_2^2} - \log \frac{\|r(0)\|_2^2}{1 - \|r(0)\|_2^2} \leq 2t \tag{38}$$

Observe that $-\log \frac{\|r(0)\|_2^2}{1-\|r(0)\|_2^2} \geq -\log \|r(0)\|_2^2 - \eta \geq 2T_0 - 1 - \eta$ where the first inequality uses the upper bound on $\|r(0)\|_2^2 \leq \eta$ when $\alpha \leq c/k^3 d \log(kd)$ and the second by Lemma 13. Therefore by rearranging the terms we have,

$$\|r(t)\|_2^2 \leq \frac{e^{2t-2T_0+1+\eta}}{1 + e^{2t-2T_0+1+\eta}}. \tag{39}$$

$\qquad\square$

As it turns out, the error term $E^T E r(t)$ in eq. (24) can be shown to decrease linearly once $\|r(t)\|_2^2$ hits the critical threshold of $1/2$. In the next lemma we show that the signal norm $\|r(t)\|_2^2$ never drops below the threshold of $3/4$ for all $t \geq T_0 + 2$. Thus, after a sufficiently large amount of time, we expect the differential equation for $r$ to behave as $\frac{dr}{dt} \approx r(1 - \|r\|_2^2)$.

**Lemma 15.** *For $t \geq T_0$,*

$$\|r(t)\|_2^2 \geq \frac{1 - \|E(0)\|_F^2}{1 + e^{-2\left(1 - \|E(0)\|_F^2\right)(t-T_0)+1}} \tag{40}$$

*As an implication, for any $t \geq T_0 + 2$, under the small initialization $\alpha \leq c/k^3 d \log(kd)$, $\|r(t)\|_2^2 \geq 3/4$.*

*Proof.* From the differential equation governing $r$, we may infer that,

$$\frac{d\|r\|_2^2}{dt} = 2\left\langle r, \frac{dr}{dt} \right\rangle = 2\|r\|_2^2(1 - \|r\|_2^2) - 2\|Er\|_2^2. \tag{41}$$

Note that $\|Er\|_2 \leq \|E\|_F \|r\|_2 \leq \|E(0)\|_F \|r\|_2$ and therefore,

$$2\|r\|_2^2(1 - \|E(0)\|_F^2 - \|r\|_2^2) \leq \frac{d\|r\|_2^2}{dt} \tag{42}$$

Rearranging and integrating both sides,

$$2(1 - \|E(0)\|_F^2)t \leq \log\left(\frac{\|r(t)\|_2^2}{1 - \|E(0)\|_F^2 - \|r(t)\|_2^2}\right) - \log\left(\frac{\|r(0)\|_2^2}{1 - \|E(0)\|_F^2 - \|r(0)\|_2^2}\right) \tag{43}$$

$$\leq \log\left(\frac{\|r(t)\|_2^2}{1 - \|E(0)\|_F^2 - \|r(t)\|_2^2}\right) + 2(1 - \|E(0)\|_F^2)T_0 + 1 \tag{44}$$

where the last inequality uses the choice of the initialization scale $\alpha \leq c\eta^2/k^3 d \log(kd)$, and the bounds on $\|E(0)\|_F^2, \|r(0)\|_2^2$ in Lemma 11. Therefore,

$$(1 - \|E(0)\|_F^2 - \|r(t)\|_2^2)e^{2\left(1 - \|E(0)\|_F^2\right)(t - T_0) - 1} \leq \|r(t)\|_2^2 \tag{45}$$

$$\implies \frac{1 - \|E(0)\|_F^2}{1 + e^{-2\left(1 - \|E(0)\|_F^2\right)(t - T_0) + 1}} \leq \|r(t)\|_2^2 \tag{46}$$

$\square$

Having established that $\|r(t)\|_2^2$ does not decay below the threshold of $3/4$ beyond time $T_0 + 2$, we establish that this condition is sufficient for the error term $E^T Er$ to begin decaying to $0$. Below we instead bound $\|Er\|_F$, and a bound on $\|E^T Er\|_F$ is obtained by upper bounding it as $\|E(t)\|_F \|Er(t)\|_F \leq \|E(0)\|_F \|Er\|_F$ by Lemma 12.

**Lemma 16.** *At any time $t \geq 0$, the error term,*

$$\|Er(t)\|_2 \leq \frac{\|Er(0)\|_2 e^t}{1 + e^{2\left(1 - \|E(0)\|_F^2\right)(t - T_0) - 1}} \tag{47}$$

*Proof.* By explicit computation,

$$\frac{dEr}{dt} = -EU^T Ur + Er(1 - \|r\|_2^2) - EE^T Er \tag{48}$$

$$= -E(E^T E + rr^T)r + Er(1 - \|r\|_2^2) - EE^T Er \tag{49}$$

$$= -2E(E^T E)r + Er(1 - 2\|r\|_2^2) \tag{50}$$

By taking an inner product with $Er$, we get that,

$$\frac{d\|Er\|_2^2}{dt} = -4\|(E^T E)r\|_2^2 + 2\|Er\|_2^2(1 - 2\|r\|_2^2). \tag{51}$$

Using the lower bound on $\|r\|_2^2$ in Lemma 15,

$$\frac{d\|Er\|_2^2}{dt} \leq 2\|Er\|_2^2\left(1 - \frac{2(1 - \|E(0)\|_F^2)}{1 + e^{-2\left(1 - \|E(0)\|_F^2\right)(t - T_0) + 1}}\right) \tag{52}$$

Rearranging and integrating both sides from time $0$ to $t$ using the fact that $\int \frac{dx}{1 + e^{-x}} = \log(1 + e^x)$,

$$\log\|Er(t)\|_2^2 - \log\|Er(0)\|_2^2 \leq 2\left(t - \log\left(1 + e^{2\left(1 - \|E(0)\|_F^2\right)(t - T_0) - 1}\right)\right) \tag{53}$$

$$\implies \|Er(t)\|_2 \leq \frac{\|Er(0)\|_2 e^t}{1 + e^{2\left(1 - \|E(0)\|_F^2\right)(t - T_0) - 1}}, \tag{54}$$

where the last inequality follows by exponentiating both sides and rearranging. $\square$

Using the fact that $\|Er\|_F$ rapidly decays to $0$ after time $T_0$, we can in fact establish a more refined lower bound on the rate at which $\|r(t)\|_2$ approaches $1$. The decay of the error term is not captured in the prior lower bound on $\|r(t)\|_2$ in Lemma 15. The error decay established in Lemma 16 is essential to proving such a result, especially as $\|r(t)\|_2$ approaches $1$. In this regime, the error term $\|Er\|_2^2$ in eq. (41) becomes comparable with the leading order term $\|r\|_2^2(1 - \|r\|_2^2)$.

**Lemma 17.** *At any time $t \leq 3T_0/2$, for some absolute constant $C_{55} > 0$,*

$$\|r(t)\|_2^2 \geq \frac{e^{2t}}{\|r(0)\|_2^{-2} + e^{2t} - 1} - C_{55}\frac{\|E(0)\|_F^2 T_0}{\|r(0)\|_2}. \tag{55}$$

*Proof.* From Lemma 16, note that,

$$\|Er(t)\|_2^2 \leq \frac{\|E(0)\|_F^2\|r(0)\|_2^2 e^{2t}}{\left(1 + e^{2\left(1-\|E(0)\|_F^2\right)(t-T_0)-1}\right)^2} \tag{56}$$

$$\lesssim \frac{\|Er(0)\|_F^2 e^{2t}}{\left(1 + e^{2(t-T_0)}\right)^2}, \tag{57}$$

where the last inequality uses the fact that when $t \leq 4T_0$, $(1 - \|E(0)\|_F^2)(t - T_0) \geq t - T_0 - 4$ (see the analysis in eqs. (33) and (35)). This inequality also uses Lemma 13 to bound $\|r(0)\|_2^2$. From Lemma 16, therefore, denoting $x = \|r\|_2^2$, for some absolute constant $C_{57} > 0$,

$$\frac{dx}{dt} = 2x(1-x) - C_{57}\frac{e^{2t}\|Er(0)\|_F^2}{\left(1 + e^{2(t-T_0)}\right)^2} \tag{58}$$

$$\implies e^{-2t}\frac{dx}{dt} = 2e^{-2t}x - 2e^{-2t}x^2 - C_{57}\frac{\|Er(0)\|_F^2}{\left(1 + e^{2(t-T_0)}\right)^2} \tag{59}$$

$$\implies \frac{d}{dt}\left\{e^{-2t}x\right\} = -2e^{-2t}x^2 - C_{57}\frac{\|Er(0)\|_F^2}{\left(1 + e^{2(t-T_0)}\right)^2} \tag{60}$$

$$\implies \frac{dy}{dt} = -2y^2 e^{2t} - C_{57}\frac{\|Er(0)\|_F^2}{\left(1 + e^{2(t-T_0)}\right)^2}, \tag{61}$$

where $y = e^{-2t}x$. Define a new variable $\tilde{y}(t)$ with $\tilde{y}(0) = y(0)$ and satisfying the following differential equation corresponding to just the first term on the RHS of eq. (61),

$$\frac{1}{\tilde{y}^2}\frac{d\tilde{y}}{dt} = -2e^{2t} \iff \tilde{y}(t) = \frac{1}{1/y(0) + e^{2t} - 1} \tag{62}$$

On the other hand, since $\frac{dy}{dt} \leq -2y^2 e^{2t}$, rearranging and integrating both sides, we get $y(t) \leq 1/(y(0) + e^{2t} - 1) = \tilde{y}(t)$. Plugging this back into eq. (61),

$$\frac{dy}{dt} \geq -2(\tilde{y})^2 e^{2t} - C_{57}\frac{\|Er(0)\|_F^2}{\left(1 + e^{2(t-T_0)}\right)^2}. \tag{63}$$

However, by definition, $\frac{d\tilde{y}}{dt} = -2(\tilde{y})^2 e^{2t}$ and therefore, integrating both sides, for some absolute constant $C'_{57} > 0$, we get,

$$y(t) - y(0) \geq \tilde{y}(t) - \tilde{y}(0) - C'_{57}\|Er(0)\|_F^2 T_0 \tag{64}$$

Plugging in $\tilde{y}(t)$ and subsequently $y(t)$ and $x(t)$ results in the equations,

$$\|r(t)\|_2^2 \geq \frac{e^{2t}}{\|r(0)\|_2^{-2} + e^{2t} - 1} - C'_{57}\|Er(0)\|_F^2 T_0 e^{2t} \tag{65}$$

When $t \leq 3T_0/2$, by using the upper bound on $T_0$ from Lemma 13, the second term on the RHS itself is upper bounded by $C''_{57}\|E(0)\|_F^2 T_0/\|r(0)\|_2$ for another absolute constant $C''_{57} > 0$. This completes the proof. $\qquad\square$

While the refined convergence lemma in Lemma 17 applies for the case when $t \leq 3T_0/2$, we also prove a lemma for the case when $t \geq 3T_0/2$.

**Lemma 18.** *At any $t \geq 3T_0/2$,*

$$e^{3t/4}(1 - \|r(t)\|_2^2) \lesssim 1 + e^{11T_0/4}\|Er(0)\|_2^2 + e^{3T_0/4}. \tag{66}$$

*Proof.* Consider any time $t \geq 3T_0/2$ which is greater than $T_0 + 2$ by the small initialization and Lemma 11. By Lemma 15, $\|r\|_2^2 \geq 3/4$. From eq. (41) and Lemma 16,

$$\forall t \geq T_0, \ \frac{d\|r\|_2^2}{dt} \geq \frac{3}{4}(1 - \|r\|_2^2) - 2\frac{\|Er(0)\|_2^2 e^{2t}}{\left(1 + e^{2\left(1 - \|E(0)\|_F^2\right)(t-T_0) - 1}\right)^2} \tag{67}$$

Multiplying both sides by $e^{3t/4}$ and rearranging,

$$e^{3t/4}\frac{d(1 - \|r\|_2^2)}{dt} + \frac{3}{4}e^{3t/4}(1 - \|r\|_2^2) \leq 2\frac{\|Er(0)\|_2^2 e^{2t} e^{3t/4}}{\left(1 + e^{2\left(1 - \|E(0)\|_F^2\right)(t-T_0) - 1}\right)^2} \tag{68}$$

$$\implies \frac{d}{dt}\left(e^{3t/4}(1 - \|r\|_2^2)\right) \leq 2\int \frac{\|Er(0)\|_2^2 e^{2t} e^{3t/4}}{\left(1 + e^{2\left(1 - \|E(0)\|_F^2\right)(t-T_0) - 1}\right)^2} dt. \tag{69}$$

Integrating from $T_0$ to $t$, we can upper bound eq. (69) as,

$$e^{3t/4}(1 - \|r(t)\|_2^2) - e^{3T_0/4}(1 - \|r(T_0)\|_2^2) \lesssim \int_{T_0}^t \frac{\|Er(0)\|_2^2 e^{2t} e^{3t/4}}{e^{4\left(1 - \|E(0)\|_F^2\right)(t-T_0)}} dt, \tag{70}$$

By the small initialization, $\|E(0)\|_F^2 \leq 1/8$ and the denominator in the integral is lower bounded by $e^{7/2(t-T_0)}$. Therefore,

$$e^{3t/4}(1 - \|r(t)\|_2^2) - e^{3T_0/4}(1 - \|r(T_0)\|_2^2) \lesssim e^{7T_0/2}\int_{T_0}^t \|Er(0)\|_2^2 e^{-3t/4} dt \tag{71}$$

$$\lesssim e^{7T_0/2}\|Er(0)\|_2^2 e^{-3T_0/4} \tag{72}$$

$$= 2e^{11T_0/4}\|Er(0)\|_2^2 \tag{73}$$

Therefore,

$$e^{3t/4}(1 - \|r(t)\|_2^2) \lesssim 1 + e^{11T_0/4}\|Er(0)\|_2^2 + e^{3T_0/4}. \tag{74}$$

$\square$

Finally we are ready to prove the lower and upper bounds on the limiting value of column norms of $U_t$ as $t \to \infty$. Lemma 19 establishes the lower bound, while Lemma 20 establishes the upper bound.

**Lemma 19.** *When the initialization parameter $\alpha \leq c/k^3 d \log(kd)$, then with probability $\geq 1 - O(1/k)$ over the initialization, for any $t \geq 4T_0$,*

$$\|U(t)e_i\|_2 \geq (1 - \eta)\frac{|\langle r(0), e_i\rangle|}{\|r(0)\|_2}. \tag{75}$$

*Proof.* From the differential equation governing $r$, eq. (24), for any coordinate $i \in [k]$,

$$\frac{d\langle r, e_i\rangle^2}{dt} = 2\langle r, e_i\rangle^2(1 - \|r\|_2^2) - 2\langle r, e_i\rangle e_i E^T Er \tag{76}$$

$$\geq 2\langle r, e_i\rangle^2(1 - \|r\|_2^2) - 2|\langle r, e_i\rangle|\|E\|_F\|Er\|_2 \tag{77}$$

From Lemma 14 and Lemma 16, and since $\|E(t)\|_F \leq \|E(0)\|_F$, at any time $t \geq 0$,

$$\frac{d\langle r, e_i\rangle^2}{dt} \geq \frac{2\langle r, e_i\rangle^2}{e^{2(t-T_0)+1+\eta} + 1} - \frac{2|\langle r, e_i\rangle|\|E(0)\|_F\|Er(0)\|_F e^t}{1 + e^{2(1 - \|E(0)\|_F^2)(t-T_0) - 1}}$$

$$\implies \frac{d|\langle r, e_i\rangle|}{dt} - \frac{|\langle r, e_i\rangle|}{e^{2(t-T_0)+1+\eta} + 1} \geq -\frac{\|E(0)\|_F^2 e^t}{1 + e^{2(1 - \|E(0)\|_F^2)(t-T_0) - 1}} \tag{78}$$

Define $q(t) = e^{-2(t-T_0)+1+\eta}$ (which we abbreviate simply as $q$) and multiply both sides by $\sqrt{1+q}$,

$$\sqrt{1+q}\frac{d|\langle r, e_i\rangle|}{dt} - |\langle r, e_i\rangle|\frac{q}{\sqrt{1+q}} \geq -\frac{2\|E(0)\|_F^2 e^t \sqrt{1+q}}{1 + e^{2(1-\|E(0)\|_F^2)(t-T_0)-1}} \tag{79}$$

Noting that $\frac{d}{dt}\sqrt{1+q} = -\frac{q}{\sqrt{1+q}}$, we get that,

$$\implies \frac{d}{dt}\left(|\langle r, e_i\rangle|\sqrt{1+q}\right) \geq -2\|E(0)\|_F^2 e^{-(t-T_0)+T_0-2} \cdot \frac{\sqrt{1+e^{2(t-T_0+1)}}}{1 + e^{2(1-\|E(0)\|_F^2)(t-T_0)-1}}. \tag{80}$$

We further lower bound the RHS for the case when $t \leq 4T_0$. Later we will give a different proof to show that when $t \geq 4T_0$, $|\langle r, e_i\rangle|$ does not change significantly then onward.

For $t \leq 4T_0$, $(1-\|E(0)\|_F^2)(t-T_0) \geq t - T_0 - 4$. This follows from the same analysis as eqs. (33) and (35). In this regime, we therefore have that,

$$\frac{\sqrt{1+e^{2(t-T_0)+1+\eta}}}{1 + e^{2(1-\|E(0)\|_F^2)(t-T_0)-1}} \lesssim \frac{1}{\sqrt{1+e^{2(t-T_0)+1+\eta}}} \tag{81}$$

On the other hand,

$$e^{-(t-T_0)+T_0-2} \lesssim e^{T_0}\sqrt{1+e^{-2(t-T_0)+1+\eta}} \tag{82}$$

Multiplying eqs. (81) and (82) results in the lower bound,

$$\frac{d}{dt}\left(|\langle r, e_i\rangle|\sqrt{1+q}\right) \gtrsim -\|E(0)\|_F^2 e^{T_0} \tag{83}$$

Integrating both sides from $t = 0$ to $4T_0$, and since $T_0 \geq 1$ by the small initialization bound,

$$|\langle r(4T_0), e_i\rangle| \geq |\langle r(0), e_i\rangle|\sqrt{\frac{1+e^{2T_0-1-\eta}}{1+e^{-4T_0}}} - \frac{c_{84}\|E(0)\|_F^2 T_0 e^{T_0}}{\sqrt{1+e^{-4T_0}}} \tag{84}$$

where $c_{84} > 0$ is an absolute constant. Since $e^{T_0} \geq -\log(\|r(0)\|_2) + 1/2$ from Lemma 13, and since $\|r(0)\|_2^2 \leq \eta$ by the small initialization, the term multiplying $|\langle r(0), e_i\rangle|$ on the RHS is upper bounded by $\sqrt{\frac{1+e^{2T_0-1-\eta}}{1+e^{-T_0}}} \geq (1-2\eta)e^{T_0-1/2-\eta/2} \geq (1-3\eta)e^{T_0-1/2}$. On the other hand, noting again by Lemma 13 that $T_0 \leq -\log(\|r(0)\|_2) + 3/2$, and since $\|r(0)\|_2^2 \gtrsim k\alpha^2$ and $\|E(0)\|_F^2 \lesssim \alpha^2 kd$ and furthermore $|\langle r(0), e_i\rangle| \geq \alpha/k^2$ from Lemma 11, as long as $\alpha \leq c\eta/k^3 d\log(kd)$ for a sufficiently small constant $c > 0$,

$$|\langle r(0), e_i\rangle| \geq \frac{c_{84}\|E(0)\|_F^2 T_0}{\eta\sqrt{1+e^{-4T_0}}}. \tag{85}$$

And by implication, eq. (84) gives,

$$|\langle r(4T_0), e_i\rangle| \geq (1-4\eta)|\langle r(0), e_i\rangle|e^{T_0-1/2}, \tag{86}$$

where the last inequality follows from the lower bound on $T_0$ in Lemma 13.

Finally we show that, from time $4T_0$ onward, $\langle r(t), e_i\rangle$ does not change much. From eq. (78),

$$\frac{d|\langle r, e_i\rangle|}{dt} \geq -\frac{\|E(0)\|_F^2 e^t}{1 + e^{2(1-\|E(0)\|_F^2)(t-T_0)-1}} \gtrsim -\frac{\|E(0)\|_F^2 e^t}{e^{(9/5)(t-T_0)}}. \tag{87}$$

where the last inequality uses the fact that $\|E(0)\|_F^2 \leq 1/10$ by Lemma 11 and the upper bound on the initialization scale $\alpha \leq c\eta^2/k^3 d\log(kd)$. Integrating both sides from $4T_0$ to $t$,

$$|\langle r(t), e_i\rangle| - |\langle r(4T_0), e_i\rangle| \gtrsim -\|E(0)\|_F^2 e^{9T_0/5}\int_{4T_0}^{\infty} e^{-4t/5}dt \tag{88}$$

$$\gtrsim -\|E(0)\|_F^2 e^{-7T_0/5} \tag{89}$$

$$\overset{(i)}{\gtrsim} -|\langle r(0), e_i\rangle|e^{-7T_0/5} \tag{90}$$

where $(i)$ follows from eq. (85). Finally, combining with eq. (86),

$$|\langle r(t), e_i\rangle| \geq (1 - 4\eta)|\langle r(0), e_i\rangle| \left(e^{T_0 - 1/2} - c_{91}e^{-7T_0/5}\right) \tag{91}$$

for some absolute constant $c_{91} > 0$. By noting that $T_0 \geq -\log\|r(0)\|_2 + 1/2$ from Lemma 13 and the fact that $\|r(0)\|_2 \lesssim \alpha^2 k$, by choosing $\alpha \leq c'_{91}/\eta\sqrt{k}$ for some absolute constant $c'_{91} > 0$ results in the inequality,

$$\forall t \geq T_0, \ |\langle r(t), e_i\rangle| \geq (1 - 5\eta)|\langle r(0), e_i\rangle|e^{T_0 - 1/2} \geq (1 - 5\eta)\frac{|\langle r(0), e_i\rangle|}{\|r(0)\|_2}. \tag{92}$$

Since $\|Ue_i\|_2 \geq |\langle r(t), e_i\rangle|$, this completes the proof. $\qquad\square$

**Lemma 20.** *Suppose* $\alpha \leq c\eta^2/\eta k^3 d \log(kd)$ *for a sufficiently small absolute constant* $c > 0$. *Then with probability* $\geq 1 - O(1/k)$ *over the initialization, for any* $t \geq 3T_0/2$,

$$\|U(t)e_i\|_2 \leq (1 + 4\sqrt{\eta})\frac{|\langle r(0), e_i\rangle|}{\|r(0)\|_2}. \tag{93}$$

*Proof.* Following the proof of the lower bound in Lemma 19, from the differential equation governing $r$, for any coordinate $i \in [k]$,

$$\frac{d\langle r, e_i\rangle^2}{dt} = 2\langle r, e_i\rangle^2(1 - \|r\|_2^2) - 2\langle r, e_i\rangle e_i E^T Er \tag{94}$$

$$\leq 2\langle r, e_i\rangle^2(1 - \|r\|_2^2) + 2|\langle r, e_i\rangle|\|E\|_F\|Er\|_2 \tag{95}$$

$$\implies \frac{d|\langle r, e_i\rangle|}{dt} \leq |\langle r, e_i\rangle|(1 - \|r\|_2^2) + \frac{\|E(0)\|_F\|Er(0)\|_2 e^t}{1 + e^{2(1 - \|E(0)\|_F^2)(t - T_0) - 1}} \tag{96}$$

where in the last inequality, we bound $\|E\|_F \leq \|E(0)\|_F$ and apply Lemma 16 to upper bound the error term $\|Er(t)\|_2$. Akin to Lemma 19, we carry out the analysis of $|\langle r(t), e_i\rangle|$ in two parts, we first analyze its growth from time 0 to $3T_0/2$. From time $3T_0/2$ to $t$ we show that $|\langle r(t), e_i\rangle|$ does not change significantly.

In particular, at any time $t \leq 3T_0/2$, since $(1 - \|E(0)\|_F^2)(t - T_0) \geq t - T_0 - 4$ (see the analysis in eqs. (33) and (35)), for some absolute constant $C_{97}$,

$$\frac{d|\langle r, e_i\rangle|}{dt} \leq |\langle r, e_i\rangle|(1 - \|r\|_2^2) + C_{97}\frac{\|E(0)\|_F\|Er(0)\|_2 e^t}{1 + e^{2(t - T_0)}} \tag{97}$$

$$\leq |\langle r, e_i\rangle|(1 - \|r\|_2^2) + C_{97}\|E(0)\|_F^2\|r(0)\|_2 e^{T_0}, \tag{98}$$

where the last inequality uses the fact that $e^t/(1 + e^{2(t - T_0)})$ is maximized at $t = T_0$. Plugging in the lower bound on $\|r\|_2^2$ in Lemma 17,

$$\frac{d|\langle r, e_i\rangle|}{dt} \leq |\langle r, e_i\rangle|\left(\frac{\|r(0)\|_2^{-2} - 1}{\|r(0)\|_2^{-2} + e^{2t} - 1} + C_{55}\frac{\|E(0)\|_F^2 T_0}{\|r(0)\|_2}\right) + C_{97}\|E(0)\|_F^2\|r(0)\|_2 e^{T_0}$$

$$\leq |\langle r, e_i\rangle|\left(\frac{p}{p + e^{2t}}\right) + C_{99}\|E(0)\|_F^2 T_0. \tag{99}$$

where $C_{99} > 0$ is a sufficiently large absolute constant, and $p = \|r(0)\|_2^{-2} - 1$. Multiplying both sides by $y = \sqrt{pe^{-2t} + 1}$ and noting that $\frac{dy}{dt} = -\left(\frac{p}{p + e^{2t}}\right)y$, we get,

$$\frac{d}{dt}\left(y|\langle r, e_i\rangle|\right) = C_{99}\|E(0)\|_F^2 T_0. \tag{100}$$

Integrating both sides from 0 to $3T_0/2$,

$$|\langle r(3T_0/2), e_i\rangle|\sqrt{pe^{-3T_0} + 1} - |\langle r(0), e_i\rangle|\sqrt{1 + p} = C'_{99}\|E(0)\|_F^2 T_0^2 \tag{101}$$

$$\implies |\langle r(3T_0/2), e_i\rangle| \leq C'_{99}\|E(0)\|_F^2 T_0^2 + \frac{|\langle r(0), e_i\rangle|}{\|r(0)\|_2} \tag{102}$$

By Lemma 11, at initialization, $\|r(0)\|_2^2 \lesssim \alpha\sqrt{k}$ and $|\langle r(0), e_i \rangle| \geq \alpha/k^2$. Therefore, by the small initialization condition, $C'_{99}\|E(0)\|_F^2 T_0^2 \leq \eta |\langle r(0), e_i \rangle|/\|r(0)\|_2$ and,

$$|\langle r(3T_0/2), e_i \rangle| \leq (1+\eta)\frac{|\langle r(0), e_i \rangle|}{\|r(0)\|_2}. \tag{103}$$

Next we show that from $3T_0/2$ to $t$, $|\langle r(t), e_i \rangle|$ does not change significantly. By the same analysis as eq. (96),

$$\frac{d|\langle r, e_i \rangle|}{dt} \leq |\langle r, e_i \rangle|(1 - \|r\|_2^2) + \frac{2\|E(0)\|_F^2 e^t}{1 + e^{2(1-\|E(0)\|_F^2)(t-T_0)-1}} \tag{104}$$

Plugging in the upper bound on $1 - \|r(t)\|_2^2$ from Lemma 18 and simplifying,

$$\frac{d|\langle r, e_i \rangle|}{dt} \lesssim |\langle r, e_i \rangle| e^{-3t/4} \left(e^{3T_0/4} + e^{11T_0/4}\|Er(0)\|_2^2\right) + \frac{2\|E(0)\|_F^2 e^t}{e^{2\left(1-\|E(0)\|_F^2\|r(0)\|_2\right)(t-T_0)}}, \tag{105}$$

$$\overset{(i)}{\leq} |\langle r, e_i \rangle| e^{-3t/4} \left(e^{3T_0/4} + e^{11T_0/4}\|Er(0)\|_2^2\right) + \frac{2\|E(0)\|_F^2\|r(0)\|_2 e^t}{e^{7/4(t-T_0)}} \tag{106}$$

$$= |\langle r, e_i \rangle| e^{-3t/4} \left(e^{3T_0/4} + e^{11T_0/4}\|Er(0)\|_2^2\right) + 2\|E(0)\|_F^2\|r(0)\|_2 e^{-3t/4} e^{7T_0/4}, \tag{107}$$

$$= \left(|\langle r, e_i \rangle| \left(1 + e^{2T_0}\|Er(0)\|_2^2\right) + 2\|E(0)\|_F^2\|r(0)\|_2 e^{T_0}\right) e^{-3(t-T_0)/4}, \tag{108}$$

where $(i)$ invokes the fact that $\|E(0)\|_F^2 \leq 1/8$ from Lemma 11, by the bound on the initialization scale $\alpha \leq c\eta/k^3 d\log(kd)$. Simplifying the terms in eq. (108),

$$e^{2T_0}\|Er(0)\|_2^2 \leq e^{2T_0}\|E(0)\|_2^2\|r(0)\|_2^2 \overset{(i)}{\lesssim} \|E(0)\|_2^2 \leq 1, \tag{109}$$

where $(i)$ is by Lemma 13 and the last equation follows from the scaling of $\alpha$ and Lemma 11. Likewise, $\|E(0)\|_F^2\|r(0)\|_2 e^{T_0} \lesssim \|E(0)\|_F^2$, and plugging this and eq. (109) into eq. (108),

$$\frac{d|\langle r, e_i \rangle|}{dt} \lesssim \left(|\langle r, e_i \rangle| + 2\|E(0)\|_F^2\right) e^{-3(t-T_0)/4} \tag{110}$$

Integrating both sides,

$$\log \frac{|\langle r(t), e_i \rangle| + 2\|E(0)\|_F^2}{|\langle r(3T_0/2), e_i \rangle| + 2\|E(0)\|_F^2} \lesssim \int_{3T_0/2}^{\infty} e^{-3(t-T_0)/4} dt \lesssim e^{-3T_0/8}. \tag{111}$$

Therefore, for some absolute constant $C_{112} > 0$,

$$|\langle r(t), e_i \rangle| \leq \left(|\langle r(3T_0/2), e_i \rangle| + 2\|E(0)\|_F^2\right)\left(1 + C_{112}e^{-3T_0/8}\right) \tag{112}$$

$$\overset{(i)}{\leq} \left(|\langle r(3T_0/2), e_i \rangle| + 2\|E(0)\|_F^2\right)\left(1 + \sqrt{\eta}\right) \tag{113}$$

$$\implies \|U(t)e_i\|_2 \overset{(ii)}{\leq} \|E(0)\|_F + \left(|\langle r(3T_0/2), e_i \rangle| + 2\|E(0)\|_F^2\right)\left(1 + \sqrt{\eta}\right), \tag{114}$$

where $(i)$ lower bounds $T_0$ using Lemma 13 and uses the fact that at initialization $\|r(0)\|_2 \gtrsim \alpha\sqrt{k}$ and therefore when $\alpha \leq c\eta^{4/3}$ for a sufficiently small constant $c > 0$, $C_{112}e^{-3T_0/8} \leq \sqrt{\eta}$. On the other hand, $(ii)$ uses triangle inequality to bound $\|U(t)e_i\|_2 \leq |\langle r(t), e_i \rangle| + \|E(t)e_i\|_2 \leq |\langle r(t), e_i \rangle| + \|E(t)\|_F \leq |\langle r(t), e_i \rangle| + \|E(0)\|_F$ by Lemma 12. Using the bound on $|\langle r(3T_0/2), e_i \rangle|$ in eq. (103),

$$\|U(t)e_i\|_2 \leq \|E(0)\|_F + \left(\frac{|\langle r(0), e_i \rangle|}{\|r(0)\|_2} + 2\|E(0)\|_F^2\right)\left(1 + \sqrt{\eta}\right) \tag{115}$$

$$\leq \frac{|\langle r(0), e_i \rangle|}{\|r(0)\|_2}(1 + 4\sqrt{\eta}). \tag{116}$$

The last inequality uses the fact that at initialization, $|\langle r(0), e_i \rangle| \geq \alpha/k^2$ and $\|r(0)\|_2 \lesssim \alpha\sqrt{k}$. Therefore, $|\langle r(0), e_i \rangle|/\|r(0)\|_2 \geq 1/k^{5/2}$ and $\|E(0)\|_F \leq \alpha\sqrt{kd}$. Therefore, by the small initialization condition on $\alpha$, $\|E(0)\|_F, \|E(0)\|_F^2 \leq \eta|\langle r(0), e_i \rangle|/\|r(0)\|_2$.

$\square$

Combining the statements of Lemma 19 and Lemma 20 shows that, with probability $\geq 1 - O(1/k)$ over the initialization, at any time $t \geq 4T_0$,

$$\frac{|\langle r(0), e_i\rangle|}{\|r(0)\|_2}(1 - 5\eta) \leq \|U(t)e_i\|_2 \leq \frac{|\langle r(0), e_i\rangle|}{\|r(0)\|_2}(1 + 4\sqrt{\eta}). \tag{117}$$

This completes the proof of Lemma 10.

### A.3 Behavior at initialization: many columns are "active"

Lemma 10 establishes the limiting behavior of gradient flow as a function of the initialization, specifically that the norm of a column grows to a value proportional to the alignment of the column with $U_\star$ at initialization, $|\langle r(0), e_i\rangle|$. In this section, we show that at initialization, $\langle r(0), e_i\rangle$ is significant for many $i \in [d]$ compared to the maximum among them.

At time $t = 0$, each coordinate of $r(0)$ is distributed as the inner product of a Gaussian vector $\sim \mathcal{N}(0, \alpha^2 I)$ with a fixed vector $U_\star$. By Gaussian concentration, we therefore expect each coordinate of $r(0)$ to concentrate around $\alpha$, and no one coordinate of $r(0)$ to be significantly larger than the others. The next lemma makes this claim precise.

**Lemma 21.** *Many columns are "active" at initialization  For any $\eta > 0$, let $S_{\max}(\eta)$ denote the set $\left\{i \in [k] : \frac{|\langle r(0), e_i\rangle|}{\max_{j \in [k]} |\langle r(0), e_j\rangle|} \geq 1 - \sqrt{\eta}\right\}$. For any $\eta \leq 1 - \frac{1}{\log(k)}$,*

$$\mathbb{P}\left(|S_{\max}(\eta)| \leq \frac{k^\eta}{2\sqrt{2(1 - \eta)\log(k)}}\right) \leq \frac{c_{118}\sqrt{\log(k)}}{k^\eta}. \tag{118}$$

*for a sufficiently small absolute constant $c_{118} > 0$. In other words, with high probability, $\widetilde{\Omega}(k^\eta)$ of the column indices $i \in [k]$ are significantly correlated with $U_\star$ compared to the maximum among all columns.*

*Proof.* By rotation invariance of Gaussians and since $\|U_\star\|_2 = 1$, for each $i$, $\langle r(0), e_i\rangle \overset{\text{i.i.d.}}{\sim} \mathcal{N}(0, \alpha^2)$. By Gaussian anti-concentration, for each $i \in [k]$, from [51, Proposition 2.1.2], for $t > 0$,

$$\mathbb{P}\left(|\langle r(0), e_i\rangle| \geq t\alpha\right) \geq \left(\frac{1}{t} - \frac{1}{t^3}\right) \cdot \frac{1}{\sqrt{2\pi}} e^{-t^2/2} \triangleq p(t). \tag{119}$$

Therefore roughly we expect $kp(t)$ of the columns $i \in [k]$ to satisfy the condition $\{|\langle r(0), e_i\rangle| \geq t\alpha\}$. In particular, by the independence across $i$, the number of successes follows a binomial distribution with number of trials $k$ and probability of success $p(t)$. Denote $S(t) = \{i \in [k] : |\langle r(0), e_i\rangle| \geq t\alpha\}$. By binomial concentration [51],

$$\mathbb{P}\left(|S(t)| \leq \frac{kp(t)}{2}\right) \leq e^{-\frac{kp(t)}{4}}. \tag{120}$$

On the other hand, for i.i.d standard normal random variables, $X_1, \cdots, X_k \sim \mathcal{N}(0, 1)$, the supremum satisfies the following concentration inequality

$$\mathbb{P}\left(\max_{1 \leq i \leq k} X_i \geq \sqrt{2\log(k)} + \eta\right) \leq e^{-\eta^2/2} \tag{121}$$

Therefore,

$$\mathbb{P}\left(\max_{i \in [k]} |\langle r(0), e_i\rangle| \leq \alpha\sqrt{2\log(k)} + \alpha\sqrt{2\eta\log(k)}\right) \geq 1 - k^{-\eta}. \tag{122}$$

For each $t > 0$, define,

$$S_{\max}(t) = \left\{i \in [k] : \frac{|\langle r(0), e_i\rangle|}{\max_{i \in [k]} |\langle r(0), e_i\rangle|} \geq \frac{t}{\sqrt{2\log(k)}(1 + \sqrt{\eta})}\right\}, \tag{123}$$

and combining eqs. (119), (120) and (122) results in the inequality,

$$\mathbb{P}\left(|S_{\max}(t)| \leq \frac{kp(t)}{2}\right) \leq e^{-\frac{kp(t)}{4}} + k^{-1/8} \leq \frac{4}{kp(t)} + k^{-\eta}. \tag{124}$$

With the choice of $t = \sqrt{2(1-\eta)\log(k)}$ and for any $\eta \leq 1 - \frac{1}{\log(k)}$, we obtain that $p(t) \geq k^{-(1-\eta)}/4\sqrt{(1-\eta)\log(k)}$. Plugging into eq. (124) and using the results in the bound,

$$\mathbb{P}\left(|S_{\max}(\sqrt{2(1-\eta)\log(k)})| \leq \frac{k^\eta}{4\sqrt{(1-\eta)\log(k)}}\right) \leq \frac{1 + 16\sqrt{2(1-\eta)\log(k)}}{k^\eta} \tag{125}$$

Using the definition of $S_{\max}(\sqrt{2(1-\eta)\log(k)})$ and the fact that $\frac{\sqrt{1-\eta}}{1+\sqrt{\eta}} \geq 1 - \sqrt{\eta}$ completes the proof. $\qquad \square$

### A.4 Putting it all together: Proof of Theorem 1

Below we state and prove a stronger version of Theorem 1 which follows by combining Lemmas 10 and 21.

**Theorem 22.** *Consider the population loss defined in (2), for the case that $r = 1$ and $k \gg 1$. Moreover, assume that the entries of the initial model $U_0$ are i.i.d. samples from $\mathcal{N}(0, \alpha^2)$, where $\alpha \leq \eta^2/k^3 d\log(kd)$. Then, for any constant $\eta \in (0, 1)$, the iterates generated by gradient descent converge to a model $U_{gd}$ where $\widetilde{\Omega}(k^\eta)$ columns of which are active and satisfy,*

$$\frac{\|U_{gd}e_i\|_2}{\max_{j\in[k]}\|U_{gd}e_j\|_2} \geq 1 - C_4\sqrt{\eta}, \tag{126}$$

*with probability $\geq 1 - poly(1/d, 1/k^\eta)$, where $C_4 > 0$ is a sufficiently large constant.*

In particular, if for some $i$,

$$\frac{|\langle r(0), e_i\rangle|}{\max_{j\in[k]}|\langle r(0), e_j\rangle|} \geq 1 - \sqrt{\eta} \tag{127}$$

Then, by Lemma 10,

$$\lim_{t\to\infty} \frac{\|U(t)e_i\|_2}{\max_{j\in[k]}\|U(t)e_j\|_2} \geq 1 - C_{128}\sqrt{\eta}. \tag{128}$$

where $C_{128} > 0$ is a sufficiently large constant. Invoking Lemma 21 completes the proof of Theorem 22.

## B Population analysis of the regularized loss: Proof of Theorem 3

In this section, we study the second-order stationary points of loss $f_{\text{pop}}(U) = \mathcal{L}_{\text{pop}}(U) + \lambda\mathcal{R}_\beta(U)$ ((9)) which is the regularized loss in the population (infinite sample) setting. The main result we prove in this section is about approximate second-order stationary points of the regularized loss $f_{\text{pop}}(U)$. When $\lambda$ and $\beta$ are chosen appropriately, we show that such points $(i)$ are "pruning friendly" in that greedily pruning the columns of $U$ based on their $L_2$ norm results in a solution $U_{\text{prune}}$ having exactly $r$ columns, and $(ii)$ the resulting solution $U_{\text{prune}}$ satisfies $\|U_{\text{prune}}U_{\text{prune}}^T\|_F^2 \leq c(\sigma_r^\star)^2$ where $c > 0$ is a small constant, and serves essentially as a "spectral initialization" in the sense of [23] for the subsequent fine-tuning phase.

The proof of Theorem 3 relies on two main observations: $(i)$ showing that at approximate second-order stationary points $UU^T \approx U_\star U_\star^T$, where the error is small when $\lambda$ is small, $(ii)$ the regularizer ensures that the columns of $U$ that are not too small in $\ell_2$-norm are all approximately orthogonal to one another, in that the angle between pairs of vectors is $\approx 90°$. Since the columns are orthogonal, the rank of $U$ equals the number of non-zero columns. However, $U$ is close to a rank $r$ matrix since $UU^T \approx U_\star U_\star^T$. This will imply that $U$ also has approximately $r$ non-zero columns. Moreover, pruning away the columns of $U$ at the correct threshold will result in a model having exactly $r$ columns remaining, while at the same time not affecting the population loss significantly.

The intuition behind $UU^T \approx U_\star U_\star^T$ at second order stationary points is straightforward - prior work [36] characterizes the behavior of such points for matrix sensing in the absence of any regularization, showing that $UU^T = U_\star U_\star^T$ at second-order stationary points, and establishing a strict saddle condition for the population loss $\mathcal{L}_{\text{pop}}$. In the presence of regularization, as long as $\lambda$ is small, we do

not expect the behavior to change significantly. As we will discuss in more detail in Appendix D, the regularizer $\mathcal{R}_\beta$ satisfies gradient and Hessian-Lipschitzness as long as $\beta$ is not too small. This will suffice in showing that the locations of first and second order stationary points do not change significantly when $\lambda$ is small.

As introduced earlier in eq. (11), the main result we prove in this section is that the "bounded" (in operator norm) $(\epsilon, \gamma)$-approximate second-order stationary points of $f_{\text{pop}}$ returned by the optimization oracle $\mathcal{O}$, in Algorithm 1 satisfies the following condition,

$$\forall i, j : \|Ue_i\|_2, \|Ue_j\|_2 \geq 2\sqrt{\beta}, \quad \frac{\langle Ue_i, Ue_j \rangle}{\|Ue_i\|_2 \|Ue_j\|_2} \approx 0.$$

In other words, all the large columns of $U$ have their pairwise angle approximately $90°$. By pruning away the columns of $U$ that have an $\ell_2$ norm less than $2\sqrt{\beta}$, the remaining columns of $U$, i.e., which are a superset of the columns of $U_{\text{prune}}$, are now approximately at $90°$ angles to one another. But since $UU^T \approx U_\star U_\star^T$, we know that there can be at most $r$ significant columns of $U$. Therefore, after pruning the resulting model has at most $r$ columns.

Note that the above discussion does not absolve the risk of over-pruning the model. If the pruning threshold is not chosen carefully, it might be possible to end up with a model having fewer than $r$ columns. Such a model cannot generalize to a vanishing error even in the population setting. As we will show, it is possible to establish that error of the pruned model, $\|U_{\text{prune}} U_{\text{prune}}^T - U_\star U_\star^T\|_F$ is also small, which comes to the rescue. We can show that it is at most $\frac{1}{2}(\sigma_r^\star)^2$. On the other hand if $U_{\text{prune}}$ indeed had fewer than $r$ columns, the approximation error must be at least $(\sigma_r^\star)^2$. This shows that the greedy pruning strategy we employ is not too aggressive.

## B.1 Proof outline of Theorem 3

In Lemma 23 we begin with a lower bound on the gradient norm of the reulgarized loss. The lower bound that at exact stationary points, every pair of columns $i, j \in [d]$ will either have $\|Ue_i\|_2 = \|Ue_j\|_2$, or will be orthogonal to each other. Next we use this calculation to show that at approximate second order stationary points of the regularized loss, every pair of columns which are not too small in $\ell_2$ norm are approximately orthogonal to each other. Next, in Lemma 27 we show that at approximate second order stationary points of the loss, we also expect $\|UU^T - U_\star U_\star^T\|_F$ to be small when $\lambda$ is small. In Lemmas 29 and 30 we show that first-order stationary points are $U$ are aligned with the correct subspace induced by the columns of $U_\star$. Finally, we combine these results in Theorem 32 to show bounds on the generalization loss of the pruned solution as a function of the problem parameters. By instantiating the problem parameters properly, we prove Theorem 3 in Appendix B.2.

Note that the gradient and Hessian of $\mathcal{R}_\beta(U)$ is calculated in Appendix E.3 and are in terms of diagonal matrices $D(U)$ and $G(U)$ defined in eqs. (330) and (331).

**Lemma 23** (A lower bound on the gradient norm). *Consider any loss function $\mathcal{L}$ of the form $f(UU^T)$ where $f : \mathbb{R}^{d \times d} \to \mathbb{R}$ is a differentiable function. For any candidate matrix $U \in \mathbb{R}^{d \times k}$,*

$$\|\nabla(\mathcal{L} + \lambda \mathcal{R}_\beta)(U)\|_F^2 \geq \lambda^2 \max_{i \neq j \in [k]} (D(U)_{ii} - D(U)_{jj})^2 \frac{\langle Ue_i, Ue_j \rangle^2}{\|Ue_i\|_2^2 + \|Ue_j\|_2^2} \tag{129}$$

*Proof.* Note that $\|\nabla(\mathcal{L} + \lambda \mathcal{R}_\beta)(U)\|_F^2 \geq \langle Z, \nabla(\mathcal{L} + \lambda \mathcal{R}_\beta)(U) \rangle^2$ for any candidate $Z \in \mathbb{R}^{d \times k} : \|Z\|_F \leq 1$. The rest of this proof will be dedicated to finding such a $Z$.

Note from Lemma 41 that $\langle \mathcal{L}(U), Z \rangle = \langle (\nabla f)(UU^T), UZ^T + ZU^T \rangle$. Suppose the perturbation $Z$ is chosen as $UW$ where $W$ is a skew-symmetric matrix. Then, $UZ^T + ZU^T = U(W + W^T)U^T = 0$. Therefore,

$$\langle \nabla \mathcal{L}(U), Z \rangle = 0 \tag{130}$$

On the other hand, from Lemma 43,

$$\langle \nabla \mathcal{R}_\beta(U), Z \rangle = \text{Tr}(D(U)U^T Z) = \text{Tr}(D(U)U^T UW). \tag{131}$$

where $D(U)$ is defined in eq. (330). Define $(i,j)$ as an arbitrary pair of distinct indices in $[k]$. Suppose $W = e_i e_j^T - e_j e_i^T$. Then,

$$\langle \nabla \mathcal{R}_\beta(U), Z \rangle^2 = (e_i^T (D(U) U^T U - U^T U D(U)) e_j)^2 \tag{132}$$

$$= (D(U)_{ii} - D(U)_{jj})^2 \langle U e_i, U e_j \rangle^2. \tag{133}$$

Dividing throughout by $\|Z\|_F^2 = \|UW\|_F^2 = \|U e_i\|_F^2 + \|U e_j\|_F^2$ and noting that $i \neq j$ are arbitrary completes the proof. $\square$

**Remark 24.** *The interpretation of the lower bound on the gradient norm in Lemma 23 is best understood by looking at its behavior at first-order stationary points, where the gradient is $0$. At a first-order stationary point, for any $i, j \in [k]$ such that $i \neq j$, either $D(U)_{ii} = D(U)_{jj}$ or $U e_i \perp U e_j$. The former condition is true iff $\|U e_i\|_2 = \|U e_j\|_2$.*

Next we prove a result which establishes near-orthogonality of the columns of $U$ at approximate second-order stationary points.

**Lemma 25.** *Consider any loss $\mathcal{L}(U)$ of the form $f(UU^T)$ where $f : \mathbb{R}^{d \times d} \to \mathbb{R}$ is doubly differentiable. Consider any $\epsilon$-approximate first-order stationary point of $\mathcal{L} + \lambda \mathcal{R}_\beta$, denoted $U$. Consider any $i \neq j \in [k]$ and define $C_{ij} = \frac{\langle U e_i, U e_j \rangle}{\sqrt{\|U e_i\|_2^2 + \|U e_j\|_2^2}}$. Then, if*

1. *$\min\{\|U e_i\|_2, \|U e_j\|_2\} \leq 2\sqrt{\beta}$, then, $|C_{ij}| \leq 2\sqrt{\beta}$.*

2. *In the complement case, defining $\|Z\|_F = \sqrt{\|U e_i\|_2^2 + \|U e_j\|_2^2}$, if,*

$$|C_{ij}| \geq 5 \sqrt{\frac{\epsilon + \gamma \min\{\|U e_i\|_2, \|U e_i\|_2\}}{\lambda}} \cdot \min\{\|U e_i\|_2, \|U e_j\|_2\}. \tag{134}$$

*then the Hessian of the regularized loss at $U$ has a large negative eigenvalue,*

$$\lambda_{\min}(\nabla^2 (\mathcal{L} + \lambda \mathcal{R}_\beta)(U)) < -\gamma. \tag{135}$$

*In summary, at second order stationary points the columns are approximately orthogonal in the sense that $|C_{ij}|$ is small.*

*Proof.* Note that, regardless of whether $U$ is approximately stationary or not, for any $i \neq j \in [k]$, if $\min\{\|U e_i\|_2, \|U e_j\|_2\} \leq 2\sqrt{\beta}$, then,

$$\frac{|\langle U e_i, U e_j \rangle}{\sqrt{\|U e_i\|_2^2 + \|U e_j\|_2^2}} \leq \frac{\|U e_i\|_2 \|U e_j\|_2}{\max_{i \in \{1,2\}} \|U e_i\|_2^2} = \min_{i \in \{1,2\}} \|U e_i\|_2 \leq 2\sqrt{\beta}. \tag{136}$$

This proves the first part of the theorem. Henceforth, we will assume that $\min\{\|U e_i\|_2, \|U e_j\|_2\} \geq 2\sqrt{\beta}$. This condition will turn out to lower bound the Frobenius norm of a matrix $Z$ which we will require later for the proof of the second part.

The approximate first-order stationarity of $U$ implies that

$$\|\nabla(\mathcal{L} + \lambda \mathcal{R}_\beta)(U)\|_F \leq \epsilon. \tag{137}$$

As in the proof of Lemma 23, the proof strategy is to construct explicit directions (matrices) capturing the directions of negative curvature of the regularized loss. From Lemma 23, the approximate first-order stationarity of $U$ implies,

$$\frac{|\langle U e_i, U e_j \rangle|}{\sqrt{\|U e_i\|_2^2 + \|U e_j\|_2^2}} \leq \frac{\epsilon}{\lambda} \cdot \frac{1}{|\Delta|}. \tag{138}$$

where $\Delta = D(U)_{ii} - D(U)_{jj}$. Consider any tuple $i \neq j \in [k]$. Without loss of generality we will work with $i = 1, j = 2$.

Consider a perturbation $Z \in \mathbb{R}^{d \times k}$ which takes the form $Z = UW$ where $W$ is a skew-symmetric matrix which satisfies the "support condition",

$$\forall i' \geq 3, \ W e_{i'} = 0, e_{i'}^T W = 0 \tag{139}$$

In general, when $\Delta$ is small in absolute value, by the support condition eq. (139) of $W$, one would expect $WD(U) \approx D(U)W$ in case $\Delta$ is small. In particular, defining the diagonal matrix $L = \mathsf{diag}\left(\frac{\Delta}{2}, -\frac{\Delta}{2}, 0, \cdots\right)$, we have that,

$$WD(U) - D(U)W = WL - LW. \tag{140}$$

From the gradient and Hessian computations in Lemma 43,

$$\mathsf{vec}(Z)^T[\nabla^2 \mathcal{R}_\beta(U)]\mathsf{vec}(Z)$$

$$= \mathsf{Tr}(D(U) \cdot Z^T Z) - \sum_{i=1}^{k} G(U)_{ii}\langle Ze_i, Ue_i\rangle^2 \tag{141}$$

$$= \mathsf{Tr}(WD(U)Z^T U) - \sum_{i=1}^{k} G(U)_{ii}\langle Ze_i, Ue_i\rangle^2 \tag{142}$$

$$= \langle \nabla \mathcal{R}_\beta(U), WZ^T\rangle + \mathsf{Tr}((WL - LW) \cdot Z^T U) - \sum_{i=1}^{k} G(U)_{ii}\langle Ze_i, Ue_i\rangle^2 \tag{143}$$

where the last equation uses the gradient computation in Lemma 43 and the fact that $W$ and $D(U)$ approximately commute per eq. (140). Likewise, analyzing the second order behavior of $\mathcal{L}$ using the Hessian computations in Lemma 41 results in the following set of equations,

$$\mathsf{vec}(Z)^T[\nabla^2 \mathcal{L}(U)]\mathsf{vec}(Z) \tag{144}$$

$$= \mathsf{vec}(UZ^T + ZU^T)[(\nabla^2 f)(UU^T)]\mathsf{vec}(UZ^T + ZU^T) + 2\langle(\nabla f)(UU^T), ZZ^T\rangle \tag{145}$$

$$\stackrel{(i)}{=} 2\langle(\nabla f)(UU^T), ZZ^T\rangle \tag{146}$$

$$= \langle(\nabla f)(UU^T), UWZ^T + ZW^T U^T\rangle \tag{147}$$

$$= \langle \nabla \mathcal{L}(U), WZ^T\rangle, \tag{148}$$

where $(i)$ uses the fact that $UZ^T + ZU^T = U(W^T + W)U^T = 0$ and the last equation uses the uses the gradient computations in Lemma 41.

Summing up eqs. (143) and (148), we get,

$$\mathsf{vec}(Z)^T[\nabla^2(\mathcal{L} + \lambda\mathcal{R}_\beta)(U)]\mathsf{vec}(Z)$$

$$\stackrel{(i)}{=} \langle \nabla(\mathcal{L} + \lambda\mathcal{R}_\beta)(U), WZ^T\rangle + \lambda\mathsf{Tr}((WL - LW) \cdot Z^T U) - \lambda\sum_{i=1}^{k} G(U)_{ii}\langle Ze_i, Ue_i\rangle^2 \tag{149}$$

Noting that $U$ is an $\epsilon$-approximate stationary point of $\mathcal{L} + \lambda\mathcal{R}_\beta$, by Cauchy–Bunyakovsky–Schwarz inequality,

$$\mathsf{vec}(Z)^T[\nabla^2(\mathcal{L} + \lambda\mathcal{R}_\beta)(U)]\mathsf{vec}(Z)$$

$$\leq \epsilon\|WZ^T\|_F + \lambda\mathsf{Tr}((WL - LW) \cdot Z^T U) - \lambda\sum_{i=1}^{k} G(U)_{ii}\langle Ze_i, Ue_i\rangle^2 \tag{150}$$

Now we are ready to choose $W$. Suppose $W$ is chosen as $(e_1 e_2^T - e_2 e_1^T)$. With this choice of $W$, we have that,

$$\|Z\|_F = \|UW\|_F = \sqrt{\|Ue_1\|_2^2 + \|Ue_2\|_2^2} \tag{151}$$

Likewise,

$$\|WZ^T\|_F = \|WW^T U^T\|_F \tag{152}$$

$$= \|U(e_1 e_1^T + e_2 e_2^T)\|_F \tag{153}$$

$$= \sqrt{\|Ue_1\|_2^2 + \|Ue_2\|_2^2} = \|Z\|_F. \tag{154}$$

Simplifying eq. (150), the last term can be evaluated to,

$$\sum_{i=1}^{k} G(U)_{ii} \langle Ze_i, Ue_i \rangle^2 = (G(U)_{11} + G(U)_{22}) \langle Ue_1, Ue_2 \rangle^2. \tag{155}$$

With this choice of parameters, eq. (150) simplifies to,

$$\mathsf{vec}(Z)^T [\nabla^2 (\mathcal{L} + \lambda \mathcal{R}_\beta)(U)] \mathsf{vec}(Z) \tag{156}$$

$$\leq \epsilon \|Z\|_F + \lambda \mathsf{Tr}((WL - LW) \cdot Z^T U) - \lambda (G(U)_{11} + G(U)_{22}) \langle Ue_1, Ue_2 \rangle^2 \tag{157}$$

Next we prove a lemma upper bounding the middle term in eq. (150).

**Lemma 26.** *The error term* $\lambda \, \mathsf{Tr}((WL - LW) \cdot Z^T U)$ *can be upper bounded by,*

$$\mathsf{Tr}((WL - LW) \cdot Z^T U) \leq 40\Delta^2 \max_{i \in \{1,2\}} \|Ue_i\|_2^2 \min_{i \in \{1,2\}} \|Ue_i\|_2. \tag{158}$$

The proof is deferred to Appendix B.2. Combining eq. (157) with Lemma 26,

$$\mathsf{vec}(Z)^T [\nabla^2 (\mathcal{L} + \lambda \mathcal{R}_\beta)(U)] \mathsf{vec}(Z) \leq \epsilon \|Z\|_F + 40\lambda \Delta^2 \|Z\|_F^2 \min_{i \in \{1,2\}} \|Ue_i\|_2$$

$$- \lambda (G(U)_{11} + G(U)_{22}) \langle Ue_1, Ue_2 \rangle^2 \tag{159}$$

Therefore, considering any approximate stationary point $U$ that satisfies,

$$\lambda (G(U)_{11} + G(U)_{22}) \frac{\langle Ue_1, Ue_2 \rangle^2}{\|Z\|_F^2} \geq \frac{\epsilon}{\|Z\|_F} + 40\lambda \Delta^2 \min_{i \in \{1,2\}} \|Ue_i\|_2 + \gamma \tag{C1}$$

the Hessian $\nabla^2 (\mathcal{L} + \lambda \mathcal{R}_\beta)(U)$ has an eigenvalue which is at most $-\gamma$. Note by a similar analysis as eq. (227),

$$G(U)_{11} + G(U)_{22} \geq \max_{i \in \{1,2\}} G(U)_{ii} \geq \max_{i \in \{1,2\}} \frac{\|Ue_i\|_2^2 + 4\beta}{(\|Ue_i\|_2^2 + \beta)^{5/2}} \tag{160}$$

$$\geq \max_{i \in \{1,2\}} \frac{1}{3\|Ue_i\|_2^3 + 12\beta^{3/2}} \tag{161}$$

$$\geq \max_{i \in \{1,2\}} \frac{1}{5\|Ue_i\|_2^3}. \tag{162}$$

where the second-to-last inequality uses the fact that $\min_{i \in \{1,2\}} \|Ue_i\|_2 \geq 2\sqrt{\beta}$. Therefore, a sufficient condition to guarantee eq. (C1) is,

$$\max_{i \in \{1,2\}} \frac{\lambda}{5\|Ue_i\|_2^3} \cdot \frac{\langle Ue_1, Ue_2 \rangle^2}{\|Z\|_F^2} \geq \frac{\epsilon}{\|Z\|_F} + 40\lambda \Delta^2 \min_{i \in \{1,2\}} \|Ue_i\|_2 + \gamma. \tag{163}$$

From eq. (138), we have that,

$$\frac{\langle Ue_1, Ue_2 \rangle^2}{\|Z\|_F^2} \leq \frac{\epsilon^2}{\lambda^2 \Delta^2} \tag{164}$$

When $\Delta$ is large, the columns are nearly orthogonal by eq. (164). When $\Delta$ is small, the barrier to setting up a negative eigenvalue of the Hessian of $\mathcal{L} + \lambda \mathcal{R}_\beta$ is small by eq. (163). In particular, under the "small-$\Delta$" condition,

$$\Delta \leq \frac{\epsilon}{4\lambda^{1/2} \cdot \min_{i \in \{1,2\}} \|Ue_i\|_2} \tag{165}$$

$\nabla^2 (\mathcal{L} + \lambda \mathcal{R}_\beta)(U)$ has a negative eigenvalue taking value at most $-\gamma$ under the sufficient condition,

$$\max_{i \in \{1,2\}} \frac{\lambda}{5\|Ue_i\|_2^3} \cdot \frac{\langle Ue_1, Ue_2 \rangle^2}{\|Z\|_F^2} \geq \frac{\epsilon}{\|Z\|_F} + 40\lambda \Delta^2 \min_{i \in \{1,2\}} \|Ue_i\|_2 + \gamma \tag{166}$$

$$\Longleftarrow \max_{i \in \{1,2\}} \frac{\lambda}{5\|Ue_i\|_2^3} \cdot \frac{\langle Ue_1, Ue_2 \rangle^2}{\|Z\|_F^2} \geq \frac{\epsilon}{\|Z\|_F} + \frac{40\epsilon \min_{i \in \{1,2\}} \|Ue_i\|_2}{16 \min_{i \in \{1,2\}} \|Ue_i\|_2^2} + \gamma \tag{167}$$

$$\Longleftarrow \frac{\langle Ue_1, Ue_2 \rangle^2}{\|Z\|_F^2} \geq \frac{18\epsilon \cdot \min_{i \in \{1,2\}} \|Ue_i\|_2^2}{\lambda} + \frac{5\gamma}{\lambda} \cdot \min_{i \in \{1,2\}} \|Ue_i\|_2^3 \tag{168}$$

From eq. (138), under the large-$\Delta$ condition, we have that,

$$\frac{\langle Ue_1, Ue_2 \rangle^2}{\|Z\|_F^2} \leq \frac{\epsilon^2}{\lambda^2} \frac{16\lambda}{\epsilon} \cdot \min_{i \in \{1,2\}} \|Ue_i\|_2^2 \leq \frac{16\epsilon}{\lambda} \cdot \min_{i \in \{1,2\}} \|Ue_i\|_2^2. \tag{169}$$

This completes the proof. □

Lemma 25 establishes the role of the regularizer - in making the large columns of $U$ nearly orthogonal at approximate second order stationary points. Next we show that at second order stationary points, $UU^T \approx U_\star U_\star^T$.

**Lemma 27.** *Consider an $\epsilon$-approximate first-order stationary point of $\mathcal{L} + \lambda\mathcal{R}_\beta$, $U$. If $\|UU^T - U_\star U_\star^T\|_F \geq 8\max\left\{\epsilon^{2/3}k^{1/6}, \lambda\sqrt{\frac{2k}{\beta}}\right\}$, then the Hessian of $\mathcal{L} + \lambda\mathcal{R}_\beta$ at $U$ has a large negative eigenvalue,*

$$\lambda_{\min}[\nabla^2(\mathcal{L} + \lambda\mathcal{R}_\beta)(U)]\textit{vec}(Z) \leq -\frac{1}{\sqrt{2k}}\|UU^T - U_\star U_\star^T\|_F. \tag{170}$$

*In other words, at an $(\epsilon, \gamma)$-approximate second order stationary point of $\mathcal{L} + \lambda\mathcal{R}_\beta$, $U$,*

$$\|UU^T - U_\star U_\star^T\|_F \lesssim \max\left\{\epsilon^{2/3}k^{1/6}, \lambda\sqrt{\frac{2k}{\beta}}, \gamma\sqrt{k}\right\} \tag{171}$$

*Proof.* In the rest of this proof we expand $U^\star$ to a $\mathbb{R}^{d\times k}$ matrix by appending with 0 columns. This allows us to define $R_\star \in \arg\min_{R:RR^T=R^T R=I}\|U - U_\star R\|_F^2$, breaking ties arbitrarily. Define $Z = U - U_\star R_\star$. With this choice of $Z$, we invoke 3 results from [36] (Lemmas 7, 40 and 41),

$$\|ZZ^T\|_F^2 \leq 2\|UU^T - U_\star U_\star^T\|_F^2. \tag{172}$$

And,

$$\|ZU^T\|_F^2 \leq \frac{1}{2\sqrt{2}-1}\|UU^T - U_\star U_\star^T\|_F^2. \tag{173}$$

And finally a bound on the Hessian,

$$\textsf{vec}(Z)^T[\nabla^2(\mathcal{L} + \lambda\mathcal{R}_\beta)(U)]\textsf{vec}(Z) \leq 2\|ZZ^T\|_F^2 - 6\|UU^T - U_\star U_\star^T\|_F^2 + 4\langle\nabla(\mathcal{L} + \lambda\mathcal{R}_\beta)(U), Z\rangle$$
$$+ \lambda\left[\textsf{vec}(Z)^T[\nabla^2\mathcal{R}_\beta(U)]\textsf{vec}(Z) - 4\lambda\langle\nabla\mathcal{R}_\beta(U), Z\rangle\right]. \tag{174}$$

Plugging eq. (172) into eq. (174), and using the fact that $U$ is an $\epsilon$-approximate first order stationary point,

$$\textsf{vec}(Z)^T[\nabla^2(\mathcal{L} + \lambda\mathcal{R}_\beta)(U)]\textsf{vec}(Z) \leq -2\|UU^T - U_\star U_\star^T\|_F^2 + 4\epsilon\|Z\|_F$$
$$+ \lambda\left[\textsf{vec}(Z)^T[\nabla^2\mathcal{R}_\beta(U)]\textsf{vec}(Z) - 4\langle\nabla\mathcal{R}_\beta(U), Z\rangle\right]. \tag{175}$$

In the next lemma we upper bound the last term.

**Lemma 28.** *The contribution from the regularizer in the last term of eq. (175) can be bounded by,*

$$\textit{vec}(Z)^T[\nabla^2\mathcal{R}_\beta(U)]\textit{vec}(Z) - 4\langle\mathcal{R}_\beta(U), \textit{vec}(Z)\rangle \leq 8\sqrt{\frac{2k}{\beta}}\|UU^T - U_\star U_\star^T\|_F. \tag{176}$$

The proof of this result is deferred to Appendix B.2

Finally, combining Lemma 28 with eq. (175),

$$\textsf{vec}(Z)^T[\nabla^2(\mathcal{L} + \lambda\mathcal{R}_\beta)(U)]\textsf{vec}(Z) \leq -2\|UU^T - U_\star U_\star^T\|_F^2 + 4\epsilon\|Z\|_F$$
$$+ 8\lambda\sqrt{\frac{2k}{\beta}}\|UU^T - U_\star U_\star^T\|_F \tag{177}$$

Yet again using the inequality $\|Z\|_F^4 \leq k\|ZZ^T\|_F^2 \leq 2k\|UU^T - U_\star U_\star^T\|_F^2$, this results in the bound,

$$\textsf{vec}(Z)^T[\nabla^2(\mathcal{L} + \lambda\mathcal{R}_\beta)(U)]\textsf{vec}(Z) \leq -2\|UU^T - U_\star U_\star^T\|_F^2 + 8\epsilon k^{1/4}\|UU^T - U_\star U_\star^T\|_F^{1/2}$$
$$+ 8\lambda\sqrt{\frac{2k}{\beta}}\|UU^T - U_\star U_\star^T\|_F \tag{178}$$

If $\|UU^T - U_\star U_\star^T\|_F \geq 8\max\left\{\epsilon^{2/3}k^{1/6}, \lambda\sqrt{\frac{2k}{\beta}}\right\}$, the RHS is upper bounded by $-\|UU^T - U_\star U_\star^T\|_F^2$. Finally, noting that $\|Z\|_F^4 \leq k\|ZZ^T\|_F^2 \leq 2k\|UU^T - U_\star U_\star^T\|_F^2$, we have that,

$$\lambda_{\min}[\nabla^2(\mathcal{L} + \lambda\mathcal{R}_\beta)(U)] \leq \frac{\mathsf{vec}(Z)^T[\nabla^2(\mathcal{L} + \lambda\mathcal{R}_\beta)(U)]\mathsf{vec}(Z)}{\|Z\|_F^2} \tag{179}$$

$$\leq -\frac{1}{\sqrt{2k}}\|UU^T - U_\star U_\star^T\|_F \tag{180}$$

$\square$

Let $V_r$ denote the matrix with columns as the non-zero eigenvectors of $U_\star U_\star^T$. In the next lemma, we show that for any stationary point $U$, all of its columns are almost entirely contained in the span of $V_r$, in that the angle between $Ue_i$ and its projection onto $V_r^\perp$ is almost $90°$. In other words, the columns of $U$ approximately lie in the correct subspace.

**Lemma 29.** *Consider an $\epsilon$-approximate first order stationary point of $\mathcal{L} + \lambda\mathcal{R}_\beta$, $U$, satisfying $\|U\|_{op} \leq 3$. Let $V_r$ denote the matrix with columns as the non-zero eigenvectors of $U_\star U_\star^T$. Then, assuming $\beta < 1$,*

$$\|V_r^\perp(V_r^\perp)^T U\|_F \leq 3\epsilon/\lambda. \tag{181}$$

*Proof.* By eq. (189) for $Z = V_r^\perp(V_r^\perp)^T U$, by the approximate stationarity of $U$,

$$2\langle UU^T - U_\star U_\star^T, UZ^T + ZU^T\rangle + \lambda\mathsf{Tr}(UD(U)Z^T) \leq \epsilon\|Z\|_F \tag{182}$$

The LHS is lower bounded by,

$$4\mathsf{Tr}(UU^T V_r^\perp(V_r^\perp)^T UU^T) + \lambda\mathsf{Tr}(UD(U)U^T V_r^\perp(V_r^\perp)^T) \tag{183}$$

$$= 4\|V_r^\perp(V_r^\perp)^T UU^T\|_F^2 + \lambda\sum_{i=1}^k (D(U))_{ii}\|V_r^\perp(V_r^\perp)^T Ue_i\|_2^2 \tag{184}$$

$$\overset{(i)}{\geq} \frac{\lambda}{3}\sum_{i=1}^k \|V_r^\perp(V_r^\perp)^T Ue_i\|_2^2 \tag{185}$$

$$= \frac{\lambda}{3}\|Z\|_F^2. \tag{186}$$

where $(i)$ uses the fact that since $\|U\|_{op} \leq 3$ and $\beta < 1$, $(D(U))_{ii} = \frac{2\beta + \|Ue_i\|_2^2}{(\|Ue_i\|_2^2 + \beta)^{3/2}} \geq \frac{2\beta + 9}{(9+\beta)^{3/2}} \geq 1/3$. Combining with eq. (182) completes the proof. $\square$

**Lemma 30.** *Consider an $\epsilon$-approximate first order stationary point $U$ of $\mathcal{L} + \lambda\mathcal{R}_\beta$. Let $V_r$ be as defined in Lemma 29. Let $S$ denote the set of columns $i \in [k]$ such that $\|Ue_i\|_2 \geq 2\sqrt{\beta}$. Then, for any $i \in S$,*

$$\frac{\|V_r^\perp(V_r^\perp)^T Ue_i\|_2}{\|Ue_i\|_2} \leq \frac{2\epsilon}{\lambda\beta^{1/4}}. \tag{187}$$

*Note that the LHS is the cosine of the angle between $Ue_i$ and its projection onto $V_r^\perp$, (or the sine of the angle between $Ue_i$ and its projection onto $V_r$). Likewise, for the remaining columns,*

$$\sum_{i\in[k]\setminus S} \|V_r^\perp(V_r^\perp)^T Ue_i\|_2^2 \leq \frac{2\epsilon^2\sqrt{\beta}}{\lambda^2}. \tag{188}$$

*Proof.* At an $\epsilon$-approximate first order stationary point the gradient is upper bounded in $L_2$-norm by $\epsilon$. From the gradient and Hessian computations in Lemma 40 and Lemma 43,

$$2\langle UU^T - U_\star U_\star^T, UZ^T + ZU^T\rangle + \lambda\mathsf{Tr}(D(U)Z^T U) \leq \epsilon\|Z\|_F \tag{189}$$

Choosing $Z = V_r^\perp (V_r^\perp)^T U$, and noting that $U_\star$ has rank $r$ and is orthogonal to $V_r^\perp$, the LHS simplifies as,

$$\epsilon \|Z\|_F \geq 4\mathsf{Tr}(UU^T V_r^\perp (V_r^\perp)^T UU^T) + \lambda \mathsf{Tr}(UD(U)U^T V_r^\perp (V_r^\perp)^T) \tag{190}$$

$$\geq \lambda \sum_{i=1}^k D(U)_{ii} \|V_r^\perp (V_r^\perp)^T U e_i\|_2^2 \tag{191}$$

$$\geq \lambda \sum_{i \in S} \frac{\|V_r^\perp (V_r^\perp)^T U e_i\|_2^2}{\|U e_i\|_2}, \tag{192}$$

where the last inequality uses the fact that for any column $i$ such that $\|U e_i\|_2 \geq 2\sqrt{\beta}$, since we have that $D(U)_{ii} \geq \frac{1}{\|U e_i\|_2}$ by Lemma 44. Putting everything together, we get that,

$$\lambda \sum_{i \in S} \|U e_i\|_2 \cdot \frac{\|V_r^\perp (V_r^\perp)^T U e_i\|_2^2}{\|U e_i\|_2^2} \leq \epsilon \|Z\|_F \tag{193}$$

Since $\|U e_i\|_2 \geq 2\sqrt{\beta}$, rearranging the terms around and finally upper bounding $\|Z\|_F$ using Lemma 29 completes the proof of eq. (187). To prove eq. (188), notice from eq. (191) that,

$$\epsilon \|Z\|_F \geq \lambda \sum_{i \in [k] \setminus S} D(U)_{ii} \|(V_r^\perp)(V_r^\perp)^T U e_i\|_2^2 \tag{194}$$

$$\geq \frac{\lambda}{2\sqrt{\beta}} \sum_{i \in [k] \setminus S} \|(V_r^\perp)(V_r^\perp)^T U e_i\|_2^2 \tag{195}$$

which uses the fact that when $\|U e_i\|_2 \leq 2\sqrt{\beta}$, then $D(U)_{ii} = \frac{\|U e_i\|_2^2 + 2\beta}{(\|U e_i\|_2^2 + \beta)^{3/2}} \geq \frac{1}{2\beta^{1/2}}$. Rearranging and substituting the bound on $\|Z\|_F$ from Lemma 29 completes the proof of eq. (188). $\square$

**Lemma 31.** *Consider any $\epsilon$-approximate first-order stationary point of $\mathcal{L} + \lambda \mathcal{R}_\beta$, $U$, and let $V_r$ be as defined in Lemma 29. Under the assumption that $\epsilon \leq \lambda/2$, for any column $i \in [k]$ such that $\|V_r V_r^T U e_i\|_2 \leq \|V_r^\perp (V_r^\perp)^T U e_i\|_2$, $\|U e_i\|_2 \leq 2\sqrt{\beta}$.*

*Proof.* Consider any $i$ such that $\|V_r V_r^T U e_i\|_2 \leq \|V_r^\perp (V_r^\perp)^T U e_i\|_2$. Therefore, from eq. (189), choosing $Z = V_r^\perp (V_r^\perp)^T U e_i e_i^T$,

$\epsilon \|V_r^\perp (V_r^\perp)^T U e_i\|_2$
$\geq 4\mathsf{Tr}(UU^T V_r^\perp (V_r^\perp)^T U e_i e_i^T U^T) + \lambda \mathsf{Tr}(UD(U)e_i e_i^T U^T V_r^\perp (V_r^\perp)^T)$
$= 4\|U^T V_r^\perp (V_r^\perp)^T U e_i\|_2^2 + 4\mathsf{Tr}(e_i^T U^T V_r V_r^T UU^T V_r^\perp (V_r^\perp)^T U e_i) + \lambda (D(U))_{ii} \|V_r^\perp (V_r^\perp)^T U e_i\|_2^2$
$\geq 4\|U^T V_r^\perp (V_r^\perp)^T U e_i\|_2^2 - 4\|U^T V_r V_r^T U e_i\|_2 \|U^T V_r^\perp (V_r^\perp)^T U e_i\|_2 + \lambda (D(U))_{ii} \|V_r^\perp (V_r^\perp)^T U e_i\|_2^2$
$\geq \lambda (D(U))_{ii} \|V_r^\perp (V_r^\perp)^T U e_i\|_2, \tag{196}$

where the last inequality uses the assumption that $\|V_r V_r^T U e_i\|_2 \leq \|V_r^\perp (V_r^\perp)^T U e_i\|_2$. Finally, observe the following bound on $(D(U))_{ii}$ computed using Lemma 44: for any column $i$ such that $\|U e_i\|_2 \geq 2\sqrt{\beta}$,

$$(D(U))_{ii} \geq \frac{1}{\|U e_i\|_2} = \frac{1}{\sqrt{\|V_r^\perp (V_r^\perp)^T U e_i\|_2^2 + \|V_r V_r^T U e_i\|_2^2}} \geq \frac{1}{2\|V_r^\perp (V_r^\perp)^T U e_i\|_2}. \tag{197}$$

Plugging this into eq. (196), we get the inequality,

$$\epsilon \|V_r^\perp (V_r^\perp)^T U e_i\|_2 \geq \frac{\lambda}{2} \|V_r^\perp (V_r^\perp)^T U e_i\|_2. \tag{198}$$

When $\epsilon \leq \frac{\lambda}{2}$, the only solution to inequality eq. (196) is $\|V_r^\perp (V_r^\perp)^T U e_i\|_2 = 0$. By the condition $\|V_r (V_r)^T U e_i\|_2 \leq \|V_r^\perp (V_r^\perp)^T U e_i\|_2$, this implies that $\|U e_i\|_2 = 0$. This contradicts the initial assumption that $\|U e_i\|_2 \geq 2\sqrt{\beta}$, thus concluding the proof of the second part of Lemma 31. $\square$

**Theorem 32.** *Consider any $(\epsilon, \gamma)$-approximate second-order stationary point of $\mathcal{L} + \lambda \mathcal{R}_\beta$ denoted $U$. Suppose $\|U\|_{op} \leq 3$. Construct $U_{prune}$ by following the pruning condition,*

$$\|Ue_i\|_2 \leq 2\sqrt{\beta} \implies U_{prune}e_i \leftarrow 0 \tag{199}$$

*Suppose $\epsilon \leq c\lambda\beta^{1/4}/r$ and $\epsilon + \gamma \leq c\lambda/r^2$ for a sufficiently small constant $c > 0$. Then,*

1. *$U_{prune}$ has at most $r$ non-zero columns.*

2. *Furthermore,*

$$\|U_{prune}U_{prune}^T - U_\star U_\star^T\|_F \lesssim \gamma\sqrt{k} + \lambda\sqrt{\frac{k}{\beta}} + \epsilon^{2/3}k^{1/6} + r\beta + \frac{2\epsilon^2\sqrt{\beta}}{\lambda^2}. \tag{200}$$

*Proof.* Recall that the columns of $U$ having $L_2$ norm at most $2\sqrt{\beta}$ are set to 0 in $U_{\text{prune}}$. Since $U$ is an $(\epsilon, \gamma)$-approximate second-order stationary point, the eigenvalues of the Hessian of $\mathcal{L} + \lambda \mathcal{R}_\beta$ at the point $U$ are all at least $-\gamma$.

In Lemma 25, by violation of the condition eq. (134), for every pair of column indices $i \neq j \in S$ where $S = \{i \in [k] : \|Ue_i\|_2 > 2\sqrt{\beta}\}$,

$$|C_{ij}| = \frac{|\langle Ue_i, Ue_j\rangle|}{\|Z_{ij}\|_2} \lesssim \sqrt{\frac{\epsilon + 3\gamma}{\lambda}} \min\{\|Ue_i\|_2, \|Ue_j\|_2\}. \tag{201}$$

where we use the assumption that the spectral norm $\|U\|_{op} \leq 3$, and therefore $\|Ue_i\|_2 \leq 3$ for all $i \in [k]$. Note that $\|Z_{ij}\|_2 = \sqrt{\|Ue_i\|_2^2 + \|Ue_j\|_2^2} \leq \sqrt{2}\max\{\|Ue_i\|_2, \|Ue_j\|_2\}$, and therefore,

$$|\langle Ue_i, Ue_j\rangle| \lesssim \sqrt{\frac{\epsilon + \gamma}{\lambda}}\|Ue_i\|_2\|Ue_j\|_2 \tag{202}$$

In other words, for each $i \neq j \in S$,

$$|\cos\theta_{ij}| \lesssim \sqrt{\frac{\epsilon + \gamma}{\lambda}} \implies \left|\theta_{ij} - \frac{\pi}{2}\right| \lesssim \sqrt{\frac{\epsilon + \gamma}{\lambda}} \tag{203}$$

Where $\theta_{ij}$ is the angle between $Ue_i$ and $Ue_j$ and the implication follows by assuming that $\epsilon, \gamma \leq c\lambda$ for a sufficiently small $c > 0$.

At any $\epsilon$-approximate first order stationary point $U$, by Lemma 30, for any $i$ such that $\|Ue_i\|_2 \geq 2\sqrt{\beta}$ the angle between $Ue_i$ and its projection onto $V_r$ is intuitively small. By eq. (203) we expect $Ue_i$ and $Ue_j$ to have an angle close to $90°$ between them. Therefore, we expect the projections of $Ue_i$ and $Ue_j$ onto $V_r$ to also have an angle close to $90°$ between them. Specifically, given vectors $v_1, v_2, v_3$ and $v_4$ such that $|\angle v_1, v_2| \leq \varepsilon_1$, $|\angle v_2, v_3 - 90°| \leq \varepsilon_2$ and $|\angle v_3, v_4| \leq \varepsilon_3$, then $|\angle v_1, v_4 - 90°| \leq \varepsilon_1 + \varepsilon_2 + \varepsilon_3$. This means that for any columns $Ue_i$ and $Ue_j$ such that $\|Ue_i\|_2$ and $\|Ue_j\|_2 \geq 2\sqrt{\beta}$,

$$\left|\theta_{ij}^{\text{proj}} - \frac{\pi}{2}\right| \lesssim \sqrt{\frac{\epsilon + \gamma}{\lambda}} + \frac{\epsilon}{\lambda\beta^{1/4}} \leq \frac{c_1}{r} \tag{204}$$

for some small absolute constant $0 \leq c_1 \leq 1/3$ and where $\theta_{ij}^{\text{proj}}$ is the angle between $V_rV_r^TUe_i$ and $V_rV_r^TUe_j$. This assumes that $\epsilon \leq c\lambda\beta^{1/4}/r$ and $\epsilon + \gamma \leq c\lambda/r^2$ for a sufficiently small constant $c > 0$. Since $\sin(x) \leq x$ for $x > 0$, this means that,

$$\forall i \neq j \in S, \quad \frac{|\langle V_rV_r^TUe_i, V_rV_r^TUe_j\rangle|}{\|V_rV_r^TUe_i\|_2\|V_rV_r^TUe_j\|_2} \leq \frac{c_1}{r}. \tag{205}$$

eq. (205) implies that the projections of the large columns of $U$ onto $V_r$ are approximately at $90°$ to each other. However, intuitively, since $V_r$ is an $r$ dimensional space, this means that at most $r$ of these projections can be large in norm. In fact it turns out that since the columns are sufficiently orthogonal to one another, *exactly* $r$ columns have a non-zero projection onto $V_r$.

**Lemma 33.** *At most $r$ columns of $U$ can have a non-zero projection onto $V_r$.*

The proof of this result is deferred to Appendix B.2.

What is the implication of Lemma 33? At most $r$ columns of $U$ (and therefore $U_{\text{prune}}$) have non-zero projections onto $V_r$. Any of remaining columns of $U$, say $i$, therefore satisfies the condition $0 = \|V_r V_r^T U e_i\|_2 \leq \|V_r^\perp (V_r^\perp)^T U e_i\|_2$. By implication of Lemma 31, the remaining columns of $U$ have $L_2$ norm at most $2\sqrt{\beta}$ assuming that $\epsilon \leq \lambda/2$. Thus these columns are pruned away in $U_{\text{prune}}$. Overall, this implies that at most $r$ columns of $U_{\text{prune}}$ are non-zero.

For the analysis of $\|U_{\text{prune}} U_{\text{prune}}^T - U_\star U_\star^T\|_F$, note that, the columns of $U$ with $L_2$ norm smaller than $2\sqrt{\beta}$ are set to 0 in $U_{\text{prune}}$. This implies that,

$$\|U_{\text{prune}} U_{\text{prune}}^T - U_\star U_\star^T\|_F - \|U U^T - U_\star U_\star^T\|_F$$

$$\leq \|U_{\text{prune}} U_{\text{prune}}^T - U U^T\|_F \tag{206}$$

$$\leq \left\| \sum_{i \in [k]} U e_i (U e_i)^T \mathbb{1}(\|U e_i\|_2 \leq 2\sqrt{\beta}) \right\|_F \tag{207}$$

$$\leq \sum_{i \in [k]} \left\| U e_i (U e_i)^T \mathbb{1}(\|U e_i\|_2 \leq 2\sqrt{\beta}) \right\|_F \tag{208}$$

$$\leq \sum_{i \in [k]} \|U e_i\|_2^2 \mathbb{1}(\|U e_i\|_2 \leq 2\sqrt{\beta}; \|V_r V_r^T U e_i\|_2 > 0) + \sum_{i \in [k]} \|V_r V_r^T U e_i\|_2^2 \mathbb{1}(\|U e_i\|_2 \leq 2\sqrt{\beta}) \tag{209}$$

$$\overset{(i)}{\leq} r\beta + \sum_{i \in [k]} \|V_r V_r^T U e_i\|_2^2 \mathbb{1}(\|U e_i\|_2 \leq 2\sqrt{\beta}) \tag{210}$$

$$\overset{(ii)}{\leq} r\beta + \frac{2\epsilon^2 \sqrt{\beta}}{\lambda^2} \tag{211}$$

where $(i)$ uses the fact that there are at most $r$ columns of $U$ having non-zero projections onto $V_r$, and $(ii)$ follows from eq. (188) of Lemma 30. Combining this inequality with the guarantee on $\|U U^T - U_\star U_\star^T\|_F$ in Lemma 27 completes the proof. $\qquad\square$

## B.2 Proof of Theorem 3

*Proof.* From Theorem 32, we have that,

$$\|U_{\text{prune}} U_{\text{prune}}^T - U_\star U_\star^T\|_F \lesssim \gamma\sqrt{k} + \lambda\sqrt{\frac{k}{\beta}} + \epsilon^{2/3} k^{1/6} + r\beta + \frac{2\epsilon^2 \sqrt{\beta}}{\lambda^2}. \tag{212}$$

Recall that the optimization and smoothing parameters are chosen as,

$$\beta = c_\beta \frac{(\sigma_r^\star)^2}{r} \tag{213}$$

$$\lambda = c_\lambda \frac{(\sigma_r^\star)^3}{\sqrt{kr}} \tag{214}$$

$$\gamma \leq c_\gamma \frac{(\sigma_r^\star)^3}{\sqrt{k} r^{5/2}} \tag{215}$$

$$\epsilon \leq c_\epsilon \frac{(\sigma_r^\star)^{7/2}}{\sqrt{k} r^{5/2}}. \tag{216}$$

For sufficiently small absolute constants $c_\beta, c_\lambda, c_\gamma, c_\epsilon > 0$. Under these choices, it is easily verified from eq. (212) that the conditions,

$$\epsilon \leq c\frac{\lambda \beta^{1/4}}{r}; \qquad \epsilon + \gamma \leq \frac{c\lambda}{r^2}. \tag{217}$$

which are conditions required by Theorem 32. The bound in eq. (212) results in,

$$\|U_{\text{prune}} U_{\text{prune}}^T - U_\star U_\star^T\|_F \leq \frac{1}{2}(\sigma_r^\star)^2. \tag{218}$$

Note also that $U_{\text{prune}}$ has at most $r$ non-zero columns. However, by eq. (218), this means that $U_{\text{prune}}$ has exactly $r$ columns, since if it had $r - 1$ non-zero columns or fewer, the error of the best solution must be at least $(\sigma_r^\star)^2$. $\qquad\square$

## B.3 Missing proofs in Appendix B

### B.3.1 Proof of Lemma 26

*Proof.* By definition of $W$ and $L$, $\text{Tr}((WL - LW) \cdot Z^T U)$, can be simplified to

$$\text{Tr}((WL - LW) \cdot Z^T U) = \Delta \text{Tr}((e_2 e_2^T - e_1 e_1^T) U^T U) \tag{219}$$

$$= \Delta(\|Ue_2\|_2^2 - \|Ue_1\|_2^2) \tag{220}$$

Note that $D(U)_{11} - D(U)_{22} = \Delta$ and $|\Delta| \leq \frac{\epsilon}{\lambda}$, so one can expect $\|Ue_1\|_2$ and $\|Ue_2\|_2$ to be close to each other. We bound $|\|Ue_1\|_2^2 - \|Ue_2\|_2^2|$ in two ways depending on the relative values of $\|Ue_1\|_2$ and $\|Ue_2\|_2$.

**Case 1: Column norms are similar.** $\max_{i \in \{1,2\}} \|Ue_i\|_2 \leq 2 \min_{i \in \{1,2\}} \|Ue_i\|_2$.

By definition of $\Delta$,

$$|\Delta| = \left| \frac{\|Ue_1\|_2^2 + 2\beta}{(\|Ue_1\|_2^2 + \beta)^{3/2}} - \frac{\|Ue_2\|_2^2 + 2\beta}{(\|Ue_2\|_2^2 + \beta)^{3/2}} \right| \tag{221}$$

$$= \left| \int_{\|Ue_2\|_2^2}^{\|Ue_1\|_2^2} \frac{3/2 \cdot x + 4\beta}{(x + \beta)^{5/2}} \mathrm{d}x \right| \tag{222}$$

$$\geq \min_{i \in \{1,2\}} \frac{3/2\|Ue_i\|_2^2 + 4\beta}{(\|Ue_i\|_2^2 + \beta)^{5/2}} |\|Ue_1\|_2^2 - \|Ue_2\|_2^2|. \tag{223}$$

Furthermore, note that for any $i \in \{1, 2\}$,

$$\frac{(\|Ue_i\|_2^2 + \beta)^{5/2}}{\frac{3}{2}\|Ue_i\|_2^2 + 4\beta} \leq \left( \frac{3}{2}\|Ue_i\|_2^2 + 4\beta \right)^{3/2} \tag{224}$$

$$\overset{(i)}{\leq} \sqrt{2} \left( \frac{3}{2}\|Ue_i\|_2^2 \right)^{3/2} + \sqrt{2} (4\beta)^{3/2} \tag{225}$$

$$\leq 3\|Ue_i\|_2^3 + 12\beta^{3/2} \tag{226}$$

$$\leq 5\|Ue_i\|_2^3 \tag{227}$$

where $(i)$ uses the convexity of $(\cdot)^{3/2}$ for positive arguments and applies Jensen's inequality, and the last inequality uses the fact that $\|Ue_1\|_2, \|Ue_2\|_2 \geq 2\sqrt{\beta}$. Overall, combining eq. (220) with eq. (223) and eq. (227) results in the bound,

$$\text{Tr}((WL - LW) \cdot Z^T U) \leq 5\Delta^2 \max_{i \in \{1,2\}} \|Ue_i\|_2^3 \tag{228}$$

$$\leq 40\Delta^2 \min_{i \in \{1,2\}} \|Ue_i\|_2^3 \tag{229}$$

where the last inequality uses the assumption that $\max_{i \in \{1,2\}} \|Ue_i\|_2 \leq 2 \min_{i \in \{1,2\}} \|Ue_i\|_2$.

**Case 2: Column norms are separated.** $\max_{i \in \{1,2\}} \|Ue_i\|_2 \geq 2 \min_{i \in \{1,2\}} \|Ue_i\|_2$.

WLOG suppose $\|Ue_1\|_2 \geq \|Ue_2\|_2$ which implies that $D(U)_{22} \geq D(U)_{11}$. The above assumption implies that $\|Ue_1\|_2 \geq 2\|Ue_2\|_2$. Then,

$$D(U)_{11} = \frac{\|Ue_1\|_2^2 + 2\beta}{(\|Ue_1\|_2^2 + \beta)^{3/2}} \tag{230}$$

$$\overset{(i)}{\leq} \frac{4\|Ue_2\|_2^2 + 2\beta}{(4\|Ue_2\|_2^2 + \beta)^{3/2}} \tag{231}$$

$$\overset{(ii)}{\leq} \frac{4}{3^{3/2}} \frac{\|Ue_2\|_2^2 + 2\beta}{(\|Ue_2\|_2^2 + \beta)^{3/2}} \tag{232}$$

$$\leq \frac{4}{5} D(U)_{22} \tag{233}$$

where $(i)$ uses the fact that $\frac{x+2\beta}{(x+\beta)^{3/2}}$ is a decreasing function in $x$ and $(ii)$ uses the fact that $\min\{\|Ue_1\|_2, \|Ue_2\|_2\} \geq 2\sqrt{\beta}$, and therefore, $4\|Ue_2\|_2^2 + \beta \geq 3(\|Ue_2\|_2^2 + \beta)$. Therefore,

$$|D(U)_{11} - D(U)_{22}| \geq \frac{1}{5}D(U)_{22} = \frac{1}{5} \frac{\|Ue_2\|_2^2 + 2\beta}{(\|Ue_2\|_2^2 + \beta)^{3/2}} \tag{234}$$

$$\overset{(i)}{\geq} \frac{1}{5} \frac{\|Ue_2\|_2^2}{\sqrt{2}\|Ue_2\|_2^3 + \sqrt{2}\beta^{3/2}} \tag{235}$$

$$\overset{(ii)}{\geq} \frac{1}{10} \frac{1}{\|Ue_2\|_2} \tag{236}$$

$$\geq \frac{1}{10} \frac{1}{\|Ue_2\|_2\|Ue_1\|_2^2} |\|Ue_1\|_2^2 - \|Ue_2\|_2^2| \tag{237}$$

where $(i)$ uses the convexity of $(\cdot)^{3/2}$ for positive arguments and an application of Jensen's inequality, while $(ii)$ uses the fact that $\min\{\|Ue_1\|_2, \|Ue_2\|_2\} \geq 2\sqrt{\beta}$. Since $|D(U)_{11} - D(U)_{22}| \leq |\Delta|$, this results in an upper bound on $|\|Ue_1\|_2^2 - \|Ue_2\|_2^2|$. Combining this with eq. (220) results in,

$$\mathsf{Tr}((WL - LW) \cdot Z^T U) \leq 10\Delta^2 \min_{i \in \{1,2\}} \|Ue_i\|_2 \max_{i \in \{1,2\}} \|Ue_i\|_2^2. \tag{238}$$

Combining eqs. (229) and (238) completes the proof of the lemma. $\qquad\square$

### B.3.2   Proof of Lemma 28

*Proof.* By definition,

$$\mathsf{vec}(Z)^T [\nabla^2 \mathcal{R}_\beta(U)]\mathsf{vec}(Z) - 4\langle \mathcal{R}_\beta(U), \mathsf{vec}(Z)\rangle \tag{239}$$

$$\leq \mathsf{Tr}(D(U)Z^T Z) - \sum_{i=1}^{k}(G(U))_{ii}\langle Ue_i, Ze_i\rangle^2 - 4\mathsf{Tr}(D(U)U^T Z) \tag{240}$$

$$\leq \mathsf{Tr}(D(U)Z^T Z) - 4\mathsf{Tr}(D(U)U^T Z) \tag{241}$$

$$\leq \frac{2}{\sqrt{\beta}}\|Z\|_F^2 + 4\|D(U)\|_F\|U^T Z\|_F \tag{242}$$

$$\overset{(i)}{\leq} \frac{2}{\sqrt{\beta}}\|Z\|_F^2 + 5\sqrt{\frac{k}{\beta}}\|UU^T - U_\star U_\star^T\|_F \tag{243}$$

$$\leq 8\sqrt{\frac{2k}{\beta}}\|UU^T - U_\star U_\star^T\|_F \tag{244}$$

where $(i)$ follows from the fact that for any $i \in [k]$, $|(D(U))_{ii}| \leq \max_{x \geq 0} \frac{x+2\beta}{(x+\beta)^{3/2}} = \frac{2}{\sqrt{\beta}}$, $(ii)$ uses eq. (173) and the fact that $\|Z\|_F^4 = \|Z^T\|_F^4 \leq \mathsf{rank}(Z^T)\|ZZ^T\|_F^2 \leq 2k\|UU^T - U_\star U_\star^T\|_F^2$ by eq. (172). $\qquad\square$

### B.3.3   Proof of Lemma 33

*Proof.* Let $S_{\max}$ denote the set of $r$ column indices $i \in [k]$ such that $\|V_r V_r^T U e_i\|_2$ are maximum and non-zero. If fewer than $r$ columns of $U$ have non-zero projections onto $V_r$ (i.e. fewer than $r$ candidates for $S_{\max}$ exist), then all the remaining columns of $U$ must have zero projections onto $V_r$ and we are done. Therefore, henceforth, we will assume that $|S_{\max}| = r$. Let $V_{\max} = \mathsf{span}(\{V_r V_r^T U e_i : i \in S_{\max}\})$, a subspace contained in $V_r$.

Define $M$ as the matrix $M = \left[ \frac{V_r V_r^T U e_i}{\|V_r V_r^T U e_i\|_2} : i \in S_{\max} \right]$. Let $V$ be the set of non-zero eigenvectors of $M M^T$. Note that the matrix $M^T M \in \mathbb{R}^{|S_{\max}| \times |S_{\max}|}$ has 1's on the diagonals, and has its off-diagonal entries upper bounded in absolute value by $c_1/r$ by eq. (205). Therefore, by the Gersgorin circle theorem, all eigenvalues of $M^T M$ and consequently all non-zero eigenvalues of $M M^T$ lie in the range $[1 - c_1, 1 + c_1]$. This means that $M \in \mathbb{R}^{d \times r}$ is full rank, and its columns are linearly independent.

This implies that $\mathsf{span}(V) = \mathsf{span}(V_r)$ and by the orthogonality condition, $V V^T = V_r V_r^T$. Therefore,

$$\|V_r V_r^T U e_i\|_2 = \|V V^T U e_i\|_2 \tag{245}$$

$$\leq \frac{1}{\lambda_{\min}(M)} \|M^T U e_i\|_2 \tag{246}$$

$$= \frac{1}{\lambda_{\min}(M)} \sqrt{\sum_{j \in S_{\max}} \frac{\langle V_r V_r^T U e_j, V_r V_r^T U e_i \rangle^2}{\|V_r V_r^T U e_j\|_2^2}} \tag{247}$$

$$\overset{(i)}{\leq} \frac{1}{\lambda_{\min}(M)} \sqrt{\sum_{j \in S_{\max}} \frac{c_1^2}{r^2} \|V_r V_r^T U e_i\|_2^2}, \tag{248}$$

where $(i)$ follows from eq. (204). Consequently by eq. (248), we have,

$$\|V_r V_r^T U e_i\|_2 \leq \frac{c_1}{1 - c_1} \|V_r V_r^T U e_i\|_2. \tag{249}$$

For $c_1 \leq 1/3$ this means that $\|V_r V_r^T U e_i\|_2 = 0$ for every $i \in [k] \setminus S_{\max}$. This proves the claim.  $\square$

# C Finite sample guarantees: Proof of Theorem 4

In this section, we provide guarantees on the approximate second order stationary points of the regularized loss $f_{\text{emp}}$ (eq. (14)) when the dataset is finite in size, and satisfies the RIP condition. We provide a formal proof of Theorem 4.

For completeness, in the finite sample setting, the empirical loss is defined as,

$$\mathcal{L}_{\text{emp}}(U) = \frac{1}{n} \sum_{i=1}^{n} (\langle UU^T - U_\star U_\star^T, A_i \rangle + \varepsilon_i)^2 \tag{250}$$

where $\varepsilon_i \overset{\text{i.i.d.}}{\sim} \mathcal{N}(0, \sigma^2)$ is the measurement noise. Without loss of generality we assume that the measurement matrices $\{A_i\}_{i=1}^n$ are symmetric, since the empirical loss $\mathcal{L}_{\text{emp}}$ is unchanged by the symmetrization $A_i \to \frac{A_i + A_i^T}{2}$. Note that the loss can be expanded as,

$$\|UU^T - U_\star U_\star^T\|_{\mathcal{H}}^2 + 2 \left\langle \frac{1}{n} \sum_{i=1}^{n} \varepsilon_i A_i, UU^T - U_\star U_\star^T \right\rangle + \frac{1}{n} \sum_{i=1}^{n} \varepsilon_i^2 \tag{251}$$

where $\langle X, Y \rangle_{\mathcal{H}} = \frac{1}{m} \sum_{i=1}^{m} \langle X, A_i \rangle \langle Y, A_i \rangle$ and $\|X\|_{\mathcal{H}}^2 = \langle X, X \rangle_{\mathcal{H}}$.

We will assume that the measurement matrices $\{A_i\}_{i=1}^n$ satisfy the RIP condition. At a high level this condition guarantees that $\langle \cdot, \cdot \rangle_{\mathcal{H}} \approx \langle \cdot, \cdot \rangle$ when the arguments are low rank matrices.

**Definition 34** (RIP condition). *A set of linear measurement matrices $A_1, \ldots, A_m$ in $\mathbb{R}^{d \times d}$ satisfies the $(k, \delta)$-restricted isometry property (RIP) if for any $d \times d$ matrix $X$ with rank at most $k$,*

$$(1 - \delta)\|X\|_F^2 \leq \frac{1}{m} \sum_{i=1}^{m} \langle A_i, X \rangle^2 \leq (1 + \delta)\|X\|_F^2 \tag{252}$$

The crucial property we use in this section is that the RIP condition on the measurements guarantees that $\langle X, Y \rangle_{\mathcal{H}} \approx \langle X, Y \rangle$. when $X$ and $Y$ are low rank matrices.

**Lemma 35.** *Let $\{A_i\}_{i=1}^m$ be a family of matrices in $\mathbb{R}^{d \times d}$ that satisfy $(k, \delta)$ RIP. Then for any pair of matrices $X, Y \in \mathbb{R}^{d \times d}$ with rank at most $k$, we have:*

$$|\langle X, Y \rangle_{\mathcal{H}} - \langle X, Y \rangle| \leq \delta \|X\|_F \|Y\|_F \tag{253}$$

**Lemma 36.** *Let $\{A_i\}_{i=1}^n$ be a set of matrices which satisfy the $(2k, \delta)$-RIP for some $\delta \leq 1/10$. Let $\varepsilon_i \overset{\text{i.i.d.}}{\sim} \mathcal{N}(0, \sigma^2)$. Consider any $\eta \in (0, 1)$. Then,*

$$\mathbb{P}\left( \left\| \frac{1}{n} \sum_i^n A_i \epsilon_i \right\|_{op} \geq 4\sigma \sqrt{\frac{d \log(d/\eta)}{n}} \right) \leq \eta. \tag{254}$$

*Proof.* Note that $\{A_i\}_{i=1}^n$ satisfies the $(2k, \delta)$-RIP for $\delta \leq 1/10$. That is, for any rank $\leq 2k$ matrix $X$,

$$\frac{9}{10}\|X\|_F^2 \leq \frac{1}{n} \sum_{i=1}^{n} \langle A_i, X \rangle^2 \leq \frac{11}{10}\|X\|_F^2. \tag{255}$$

Choosing $X = vu^T$ for any pair of vectors $u, v$, we get,

$$\forall u, v \in \mathbb{R}^d, \quad \frac{1}{n} \sum_{i=1}^{n} (u^T A_i v)^2 \leq \frac{11}{10}\|u\|_2^2 \|v\|_2^2. \tag{256}$$

This implies that for each $i \in [k]$, $\|A_i\|_{op} \leq \sqrt{2n}$. Furthermore, plugging in $v = e_1, \cdots, e_d$ and summing,

$$\frac{1}{n} \sum_{i=1}^{n} \|A_i^T u\|_2^2 \leq \frac{11}{10} d \|u\|_2^2 \tag{257}$$

This implies that $\left\|\frac{1}{n}\sum_{i=1}^n A_i A_i^T\right\|_{\mathrm{op}} = \lambda_{\max}\left(\frac{1}{n}\sum_{i=1}^n A_i A_i^T\right) \le 2d$. Hence, we obtain,

$$\left\|\sum_{i=1}^n \mathbb{E}\left((\epsilon_i A_i)(\epsilon_i A_i)^T\right)\right\|_{\mathrm{op}} \le \sigma^2 \left\|\sum_{i=1}^n A_i A_i^T\right\|_{\mathrm{op}} \le 2\sigma^2 dn. \tag{258}$$

By applying the matrix Bernstein inequality [54, 55], for any $t > 0$,

$$\mathbb{P}\left(\left\|\frac{1}{n}\sum_{i=1}^n A_i \epsilon_i\right\|_2 \ge t\right) \le d\cdot\exp\left(\frac{-3t^2 n^2}{12 dn\sigma^2 + 2\sigma\sqrt{2n}nt}\right) = d\cdot\exp\left(\frac{-3t^2 n}{12 d\sigma^2 + 4\sigma\sqrt{n}t}\right).$$

Choosing $t = 3\sigma\sqrt{\frac{d\log(d/\eta)}{n}}$, the RHS is upper bounded by $\eta$. This completes the proof. $\qquad\square$

By virtue of this result, we expect that for all $U$, with probability $\ge 1 - \eta$, the loss $\mathcal{L}$ in eq. (251) is $\approx \|UU^T - U_\star U_\star^T\|_F^2$ (up to additive constants) when $n$ is large. Indeed, by an application of Holder's inequality, $\mathsf{Tr}(AB) \le \|A\|_{\mathrm{op}}\|B\|_* \le \|A\|_{\mathrm{op}}\sqrt{\mathsf{rank}(B)}\|B\|_F$ and,

$$\left|\mathcal{L}_{\mathrm{emp}}(U) - \|UU^T - U_\star U_\star^T\|_{\mathcal{H}}^2 - \frac{1}{n}\sum_{i=1}^n \varepsilon_i^2\right| \lesssim \sigma\sqrt{\frac{kd\log(d/\eta)}{n}}\|UU^T - U_\star U_\star^T\|_F. \tag{259}$$

Likewise,

$$\left|\langle\nabla\mathcal{L}_{\mathrm{emp}}(U), Z\rangle - \langle\nabla\|UU^T - U_\star U_\star^T\|_{\mathcal{H}}^2, Z\rangle\right| \lesssim \sigma\sqrt{\frac{kd\log(d/\eta)}{n}}\|UZ^T + ZU^T\|_F \tag{260}$$

And finally,

$$\left|\mathsf{vec}(Z)^T[\nabla^2\mathcal{L}_{\mathrm{emp}}(U)]\mathsf{vec}(Z) - \mathsf{vec}(Z)^T[\nabla^2\|UU^T - U_\star U_\star^T\|_{\mathcal{H}}^2]\mathsf{vec}(Z)\right| \lesssim \sigma\sqrt{\frac{kd\log(d/\eta)}{n}}\|ZZ^T\|_F \tag{261}$$

where the hidden constants in each of these inequalities are at most 16.

In the sequel, we condition on the event that eqs. (259) to (261) hold, which occurs with probability $\ge 1 - \eta$.

**Lemmas 23 and 25:** The conclusion of these lemmas can still be applied since the finite sample loss function eq. (250) is still of the form $f(UU^T)$ for a doubly differentiable $f$.

**Lemma 27:** This lemma is slightly modified in the finite sample setting. The new result is provided below.

**Lemma 37** (Modified Lemma 27). *At an $(\epsilon, \gamma)$-approximate second order stationary point of $\mathcal{L} + \lambda\mathcal{R}_\beta$,*

$$\|UU^T - U_\star U_\star^T\|_F \lesssim \max\left\{\epsilon^{2/3}k^{1/6}, \lambda\sqrt{\frac{k}{\beta}} + \sigma\sqrt{\frac{kd\log(d/\eta)}{n}}, \sqrt{k}\gamma\right\}. \tag{262}$$

*Proof.* From [36, Lemma 7] and eq. (261) as in Lemma 27, for the choice $Z = U - U_\star R_\star$, with $R_\star \in \arg\min_{R:RR^T=R^TR=I}\|U - U_\star R\|_F^2$, we have the following bound,

$$\mathsf{vec}(Z)^T[\nabla^2\mathcal{L}_{\mathrm{emp}}(U)]\mathsf{vec}(Z) \le 2\|ZZ^T\|_{\mathcal{H}}^2 - 6\|UU^T - U_\star U_\star^T\|_{\mathcal{H}}^2 + 4\langle\nabla\mathcal{L}_{\mathrm{emp}}(U), Z\rangle$$
$$+ \lambda\left[\mathsf{vec}(Z)^T[\nabla^2\mathcal{R}_\beta(U)]\mathsf{vec}(Z) - 4\lambda\langle\nabla\mathcal{R}_\beta(U), Z\rangle\right]$$
$$+ 16\sigma\sqrt{\frac{kd\log(kd/\eta)}{n}}\|ZZ^T\|_F \tag{263}$$

Plugging eq. (172) into eq. (263) and using Lemma 28 and the fact that $\|\nabla\mathcal{L}_{\mathrm{emp}}(U)\|_F \le \epsilon$,

$$\mathsf{vec}(Z)^T[\nabla^2\mathcal{L}_{\mathrm{emp}}(U)]\mathsf{vec}(Z) \le 2\|ZZ^T\|_{\mathcal{H}}^2 - 6\|UU^T - U_\star U_\star^T\|_{\mathcal{H}}^2 + 4\epsilon\|Z\|_F$$
$$+ \left(8\lambda\sqrt{\frac{2k}{\beta}} + 32\sigma\sqrt{\frac{kd\log(d/\eta)}{n}}\right)\|UU^T - U_\star U_\star^T\|_F. \tag{264}$$

By the RIP condition on the measurements, $\|ZZ^T\|^2_{\mathcal{H}} \leq (1+\delta)\|ZZ^T\|^2_F$ and likewise, $\|UU^T - U_\star U_\star^T\|^2_{\mathcal{H}} \geq (1-\delta)\|UU^T - U_\star U_\star^T\|^2_F$. Therefore, assuming $\delta \leq 1/10$ and simplifying as done in eqs. (177) and (178), we get,

$$\mathsf{vec}(Z)^T[\nabla^2 \mathcal{L}_{\mathrm{emp}}(U)]\mathsf{vec}(Z) \leq -\|UU^T - U_\star U_\star^T\|^2_F + 8\epsilon k^{1/4}\|UU^T - U_\star U_\star^T\|^{1/2}_F$$

$$+ \left(8\lambda\sqrt{\frac{2k}{\beta}} + 32\sigma\sqrt{\frac{kd\log(d/\eta)}{n}}\right)\|UU^T - U_\star U_\star^T\|_F.$$

The rest of the analysis resembles the calculations from eq. (178) to eq. (180). $\qquad\square$

**Lemmas 29 to 31:** These lemmas are slightly changed in the finite sample setting. The new results essentially replace $\epsilon$ by a slightly larger value $\nu$ defined below,

$$\nu = \epsilon + \delta\|UU^T - U_\star U_\star^T\|_F + \sigma\sqrt{\frac{kd\log(d/\eta)}{n}}. \tag{265}$$

**Lemma 38.** *Consider an $\epsilon$-approximate first order stationary point $U$ satisfying $\|U\|_{op} \leq 3$. Let $V_r$ denote the top-$r$ eigenspace of $U_\star U_\star^T$. Then, we have that,*

$$\|V_r^\perp (V_r^\perp)^T U\|^3_F \lesssim \nu k. \tag{266}$$

*Let $S = \{i \in [k] : \|Ue_i\|_2 \geq 2\sqrt{\beta}\}$ be the set of large norm columns of $U$. For any column $i \in S$,*

$$\frac{\|V_r^\perp (V_r^\perp)^T Ue_i\|_2}{\|Ue_i\|_2} \leq \frac{2\nu}{\lambda\beta^{1/4}}. \tag{267}$$

*In contrast, for the remaining columns,*

$$\sum_{i \in [k]\setminus S} \|V_r^\perp (V_r^\perp)^T Ue_i\|^2_2 \leq \frac{2\nu^2\sqrt{\beta}}{\lambda^2}. \tag{268}$$

*Lastly, for any column $i \in [k]$ such that $\|V_r V_r^T Ue_i\|_2 \leq \|V_r^\perp (V_r^\perp)^T Ue_i\|_2$, assuming $\nu \leq \lambda/2$,*

$$\|Ue_i\|_2 \leq 2\sqrt{\beta}. \tag{269}$$

*Proof.* From eq. (260) and the gradient computations in Lemma 42 and Lemma 43, with $Z = V_r^\perp (V_r^\perp)^T U$, approximate first-order stationarity of $U$ implies that,

$$\langle UU^T - U_\star U_\star^T, UZ^T + ZU^T\rangle_{\mathcal{H}} + \lambda\mathsf{Tr}(D(U)Z^T U) - \sigma\sqrt{\frac{kd\log(d/\eta)}{n}}\|UZ^T + ZU^T\|_F \leq \epsilon\|Z\|_F \tag{270}$$

In the infinite sample analyses (Lemmas 29 and 30) the first term on the LHS is non-negative for this choice of $Z$. In the finite sample case, we instead lower bound as follows. Noting that $Z = V_r^\perp (V_r^\perp)^T U$ is rank $k - r$, this means that $UZ^T + ZU^T$ is a symmetric matrix of rank $\leq 2(k - r) \leq 2k$ by the subadditivity of the rank of matrices. Likewise, $UU^T - U_\star U_\star^T$ is of rank $\leq k + r \leq 2k$. Therefore, by Assumption 1,

$$\langle UU^T - U_\star U_\star^T, UZ^T + ZU^T\rangle_{\mathcal{H}} \tag{271}$$

$$\geq \langle UU^T - U_\star U_\star^T, UZ^T + ZU^T\rangle - \delta\|UU^T - U_\star U_\star^T\|_F\|UZ^T + ZU^T\|_F \tag{272}$$

$$\geq \langle UU^T - U_\star U_\star^T, UZ^T + ZU^T\rangle - 6\delta\|Z\|_F\|UU^T - U_\star U_\star^T\|_F \tag{273}$$

where the last inequality uses the assumption that $\|U\|_{\mathrm{op}} \leq 3$ which results in the bound,

$$\|ZU^T + UZ^T\|_F \leq 2\|U\|_{\mathrm{op}}\|Z\|_F \leq 6\|Z\|_F. \tag{274}$$

Therefore, in the finite sample case, instead of eq. (182) (and likewise, eq. (189)) we have,

$$\langle UU^T - U_\star U_\star^T, UZ^T + ZU^T\rangle + \lambda\mathsf{Tr}(D(U)Z^T U) \tag{275}$$

$$\lesssim \left(\epsilon + \delta\|UU^T - U_\star U_\star^T\|_F + \sigma\sqrt{\frac{kd\log(d/\eta)}{n}}\right)\|Z\|_F \tag{276}$$

$$= \nu\|Z\|_F. \tag{277}$$

The proof of Lemma 38 follows directly by replacing $\epsilon$ by $\nu$ everywhere in the remainder of the proofs of Lemmas 29 to 31. $\qquad\square$

Finally, we extend Theorem 32 to the finite sample setting and in combination with the choice of parameters present the main result in the finite sample setting, a restatement of Theorem 4.

**Theorem 39** (Main result in the finite sample setting). *Suppose the parameters $\epsilon, \gamma, \lambda$ and $\beta$ are chosen as in the population setting (Theorem 3) and consider the solution $U_{prune}$ returned by Algorithm 1. Suppose,*

$$n \geq c_n \frac{\sigma^2 k d \log(d/\eta)}{(\epsilon^\star)^2} = \Theta\left(\frac{\sigma^2 k^2 r^5 d \log(d/\eta)}{(\sigma_r^\star)^4}\right); \qquad \delta \leq c_\delta \frac{(\sigma_r^\star)^{3/2}}{\sqrt{k} r^{5/2}}. \tag{278}$$

*for appropriate absolute constants $c_n, c_\delta > 0$. Then, with probability $\geq 1 - \eta$,*

1. *$U_{prune}$ has exactly $r$ non-zero columns.*

2. *Furthermore,*

$$\|U_{prune} U_{prune}^T - U_\star U_\star^T\|_F \leq \frac{1}{2}(\sigma_r^\star)^2. \tag{279}$$

*In other words, Algorithm 1 results in a solution $U_{prune}$ having exactly $r$ non-zero columns, and also serving as a spectral initialization.*

*Proof.* Up until eq. (203), the proof is unchanged. Plugging in the new upper bound on the cosine of the angle between $V_r^\perp (V_r^\perp)^T U e_i$ and $U e_i$ in eq. (267) in Lemma 38, we get,

$$\left|\theta_{ij}^{\text{proj}} - \frac{\pi}{2}\right| \lesssim \sqrt{\frac{\epsilon + \gamma}{\lambda}} + \frac{\nu}{\lambda \beta^{1/4}}. \tag{280}$$

Assume for a sufficiently small constant $c_2 > 0$, $\epsilon + \gamma \leq c_2 \lambda / r^2$. Then the first term on the RHS is upper bounded by $\frac{c_1}{2r}$ for a sufficiently small $c_1 \leq 1/3$. Recall the definition,

$$\nu = \epsilon + 2\delta \|UU^T - U_\star U_\star^T\|_F + \sigma \sqrt{\frac{k d \log(d/\eta)}{n}}. \tag{281}$$

And furthermore, by Lemma 37,

$$\|UU^T - U_\star U_\star^T\|_F \lesssim \max\left\{\epsilon^{2/3} k^{1/6}, \lambda \sqrt{\frac{k}{\beta}} + \sigma \sqrt{\frac{k d \log(d/\eta)}{n}}, \sqrt{k}\gamma\right\} \tag{282}$$

Observe that if everywhere $\nu$ was replaced by $\epsilon$ (or even a constant factor approximation to it), the proof of Theorem 32 essentially carries over unchanged. Let $\epsilon^\star = c_\epsilon \frac{(\sigma_r^\star)^{7/2}}{\sqrt{k} r^{5/2}}$ be the choice of $\epsilon$ in the population setting. This is the "target" value of $\epsilon$ in the empirical setting and we show that as long as $\delta$ is sufficiently small ($O(1/\sqrt{k})$) and $n$ is sufficiently large ($\widetilde{\Omega}(dk^2)$) $\nu$ is at most $3\epsilon^\star$.

In particular, with the same choice of parameters ($\epsilon, \gamma, \beta$ and $\lambda$) as in the population setting, suppose that,

$$n \geq c_n \frac{\sigma^2 k d \log(d/\eta)}{(\epsilon^\star)^2} = \Theta\left(\frac{\sigma^2 k^2 r^5 d \log(d/\eta)}{(\sigma_r^\star)^4}\right); \qquad \delta \leq c_\delta \frac{(\sigma_r^\star)^{3/2}}{\sqrt{k} r^{5/2}}. \tag{283}$$

Note that by choice of the parameters $\beta, \lambda, \gamma, \epsilon$, observe that,

$$2\delta \|UU^T - Y_\star U_\star^T\|_F \lesssim 2\delta \max\left\{\epsilon^{2/3} k^{1/6}, \lambda \sqrt{\frac{k}{\beta}}, \sqrt{k}\gamma\right\} \leq \epsilon^\star. \tag{284}$$

Therefore, combining eqs. (283) and (284) with the definition of $\nu$ in eq. (281), we have that $\nu \in [\epsilon^\star, 3\epsilon^\star]$. With this choice, since $\epsilon = \epsilon^\star$ and $\nu$ are within constant multiples of each other, the rest of the proof of Theorem 32 in the population setting carries over. Subsequently plugging in the choice of $\lambda, \beta, \epsilon$ and $\gamma$ results in the following two statements:

1. $U_{\text{prune}}$ has at most $r$ non-zero columns.

2. $\|U_{\text{prune}} U_{\text{prune}} - U_\star U_\star^T\|_F \leq \frac{(\sigma_r^\star)^2}{2}$.

However under these two results, $U_{\text{prune}}$ must have exactly $r$ non-zero columns; if it had strictly fewer than $r$ non-zero columns, then it is impossible to satisfy $\|U_{\text{prune}}U_{\text{prune}} - U_\star U_\star^T\|_F < (\sigma_r^\star)^2$. This completes the proof of the theorem.

$\square$

# D   Efficiently finding approximate second order stationary points

In this section we discuss efficiently finding second order stationary points of the loss $\mathcal{L} + \lambda\mathcal{R}_\beta$. We establish smoothness conditions on $\mathcal{L}$ and $\mathcal{R}_\beta$ and show that running gradient descent with sufficiently bounded perturbations satisfies the property that all iterates are bounded in that $\|U_t\|_2 \leq 3$, as long as the algorithm is initialized within this ball.

**Perturbed gradient descent:**   We consider a first order method having the following update rule: for all $t \geq 0$,

$$U_{t+1} \leftarrow U_t - \alpha(\nabla(\mathcal{L} + \lambda\mathcal{R}_\beta)(U_t) + P_t) \tag{285}$$

where $P_t$ is a perturbation term, which for example could be the stochastic noise present in SGD. Over the course of running the update rule eq. (18), we show that the iterates $\|U_t\|_2$ remains bounded under mild conditions.

## D.1   Proof of Theorem 7

In this section we prove Theorem 7. Below we first show gradient Lipschitzness on the domain $\{U : \|U\|_{\text{op}} \leq 3\}$. Observe that,

$$\|\nabla\mathcal{L}_{\text{pop}}(U) - \nabla\mathcal{L}_{\text{pop}}(V)\|_F \tag{286}$$

$$= \|(UU^T - U_\star U_\star^T)U - (VV^T - U_\star U_\star^T)V\|_F \tag{287}$$

$$= \|UU^T(U - V) + (U(U - V)^T + (U - V)V^T)V - U_\star U_\star^T(U - V)\|_F \tag{288}$$

$$\overset{(i)}{\leq} \|U\|_{\text{op}}^2\|U - V\|_F + \|U\|_{\text{op}}\|U - V\|_F\|V\|_{\text{op}} + \|U - V\|_F\|V\|_{\text{op}}^2 + \|U_\star U_\star^T\|_{\text{op}}\|U - V\|_F \tag{289}$$

$$\lesssim \|U - V\|_F, \tag{290}$$

where $(i)$ repeatedly uses the bound $\|AB\|_F \leq \|A\|_{\text{op}}\|B\|_F$ and the last inequality follows from the fact that $\|U\|_{\text{op}} \leq 3$ and $\|U_\star\|_{\text{op}} = 1$. On the other hand, by the gradient computations in Lemma 43,

$$\|\nabla^2\mathcal{R}_\beta(U)\|_{\text{op}} \leq \sup_{Z:\|Z\|_F \leq 1} \text{vec}(Z)[\nabla^2\mathcal{R}_\beta(U)]\text{vec}(Z) \tag{291}$$

$$\leq \sup_{Z:\|Z\|_F \leq 1} \langle D(U), Z^T Z \rangle \tag{292}$$

$$= \sup_{Z:\sum_{i=1}^k \|Ze_i\|_2^2 \leq 1} \sum_{i=1}^k D(U)_{ii}\|Ze_i\|_2^2 \tag{293}$$

$$= \max_{i\in[k]} D(U)_{ii} \leq \frac{2}{\sqrt{\beta}}. \tag{294}$$

where the last inequality uses the fact that $\frac{x+2\beta}{(x+\beta)^{3/2}} \leq \frac{2}{\sqrt{\beta}}$. This implies that,

$$\|\nabla\mathcal{R}_\beta(U) - \nabla\mathcal{R}_\beta(V)\|_{\text{op}} \leq \frac{2}{\sqrt{\beta}}\|U - V\|_F. \tag{295}$$

Combining eqs. (290) and (294), by triangle inequality,

$$\|\nabla f_{\text{pop}}(U) - \nabla f_{\text{pop}}(V)\|_F \lesssim \|U - V\|_F \tag{296}$$

where the assumption $\lambda/\sqrt{\beta} \leq 1$ is invoked.

Next we prove Hessian Lipschitzness of $f_{\text{pop}}$. By [56, Theorem 3], the Hessian of $\mathcal{L}_{\text{pop}}$ satisfies the condition,

$$\|\nabla^2 \mathcal{L}_{\text{pop}}(U) - \nabla^2 \mathcal{L}_{\text{pop}}(V)\|_{\text{op}} \lesssim \|U - V\|_F. \tag{297}$$

Although the analysis in [56] is provided for the exactly specified case ($k = r$), it carries over unchanged to the overparameterized case as well. On the other hand, for any $Z, Z_U \in \mathbb{R}^{d \times k}$ such that $\|Z\|_F, \|Z_U\|_F \leq 1$, the rate of change of the Hessian at $U$ evaluated along the direction $Z$,

$$\lim_{t \to 0} \text{vec}(Z)^T \left[ \frac{\nabla^2 \mathcal{R}_\beta(U) - \nabla^2 \mathcal{R}_\beta(U + tZ_U)}{t} \right] \text{vec}(Z) \tag{298}$$

$$= \lim_{t \to 0} \frac{\langle D(U) - D(U + tZ_U), Z^T Z \rangle}{t} \tag{299}$$

$$+ \frac{\sum_{i=1}^{k} G(U + tZ_U)_{ii} \langle Ze_i, (U + tZ_U)e_i \rangle^2 - G(U)_{ii} \langle Ze_i, Ue_i \rangle^2}{t}. \tag{300}$$

The first term of the RHS of eq. (300) can be bounded as,

$$\lim_{t \to 0} \frac{\langle D(U) - D(U + tZ_U), Z^T Z \rangle}{t} = -\sum_{i=1}^{k} \frac{\|Ue_i\|_2^2 + 4\beta}{(\|Ue_i\|_2^2 + \beta)^{5/2}} \langle Ue_i, Z_U e_i \rangle \cdot \|Ze_i\|_2^2 \tag{301}$$

$$\overset{(i)}{\leq} \max_{i \in [k]} \left| \frac{\|Ue_i\|_2^2 + 4\beta}{(\|Ue_i\|_2^2 + \beta)^{5/2}} \langle Ue_i, Z_U e_i \rangle \right| \sum_{i=1}^{k} \|Ze_i\|_2^2 \tag{302}$$

$$\overset{(ii)}{\leq} \max_{i \in [k]} \frac{\|Ue_i\|_2^2 + 4\beta}{(\|Ue_i\|_2^2 + \beta)^{5/2}} \|Ue_i\|_2 \tag{303}$$

$$\lesssim \frac{1}{\beta}, \tag{304}$$

where $(i)$ follows by Holder inequality, and $(ii)$ uses the fact that $\|Z\|_F, \|Z_U\|_F \leq 1$. The last inequality uses the fact that $\min_{x \geq 0} \frac{\sqrt{x}(x + 4\beta)}{(x + \beta)^{5/2}} \lesssim \frac{1}{\beta}$.

On the other hand, by the chain rule, the second term on the RHS of eq. (300) can be computed in two parts, the first being,

$$\lim_{t \to 0} \frac{(G(U + tZ_U)_{ii} - G(U)_{ii}) \langle Ze_i, Ue_i \rangle^2}{t} = -\sum_{i=1}^{k} \frac{3(\|Ue_i\|_2^2 + 6\beta)}{(\|Ue_i\|_2^2 + \beta)^{7/2}} \langle Ue_i, Z_U e_i \rangle \langle Ze_i, Ue_i \rangle^2$$

$$\leq \sum_{i=1}^{k} H(U)_{ii} \|Ue_i\|_2 \langle Ze_i, Ue_i \rangle^2 \tag{305}$$

which uses the fact that $\|Z_U\|_F \leq 1$ $H(U)_{ii} = \frac{3(\|Ue_i\|_2^2 + 6\beta)\|Ue_i\|_2}{(\|Ue_i\|_2^2 + \beta)^{7/2}}$. Note that for any $c \geq 0$, the optimization problem which takes the form $\max_{v: \|v\|_2^2 \leq c} \langle v, Ue_i \rangle$ is maximized when $v \propto Ue_i$. Therefore, fixing the solution to the remaining coordinates and optimizing over $Z$ for a single $i$, one may substitute $Ze_i = x_i \cdot Ue_i / \|Ue_i\|_2$ for some $x_i \in \mathbb{R}$ and bound eq. (305) as,

$$\lim_{t \to 0} \frac{(G(U + tZ_U)_{ii} - G(U)_{ii}) \langle Ze_i, Ue_i \rangle^2}{t} \leq \max_{\sum_{i=1}^{k} x_i^2 \leq 1} \sum_{i=1}^{k} H(U)_{ii} x_i^2 \|Ue_i\|_2^3 \tag{306}$$

$$= \max_{i \in [k]} H(U)_{ii} \|Ue_i\|_2^3 \tag{307}$$

$$\lesssim \frac{1}{\beta}, \tag{308}$$

where the last inequality uses the fact that $\min_{x \geq 0} \frac{3\sqrt{x^3}(x+6\beta)}{(x+\beta)^{7/2}} \lesssim \frac{1}{\beta}$. The second part of the second term on the RHS of eq. (300) is,

$$\lim_{t \to 0} \sum_{i=1}^{k} \frac{G(U)_{ii} \left( \langle Ze_i, (U + tZ_U)e_i \rangle^2 - \langle Ze_i, Ue_i \rangle^2 \right)}{t} \tag{309}$$

$$= 2 \sum_{i=1}^{k} G(U)_{ii} \langle Ze_i, Ue_i \rangle \langle Ze_i, Z_U e_i \rangle \tag{310}$$

$$\lesssim \sum_{i=1}^{k} G(U)_{ii} \|Ue_i\|_F \langle Ze_i, Z_U e_i \rangle \tag{311}$$

$$\lesssim \sum_{i=1}^{k} G(U)_{ii} \|Ue_i\|_F \|Ze_i\|_2 \|Z_U e_i\|_2 \tag{312}$$

$$\leq \sqrt{\sum_{i=1}^{k} (G(U)_{ii})^2 \|Ue_i\|_F^2 \|Ze_i\|_2^2} \tag{313}$$

$$\leq \max_{i \in [k]} G(U)_{ii} \|Ue_i\|_F \tag{314}$$

$$\lesssim \frac{1}{\beta} \tag{315}$$

where in $(i)$ we use the fact that $x_i = \langle Ze_i, Z_U e_i \rangle$ satisfies $|\sum_i x_i| = |\langle Z, Z_U \rangle| \leq \|Z\|_F \|Z_U\|_F \leq 1$. The last inequality uses the fact that $\max_{x \geq 0} \frac{\sqrt{x}(x+4\beta)}{(x+\beta)^{5/2}} \lesssim 1/\beta$. Combining eqs. (304), (308) and (315) in eq. (300) results in the bound,

$$\max_{Z,Z_U : \|Z\|_F, \|Z_U\|_F \leq 1} \lim_{t \to 0} \frac{\mathsf{vec}(Z)^T [\nabla^2 \mathcal{R}_\beta(U) - \mathcal{R}_\beta(U + tZ_U)] \mathsf{vec}(Z)}{t} \lesssim \frac{1}{\beta} \tag{316}$$

By choosing $Z_U = (U - V)/\|U - V\|_F$, this implies that,

$$\max_{Z : \|Z\|_F \leq 1} \mathsf{vec}(Z)^T [\nabla^2 \mathcal{R}_\beta(U) - \nabla^2 \mathcal{R}_\beta(V)] \mathsf{vec}(Z) \lesssim \frac{1}{\beta} \|U - V\|_F. \tag{317}$$

This implies that,

$$\|\nabla^2 \mathcal{R}_\beta(U) - \nabla^2 \mathcal{R}_\beta(V)\|_{\mathrm{op}} \lesssim \frac{1}{\beta} \|U - V\|_F. \tag{318}$$

Combining with eq. (297) by triangle inequality and noting the assumption that $\lambda \leq \beta$ results in the equation,

$$\|\nabla^2 f_{\mathrm{pop}}(U) - f_{\mathrm{pop}}(V)\|_{\mathrm{op}} \lesssim \|U - V\|_F. \tag{319}$$

This completes the proof.

### D.2  Proof of Theorem 8

In this section we prove Theorem 8 formally. By the gradient computations in Lemmas 40 and 43 and triangle inequality,

$$\|U_{t+1}\|_{\mathrm{op}} \leq \|(I - \alpha U_t U_t^T)U_t\|_{\mathrm{op}} + \alpha \|U_\star U_\star^T U_t\|_{\mathrm{op}} + \alpha\lambda \|D(U_t)\|_{\mathrm{op}} \|U_t\|_{\mathrm{op}} + \alpha \tag{320}$$

$$\leq \|(I - \alpha U_t U_t^T)U_t\|_{\mathrm{op}} + \alpha \|U_\star U_\star^T U_t\|_{\mathrm{op}} + \alpha \frac{\lambda}{\sqrt{\beta}} \|U_t\|_{\mathrm{op}} + \alpha \tag{321}$$

where the last inequality follows from the fact that $D(U_t)$ is a diagonal matrix with largest entry upper bounded by $2/\sqrt{\beta}$. Note that $\|(I - \alpha U_t U_t^T)U_t\|_{\mathrm{op}} = \max_i \sigma_i(1 - \alpha\sigma_i^2)$ where $\{\sigma_i\}_{i=1}^{k}$ denotes the singular values of $U_t$. For $x \in [0, 1/2\alpha]$, $x(1 - \alpha x^2)$ is an increasing function. Therefore, assuming $\|U_t\|_{\mathrm{op}} \in [0, 1/2\alpha]$,

$$\|(I - \alpha U_t U_t^T)U_t\|_{\mathrm{op}} \leq \|U_t\|_{\mathrm{op}}(1 - \alpha\|U_t\|_{\mathrm{op}}^2). \tag{322}$$

Combining everything together,

$$\|U_{t+1}\|_{\mathrm{op}} \le \|U_t\|_{\mathrm{op}}(1 - \alpha\|U_t\|_{\mathrm{op}}^2) + \alpha\|U_t\|_{\mathrm{op}} + \alpha\frac{\lambda}{\sqrt{\beta}}\|U_t\|_{\mathrm{op}} + \alpha \qquad (323)$$

$$\le \|U_t\|_{\mathrm{op}}(1 - \alpha\|U_t\|_{\mathrm{op}}^2) + 2\alpha\|U_t\|_{\mathrm{op}} + \alpha \qquad (324)$$

where the last inequality assumes $\lambda/\sqrt{\beta} \le 1$. Depending on whether $\|U_t\|_{\mathrm{op}} \ge 2$ or $\|U_t\|_{\mathrm{op}} \le 2$, and recalling that $\alpha \le 1/8$, we may derive two bounds from the above,

$$0 \le \|U_t\|_{\mathrm{op}} \le 2 \implies \|U_{t+1}\|_{\mathrm{op}} \le \|U_t\|_{\mathrm{op}}(1 + 2\alpha) + \alpha \qquad (325)$$

$$2 \le \|U_t\|_{\mathrm{op}} \le 4 \implies \|U_{t+1}\|_{\mathrm{op}} \le \|U_t\|_{\mathrm{op}}(1 - \alpha) \qquad (326)$$

Suppose the initial iterate satisfied $\|U_0\|_{\mathrm{op}} \le 2$, and let $t_0$ be any iteration where $\|U_{t_0}\|_{\mathrm{op}} \le 2$, but $\|U_{t_0+1}\|_{\mathrm{op}} \ge 2$. Then by eq. (325), $\|U_{t_0+1}\|_{\mathrm{op}} \le 3$. However, in the subsequent iterations, $\|U_t\|_{\mathrm{op}}$ must decrease until it is no larger than 2, by virtue of eq. (326). This implies the statement of the theorem.

## E  Gradient and Hessian computations

### E.1  Population mean square error

**Lemma 40.** *Consider the function $\mathcal{L}_{pop}(U) = \|UU^T - U_\star U_\star^T\|_F^2$. Then,*

1. *The gradient of $\mathcal{L}$ satisfies $\langle\nabla\mathcal{L}_{pop}(U), Z\rangle = 2\langle(UU^T - U_\star U_\star^T), UZ^T + ZU^T\rangle$.*

2. *For any $Z \in \mathbb{R}^{d\times k}$,*

$$\textit{vec}(Z)^T[\nabla^2\mathcal{L}_{pop}(U)]\textit{vec}(Z) = 4\langle Z, (UU^T - U_\star U_\star^T)Z\rangle + 2\|UZ^T + ZU^T\|_F^2. \quad (327)$$

*Proof.* The first part is proved in [57, eq. (59)]. The second part is proved shortly after in [57, eq. (61)]. $\qquad\square$

**Lemma 41.** *Consider the function $\mathcal{L}(U) = f(UU^T) : \mathbb{R}^{d\times k} \to \mathbb{R}$ for some doubly differentiable function $f : \mathbb{R}^{d\times d} \to \mathbb{R}$. Then,*

1. *For any $Z \in \mathbb{R}^{d\times k}$, $\langle\nabla\mathcal{L}(U), Z\rangle = \langle(\nabla f)(UU^T), UZ^T + ZU^T\rangle$.*

2. *For any $Z \in \mathbb{R}^{d\times d}$,*

$$\textit{vec}(Z)^T[\nabla^2\mathcal{L}(U)]\textit{vec}(Z)$$
$$= \textit{vec}(UZ^T + ZU^T)^T[(\nabla^2 f)(UU^T)]\textit{vec}(UZ^T + ZU^T) + 2\langle(\nabla f)(UU^T), ZZ^T\rangle \quad (328)$$

*Proof.* This result is straightforward to prove by direct computation. $\qquad\square$

### E.2  Empirical mean square error

**Lemma 42.** *Consider the loss $f(U) = \|UU^T - U_\star U_\star\|_{\mathcal{H}}^2$ where $\|\cdot\|_{\mathcal{H}}$ is defined in eq. (251). Then,*

1. *The gradient of $f$ satisfies, $\langle\nabla f(U), Z\rangle = 2\langle UU^T - U_\star U_\star^T, UZ^T + ZU^T\rangle_{\mathcal{H}}$.*

2. *For any $Z \in \mathbb{R}^{d\times d}$,*

$$\textit{vec}(Z)^T[\nabla^2 f(U)]\textit{vec}(Z) = 4\langle UU^T - U_\star U_\star^T, ZZ^T\rangle_{\mathcal{H}} + 2\|UZ^T + ZU^T\|_{\mathcal{H}}^2. \quad (329)$$

*Proof.* These results are proved in [36, eq. (18)]. $\qquad\square$

### E.3 Regularization $\mathcal{R}_\beta$

**Lemma 43.** *Consider the function $\mathcal{R}_\beta : \mathbb{R}^{d \times m} \to \mathbb{R}$ defined as $\mathcal{R}_\beta(U) = \sum_{i=1}^m L_2^\beta(Ue_i)$. Define,*

$$D(U) = \text{\textbf{diag}}\left(\left\{ \frac{(\|Ue_i\|_2^2 + 2\beta)}{(\|Ue_i\|_2^2 + \beta)^{3/2}} : i \in [m] \right\}\right), \text{ and,} \tag{330}$$

$$G(U) = \text{\textbf{diag}}\left(\left\{ \frac{(\|Ue_i\|_2^2 + 4\beta)}{(\|Ue_i\|_2^2 + \beta)^{5/2}} : i \in [m] \right\}\right). \tag{331}$$

*For any $Z \in \mathbb{R}^{d \times m}$,*

1. *The gradient of $\mathcal{R}_\beta$ is, $\nabla \mathcal{R}_\beta(U) = UD(U)$.*

2. *The Hessian of $\mathcal{R}_\beta$ satisfies,*

$$\text{\textbf{vec}}(Z)^T [\nabla^2 \mathcal{R}_\beta(U)] \text{\textbf{vec}}(Z) = \langle D(U), Z^T Z \rangle - \sum_{i=1}^m G(U)_{ii} \langle Ue_i, Ze_i \rangle^2 \tag{332}$$

*Proof.* Note that the gradient of $\ell_2^\beta(v)$ as a function of $v$ is,

$$\nabla L_2^\beta(v) = \frac{2v}{\sqrt{\|v\|_2^2 + \beta}} - \frac{1}{2} \frac{\|v\|_2^2}{(\|v\|_2^2 + \beta)^{3/2}} \cdot (2v) = \frac{(\|v\|_2^2 + 2\beta)v}{(\|v\|_2^2 + \beta)^{3/2}}. \tag{333}$$

Therefore,

$$\langle Z, \nabla \mathcal{R}_\beta(U) \rangle = \sum_{i=1}^d \frac{(\|Ue_i\|_2^2 + 2\beta)}{(\|Ue_i\|_2^2 + \beta)^{3/2}} \langle Ue_i, Ze_i \rangle \tag{334}$$

$$= \sum_{i=1}^d \frac{(\|Ue_i\|_2^2 + 2\beta)}{(\|Ue_i\|_2^2 + \beta)^{3/2}} \text{Tr}\left(Ze_i e_i^T U^T\right) \tag{335}$$

$$= \text{Tr}\left(D(U) \cdot U^T Z\right), \tag{336}$$

where $D(U) = \text{\textbf{diag}}\left(\left\{ \frac{(\|Ue_i\|_2^2 + 2\beta)}{(\|Ue_i\|_2^2 + \beta)^{3/2}} : i \in [m] \right\}\right)$ is as defined in eq. (330).

On the other hand, the Hessian of $\ell_2^\beta(v)$ as a function of $v$ is,

$$\nabla^2 \ell_2^\beta(v) = \frac{(\|v\|_2^2 + 2\beta)}{(\|v\|_2^2 + \beta)^{3/2}} I - \frac{3(\|v\|_2^2 + 2\beta)}{(\|v\|_2^2 + \beta)^{5/2}} vv^T + \frac{2}{(\|v\|_2^2 + \beta)^{3/2}} vv^T \tag{337}$$

$$= \frac{(\|v\|_2^2 + 2\beta)}{(\|v\|_2^2 + \beta)^{3/2}} I - \frac{\|v\|_2^2 + 4\beta}{(\|v\|_2^2 + \beta)^{5/2}} vv^T. \tag{338}$$

The Hessian of $\mathcal{R}_\beta$ is block diagonal with the $i^{th}$ block equal to $\nabla^2 \ell_2^\beta(Ue_i)$. Therefore,

$$\text{\textbf{vec}}(Z)^T [\nabla^2 \mathcal{R}_\beta(U)] \text{\textbf{vec}}(Z) = \sum_{i=1}^m \frac{(\|Ue_i\|_2^2 + 2\beta)}{(\|Ue_i\|_2^2 + \beta)^{3/2}} \|Ze_i\|_2^2 - \frac{\|Ue_i\|_2^2 + 4\beta}{(\|Ue_i\|_2^2 + \beta)^{5/2}} \langle Ue_i, Ze_i \rangle^2 \tag{339}$$

$$= \sum_{i=1}^m D(U)_{ii} \|Ze_i\|_2^2 - G(U)_{ii} \langle Ue_i, Ze_i \rangle^2 \tag{340}$$

$$= \text{Tr}(D(U) \cdot ZZ^T) - \sum_{i=1}^m G(U)_{ii} \langle Ue_i, Ze_i \rangle^2 \tag{341}$$

$\square$

Finally we introduce a lemma bounding the entries of $D(U)$ and $G(U)$.

**Lemma 44.** *Suppose for some $i \in [k]$, $\|Ue_i\|_2 \geq 2\sqrt{\beta}$. Then the corresponding diagonal entries of $D(U)$ and $G(U)$ satisfy,*

$$(D(U))_{ii} \geq \frac{1}{\|Ue_i\|_2} \tag{342}$$

$$(G(U))_{ii} \geq \frac{1}{\|Ue_i\|_2^3}. \tag{343}$$

*On the other hand, for any $U$ such that $\|U\|_{op} \leq 3$,*

$$(D(U))_{ii} \leq \frac{\|Ue_i\|_2^2 + 2\beta}{(\|Ue_i\|_2^2 + \beta)^{3/2}} \tag{344}$$

*Proof.* The proof for eq. (342) follows by observing that $D(U)_{ii} = \frac{\|Ue_i\|_2^2 + 2x}{(\|Ue_i\|_2^2 + x)^{3/2}}$ where $x = \beta$. Differentiating we observe that the derivative of function in $x$ is,

$$\frac{2(\|Ue_i\|_2^2 + x) - 3/2(\|Ue_i\|_2^2 + 2x)}{(\|Ue_i\|_2^2 + x)^{5/2}} = \frac{\frac{1}{2}\|Ue_i\|_2^2 - x}{(\|Ue_i\|_2^2 + x)^{5/2}} \tag{345}$$

which is increasing as long as $x \leq \frac{1}{2}\|Ue_i\|_2^2$. Note that when $\|Ue_i\|_2 \geq 2\sqrt{\beta} \implies x = \beta \leq \frac{1}{4}\|Ue_i\|_2^2$, the function is increasing in $x$ and therefore the minimum is achieved with $\beta = 0$. This results in the lower bound,

$$(D(U))_{ii} \geq \frac{1}{\|Ue_i\|_2}. \tag{346}$$

Likeiwse, $G(U) = \frac{\|Ue_i\|_2^2 + 4x}{(\|Ue_i\|_2^2 + x)^{5/2}}$ where $x = \beta$, and differentiating in $x$, we get,

$$\frac{4(\|Ue_i\|_2^2 + x) - 5/2(\|Ue_i\|_2^2 + 2x)}{(\|Ue_i\|_2^2 + x)^{5/2}} = \frac{\frac{3}{2}\|Ue_i\|_2^2 - x}{(\|Ue_i\|_2^2 + x)^{5/2}} \tag{347}$$

which is increasing as long as $x \leq \frac{3}{2}\|Ue_i\|_2^2$. Yet again, the function is in the increasing regime in $x = \beta$ under the constraint $2\sqrt{\beta} \leq \|Ue_i\|_2$; the minimum is achieved at $\beta = 0$, which results in eq. (343). $\square$

## F Implications for shallow neural networks with quadratic activation functions

The extension of results from the matrix sensing model to the training of quadratic neural networks was previously carried out in [19, Section 5] and originally in [21], which we explain in more detail below. Indeed, when the measurement matrices are of the form $A_i = x_i x_i^T$ for some vector $x_i \in \mathbb{R}^d$, the functional representation of the output can be written as $\langle A_i, UU^T \rangle = \sum_{i=1}^{k} \sigma(\langle x_i, Ue_i \rangle)$, where $\sigma(\cdot) = (\cdot)^2$ takes the form of a 1-hidden layer shallow network with quadratic activations, with the output layer frozen as all 1's. The columns of $U$ correspond to the weight vectors associated with individual neurons of the network. Likewise, sparsity in the column domain corresponds to learning networks with only a few non-zero neurons. In this section we provide a high level sketch of how results for matrix sensing can be extended for the training of neural networks with quadratic activations. We avoid going through the formal details for the sake of simplicity and ease of exposition.

Why can the results proved for matrix sensing not directly be applied here? It turns out that when the $x_i$ are i.i.d. Gaussian vectors, even if $n \to \infty$, rank 1 measurements do not satisfy the restricted isometry property. In particular, $\frac{1}{n}\sum_{i=1}^{n}\langle A_i, X\rangle^2 \not\approx \|X\|_F^2$. However, these measurements satisfy a different form of low rank approximation, as established in [19, Lemma 5.1] and as we informally state below. In particular, when $x_i \sim \mathcal{N}(0, I)$ are Gaussian, for any matrix $X \in \mathbb{R}^{d \times d}$,

$$\frac{1}{n}\sum_{i=1}^{n}\langle A_i, X\rangle^2 \approx 2\|X\|_F^2 + (\text{Tr}(X))^2 \tag{348}$$

where the approximation becomes exact as $n \to \infty$. In particular, with the choice of $X = UU^T - U_\star U_\star^T$, the mean squared error of the learner training a quadratic neural network takes the form,

$$\mathcal{L}_{\text{NN}}(U) = \frac{1}{n}\sum_{i=1}^{n}\left(\langle A_i, UU^T\rangle - \langle A_i, U_\star U_\star^T\rangle\right)^2 \approx 2\|UU^T - U_\star U_\star^T\|_F^2 + \left(\|U\|_F^2 - \|U_\star\|_F^2\right)^2$$

Notice that the RHS is, up to scaling factors, the mean squared error for matrix sensing, with an additional loss $(\|U\|_F^2 - \|U_\star\|_F^2)^2$ added to it. This loss can easily be estimated since it only relies on estimating a scalar, $\|U_\star\|_F^2$, which is approximated by $\frac{1}{n}\sum_{i=1}^{n} y_i = \frac{1}{n}\sum_{i=1}^{n}\langle x_i x_i^T, U_\star U_\star^T\rangle + \varepsilon_i \approx \|U_\star\|_F^2$. In the sequel, we assume that $\|U_\star\|_F$ was known exactly and consider the loss,

$$f_{\text{NN}}(U) = \mathcal{L}_{\text{NN}}(U) - \left(\|U\|_F^2 - \|U_\star\|_F^2\right)^2 + \mathcal{R}_\beta(U) \tag{349}$$

which subtracts the "correction term" $(\|U\|_F^2 - \|U_\star\|_F^2)^2$ from the mean squared error $\mathcal{L}_{\text{NN}}(U)$ and also adds back the group Lasso regularizer on $U$. Overall $f_{\text{NN}}(U)$ approximately equals $2\|UU^T - U_\star U_\star^T\|_F^2 + \mathcal{R}_\beta(U)$ and therefore, running perturbed gradient descent on this loss and reusing the analysis for matrix sensing shows that the algorithm eventually converges to a solution $UU^T \approx U_\star U_\star^T$ and such that $U$ has approximately $r$ columns.