# OpenReview forum: "Greedy Pruning with Group Lasso Provably Generalizes for Matrix Sensing"
_NeurIPS.cc/2023/Conference — NeurIPS 2023 poster_

### Official Review · Reviewer_J6kp · 2023-06-22

**Soundness:** 3 good
**Presentation:** 3 good
**Contribution:** 3 good
**Rating:** 7
**Confidence:** 4

**Summary:**

This paper studies the pruning for the matrix sensing problem. Particularly, they study the over-parameterized symmetric matrix sensing problem. They show that the approximate solution from a smoothed version of the group-Lasso problem can be pruned to small error values using a greedy pruning algorithm. If the pruned solution is further fine-tuned, the reconstruction error can be further reduced.

**Strengths:**

- The results that show fine-tuning a local minimum can further improve its quality is interesting given the recent similar experimental observations in deep learning.
- Thm. 3 does not require global optimality which is interesting.

**Weaknesses:**

- Thm. 1 concerns the continuous time version of gradient descent (gradient flow) which would limit the applicability of the theorem in practice. However, this should be fine overall.
- The penalty discussed in Section 4 is somewhat arbitrary. I think the authors should motivate the choice of this penalty better. Is this penalty used in practice?
- Verifying SOSP property for a solution can be hard in practice.





**Questions:**

- Is the upper bound of Thm. 3 tight? In other words, is fine-tuning necessary?

---

> ### Author Rebuttal · Authors · 2023-08-09
>
> >  [1.] The penalty discussed in Section 4 is somewhat arbitrary. I think the authors should motivate the choice of this penalty better. Is this penalty used in practice?
>
> The penalty we discuss in Section 4 is simply a smoothing of the $\ell_2$ norm (which is not differentiable at $0$). In particular, by replacing $\| v \|_2 \approx \frac{\| v \|_2^2}{\sqrt{\| v \|_2^2 + \delta}}$ for a small $\delta > 0$, we now have a loss which is differentiable around $0$. This relaxation of the group LASSO objective allows us to study the stationary points of a non-convex, but now smooth objective.
>
> > [2.] Verifying SOSP property for a solution can be hard in practice.
>
> Verifying the SOSP property is indeed hard in practice, but for an algorithm like perturbed gradient descent, there is no need to have to check this property as the algorithm is theoretically guaranteed to reach such solutions. In practice, stochastic gradient methods (operating on batches of data) are implemented, which bears some resemblance to the perturbed gradient descent we study in this paper.
>
> > [3.] Is the upper bound of Thm. 3 tight? In other words, is fine-tuning necessary?
>
> That is a good point. The result of Theorem 3 establishes an upper bound on the error of the pruned solution. However, to demonstrate the necessity of fine-tuning and establishing the tightness of the upper bound, a lower bound would need to be proven. In practical scenarios, we consistently observe that some degree of fine-tuning is necessary and beneficial, enhancing the quality of the pruned solution. It would be valuable if a lower bound for the error of the pruned model could also be provided to validate the accuracy of the upper bound.
>
> Nevertheless, even if the initial error before fine-tuning matches the lower bound, it can be rapidly reduced during the fine-tuning process, thanks to the linear convergence rate we experience in this phase.

---

> > ### Comment · Reviewer_J6kp · 2023-08-10
> >
> > Thank you for your clarifications.
> >
> > My only comment is that adding some more context (and possibly references to certain theorems proving SOSP for perturbed GD) would be good. I will increase my score.

---

### Official Review · Reviewer_oETd · 2023-07-05

**Soundness:** 2 fair
**Presentation:** 3 good
**Contribution:** 3 good
**Rating:** 5
**Confidence:** 4

**Summary:**

The authors study an algorithm framework for symmetric matrix sensing based on overparametrized Burer-Monteiro matrix factorization which additionally imposes a group lasso penalty on the columns of the factor matrix, and which intertwines a gradient descent-type approach for this objective with iterative pruning.

They argue that gradient descent applied to the unregularized loss does, on the other hand, lead to factor matrices whose generalization properties are negatively affected by magnitude pruning.

As main theoretical contribution, the authors show in Theorem 4 that for a sample complexity whoses dependence is quadratic in the ground truth rank and of order 5 in the overparametrization rank, algorithmic parameters can be chosen appropriately so that after pruning, an approximate second-order stationary point (which can be arguably found via perturbed gradient descent) can be followed by gradient descent to obtain a solution that is close to the ground truth.

A connection to quadratic shallow neural networks is further used to justify similar approaches combing greedy pruning of a regularized loss with post-pruning fine tuning for more complex machine learning models.

**Strengths:**

- The paper provides insights towards relevant insights on how an overparametrized Burer-Monteiro factorization can be made successful and computationally efficient.
- The manuscript is well-structured and clearly written.
- From my point of view, the work provides a theoretical justification of the popular practice in deep neural network training of alternating the pruning weight matrices and fine-tuning steps.
- The technical work to establish the results is quite extensive.
- The benefit of group lasso regularization is for the problem type considered is clearly illustrated in a simple computaitonal experiment.


**Weaknesses:**

- Many of the results in the paper focus on the presented population setting, in which the ground truth is assumed to be known to the algorithm. While they can arguably be seen as paving the road for the setting in which the (regularized) empirical loss minimized, the paper could have benefitted from a clearer focus and presentation on this more realistic case.
- Some relevant literature has not been included in the manuscript. In particular, the proposed group Lasso regularization is presented to be new for the problem in consideration, however, similar regularization and a tailored algorithm was already proposed for the very problem (in the more general rectangular case) in
	- Giampouras, Paris V., Athanasios A. Rontogiannis, and Konstantinos D. Koutroumbas. "Alternating iteratively reweighted least squares minimization for low-rank matrix factorization." _IEEE Transactions on Signal Processing_ 67.2 (2018): 490-503.
	It also already introduced the pruning step in its algorithm.
	Furthermore, a connection of the group Lasso penalty considered with a Schatten-1/2 norm on the product matrix variable was established in:
	- Giampouras, P., Vidal, R., Rontogiannis, A., & Haeffele, B. (2020). A novel variational form of the Schatten-$ p $ quasi-norm. _Advances in Neural Information Processing Systems_ (NeurIPS 2020), _33_, 21453-21463.
- The proof of Theorem 1 is potentially erroneous. In particular, in Appendix A, the quantity $r(t)= U^T U_\star$ is not a vector in $\mathbb{R}^k$ , but a $(k \times k)$-matrix. In the proof, it is consistently treated as a vector, which leads to several problematic statements.

**Questions:**

- $f_\operatorname{emp}$  is somehow only defined on page 7. Mentioning it before the introduction of Algorithm 1 will improve the readability of the paper.
- Can the assumptions of Theorem 4 and Theorem 5 be better clarified? It seems that the RIP is used, it would be better be mentioned in the theorems.
- How do we end up with the sample complexity mentioned in Remark 6?
- In line 314, it is mentioned that linear convergence cannot be established due to ill-conditioning of the objective. Please elaborate on whether this could be mitigated algorithms that take the problem structure better into account than standard gradient descent, such as preconditioned examples (e.g., PrecGD from "Zhang, J., Fattahi, S., & Zhang, R. Y. (2021). Preconditioned gradient descent for over-parameterized nonconvex matrix factorization. _Advances in Neural Information Processing Systems_, _34_, 5985-5996.").

**Limitations:**

The authors mostly covered well the limitations of their approach. In the sample complexity results, it would have been beneficial, if would have been better if the suboptimal parameter dependency (at least on $k$ and $n$). had been mentioned.

---

> ### Author Rebuttal · Authors · 2023-08-09
>
> > Many of the results in the paper focus on the presented population setting ... on this more realistic case.
>
> Thank you for your comment. As the reviewer pointed out, our paper presents results for both the population loss and the finite sample loss. We chose to begin by showcasing the results for the population loss, as it facilitates easier comprehension of the updates and analysis and highlights the intuitions behind the proposed approach (such as eq. (13)). Once this groundwork is established, we leverage some of the intuitions from the population loss to derive our results for the more realistic finite sample setting.
>
> > Some relevant literature has not been included ... (NeurIPS 2020), 33, 21453-21463.
>
> Thank you for bringing these papers to our attention. We would like to clarify that our paper does not claim to be the first to utilize Group Lasso regularization for low-rank matrix factorization; such a claim is never made in our work. Our primary contribution lies in establishing a formal, and theoretical connection between this regularization technique and learning a column sparse model, along with its relevance to pruning. This is not addressed in the existing literature. Notably, our research is the first to present both negative and positive results on the connection of group lasso and sparse models in the overparameterized setting. Prior to our work, there were no negative results showing the failure of gradient descent updates to learn a column-sparse model without this regularization. Similarly, there was no theory demonstrating that, with this regularization, the obtained model not only possesses the correct rank but also exhibits sparse columns, with the number of active columns matching the true rank.
>
>
> That said, we agree with the reviewer that the mentioned papers should be added to the literature review of our paper. We will explicitly highlight the contributions of these papers. For instance, in [Giampouras et al., 2018], the authors proposed a regularizer (which is slightly different from Group Lasso) that enforces column sparsity on the two matrix factors. However, they do not provide a formal characterization of whether such regularization ``provably" leads to a column-sparse model and, if so, whether it allows for exact recovery of the true rank $r$ active columns. Notably, their work proves asymptotic convergence of alternating minimization to a stationary point of the regularized loss. Furthermore, in [Giampouras et al., 2020], the authors introduced a novel variational form of the Schatten-$p$ quasi-norm, which can be considered a direct generalization of the variational form of the nuclear norm. They demonstrated that the local minimizers of the factorized problem also serve as local minimizers of the original problem, and they presented rank-one updates to escape from "bad" stationary points. Despite these contributions, similar to the other work, this paper does not establish that the local minimizer of the regularized problem or the obtained model using their proposed method would lead to a model with only $r$ columns that have a large norm, ensuring column sparsity.
>
> > The proof of Theorem 1 is potentially erroneous...
>
> We respectfully disagree with this comment, as we have specifically mentioned in our paper that the proof of Theorem 1 focuses on the case that the correct rank is $r=1$. Hence, there is no error in the proof. The result shows thar vanilla gradient descent results in a column-dense solution that cannot be easily pruned even when the underlying ground truth is rank $1$.
>
> >Clarifying minor comments: [1.] Can the assumptions of Theorem 4 ...
>
> The reviewer is correct and we will explicitly mention in the statement of these theorems that our results in Theorems 4 and 5 require the RIP condition. Thanks for raising this point.
>
> > [2.] How do we end up ...
>
> The sample complexity bound in Remark 6 follows by combining the sample complexity bound in Theorems 4 and 5. The bound comes from adding up the $\widetilde{O} \left(\frac{\sigma^2}{\left(\sigma_r^{\star}\right)^4} d k^2 r^5 \right)$ samples required in the pruning phase (Theorem 4) + the $O \left(\frac{\sigma^2}{(\sigma_r^\star)^4} \frac{rd}{\varepsilon^2} \right)$ samples required in the fine-tuning phase (Theorem 5, eq. (16)) to get to $\varepsilon$ error. Note that under Gaussian measurements the RIP condition in Theorem 4 for $\delta \le (\sigma_r^\star)^{3/2}/\sqrt{k} r^{5/2}$ is satisfied with these many measurements.
>
> > [3.] In line 314, it is mentioned ...
>
> The reviewer's comment is correct, however, our claim was stated specifically for gradient descent. There are a few papers proposing algorithms that circumvent the ill-conditioning of the objective, including Zhang et al (2021). We will rephrase this statement to refer to the above paper. The point we were trying to make here is that gradient descent itself no longer suffers from this slowdown after pruning is carried out since ill-conditioning is only an issue for the overparameterized case. Thanks for your comment.

---

> > ### Comment · Reviewer_oETd · 2023-08-16
> >
> > Thank you for the clarifications. Regarding Theorem 1, I indeed missed that you restricted yourselves to the case of $r=1$, in which case the stated matrices can indeed interpreted as vectors.
> > The concerns I had have been also addressed, if the literature review is updated accordingly in the final version.
> > I am accordingly increasing my score by 1.

---

> ### Comment · Area_Chair_Vy3X · 2023-08-16
>
> Thanks for your review, the authors have provided a rebuttal. Please reply below and let us know if your concerns are addressed, in particular the question about the correctness of Theorem 1.
>
> Best,
> the AC

---

### Official Review · Reviewer_mDyb · 2023-07-06

**Soundness:** 4 excellent
**Presentation:** 4 excellent
**Contribution:** 3 good
**Rating:** 7
**Confidence:** 3

**Summary:**

In "Greedy Pruning with Group Lasso Provably Generalizes for Matrix Sensing" the authors propose a two phase scheme to solve the noisy overparametrized matrix sensing problem. The goal is not to only solve the noisy overparametrized matrix sensing problem but also to provide a theoretically supported intuition for pruning + refining schemes used to train more complex learning models. More specifically, the authors establish that pruning --which entails setting to zero columns with norm below a certain threshold-- any given approximate second order stationary point of the group-lasso regularized matrix sensing problem yields a point with sufficiently small generalization error allowing the use gradient descent and achieving optimal statistical precision.

**Strengths:**

The paper is very well written and easy to follow. The problem considered (noisy overparametrized noisy matrix sensing)  is relevant by itself and additionally can provide insights for more complicated learning models which can be of great interest to the community. The results ar

Pruning as a technique to solve this specific problem is very well motivated based on both Theorem 1 and achievable statistical precision in the overparametrized setting (c.f. line 312).

**Weaknesses:**

In Theorem 3, the regularizer weight and the thresholding for pruning are both explicitly dependent on the target rank r. Thus, knowledge of r is required to a degree anyway. Consequently, the comparison made to the overparametrized setting from [23] does not seem entirely fair.

**Questions:**

1. The authors suggest the use of perturbed gradient descent to find an approximate second order stationary point. I think it would be a good idea, for completeness, to write the number of oracle calls required to achieve the desired precision using this scheme.

---

> ### Author Rebuttal · Authors · 2023-08-09
>
> We thank the reviewer for their comments.
>
> Although the stated bounds in Theorem 3 explicitly use $r$ and $\sigma^\star_r$, this is just for simplicity of expression. It is possible to obtain bounds which rely on $\lambda$ and $\beta$ lying in a particular range without explicit knowledge of $r$ and $\sigma^\star_r$. Results of this form are the standard de-facto even for the related problem of sparse regression (LASSO). See eg. Theorem 1 of Loh et al (2013). That said, if we are off by a constant factor in each of these parameters, we can still recover the current theorem statement with the sample complexity guarantee larger by a constant.
>
> We shall also include a bound on the number of optimization steps required by perturbed gradient descent to find an $(\epsilon,\gamma)$-SOSP. The authors of [1] show that it is upper bounded by $\widetilde{O} (1/\varepsilon^2)$ for a Lipschitz and $\rho$-Hessian smooth objective when $\varepsilon= \min \\{\epsilon, \gamma^2/\rho\\}$.
>
> [1] Jin et al., 2017. How to Escape Saddle Points Efficiently

---

> > ### Comment · Reviewer_mDyb · 2023-08-14
> >
> > Thank you for addressing my comments. I will keep my current score.

---

### Decision · Program_Chairs · 2023-09-21

**Decision:**

Accept (poster)

**Comment:**

This work provides a theoretical justification for greedy pruning, using group lasso in a matrix sensing application. The problem is of interest and the reviewers' concerns were addressed in the discussion period. Please incorporate the changes requested by the reviewers in the final camera ready revision, e.g. "adding some more context (and possibly references to certain theorems proving SOSP for perturbed GD)".